  SciPost Phys. Lect. Notes 19 (2020)

# Introduction to the nested algebraic Bethe ansatz

**N. A. Slavnov**$^\star$

Steklov Mathematical Institute of Russian Academy of Sciences, Moscow, Russia

$\star$ nslavnov@mi-ras.ru

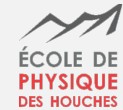

*Part of the Integrability in Atomic and Condensed Matter Physics*
*Session 111 of the Les Houches School, August 2018*
*published in the Les Houches Lecture Notes Series*

## Abstract

We give a detailed description of the nested algebraic Bethe ansatz. We consider integrable models with a $\mathfrak{gl}_3$-invariant $R$-matrix as the basic example, however, we also describe possible generalizations. We give recursions and explicit formulas for the Bethe vectors. We also give a representation for the Bethe vectors in the form of a trace formula.

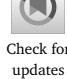
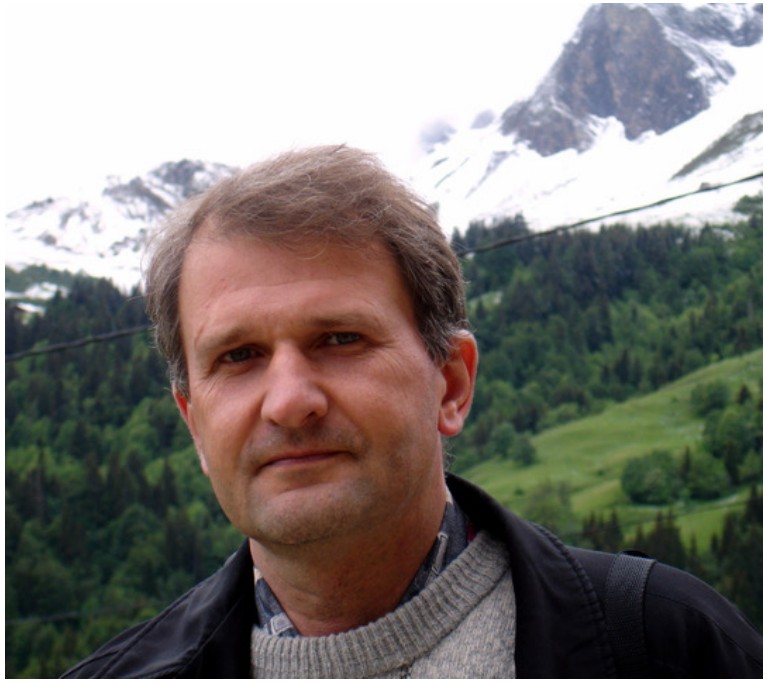

# 1 Introduction

Algebraic Bethe ansatz (ABA) is a part of the Quantum Inverse Scattering Method (QISM), that emerged in the late 70's in the works of the Leningrad School [1, 2]. Almost simultaneously with this method, a nested algebraic Bethe ansatz (NABA) was developed in [3–6]. The NABA is a method that allows us to find the spectrum of quantum integrable models describing systems with several types of excitations. It is an algebraic interpretation of the approach proposed in the works [7–9].

These notes are based on a series of lectures given by the author at the Les Houches summer school 2018 *Integrability in Atomic and Condensed Matter Physics*. We introduce the reader to the basic principles of the NABA. The presentation follows the classical scheme described in the papers [3–6]. However, we give much more details and illustrate the general principles with concrete calculations. To simplify the discussion, we will mainly confine ourselves to the case of models, which are described by a $\mathfrak{gl}_3$-invariant *R*-matrix. A number of statements, however, are formulated for a fairly general case. Besides, we give a number of comments on how one can generalize the results obtained for the $\mathfrak{gl}_3$ case to the models with symmetries of higher ranks.

We also describe a method developed in the works [10–12]. In contrast to the classical NABA scheme, which allows to construct Bethe vectors recursively, this approach immediately sets explicit formulas for these vectors. We present a proof of the equivalence of this approach to the NABA.

In these notes, we focus on the mathematical aspects of NABA and do not consider the physical applications of models solvable by this method. Note, however, that these physical applications are very wide, since the NABA models provide a more realistic description of strongly interacting systems. The reason is that in the NABA we are dealing with several creation operators. This allows us to consider systems where several degrees of freedom of fundamental particles interact, for example, the spin and the charge of the electrons. Therefore, the NABA solvable models have found wide application primarily in the physics of strongly correlated electronic systems (Yang–Gaudin model [7, 13–15], t-J model and Hubbard model [16–19]). We can also consider systems consisting of several types of particles, such as systems with impurities (Kondo model) [20–23]. For a more detailed description of the application of NABA to Fermi gases and ultracold atom systems, we refer the reader to review [24]. It is also worth mentioning that the Hamiltonians of integrable systems with a large number of degrees of freedom arise in supersymmetric gauge theories [25].

To conclude this short introduction, we would like to mention that the NABA is a generalization of the ABA and uses basically the same concepts. Some techniques are also borrowed from the ABA. Therefore, to understand the stuff, the reader must possess the basic principles and techniques of the ABA. In addition to the original works mentioned above, they can be found in [26–29].

## 1.1 Reminder of the algebraic Bethe ansatz

A key equation of the ABA is an *RTT*-relation [1, 2, 26–29]

$$R(u,v)\bigl(T(u)\otimes I\bigr)\bigl(I\otimes T(v)\bigr) = \bigl(I\otimes T(v)\bigr)\bigl(T(u)\otimes I\bigr)R(u,v). \tag{1.1}$$

Here $T(u)$ is a monodromy matrix

$$T(u) = \begin{pmatrix} A(u) & B(u) \\ C(u) & D(u) \end{pmatrix}, \tag{1.2}$$

whose matrix elements act in some Hilbert space $\mathcal{H}$. The monodromy matrix also acts in the space $\mathbb{C}^2$ which is called an auxiliary space. The $R$-matrix $R(u, v)$ acts in $\mathbb{C}^2 \otimes \mathbb{C}^2$. Another commonly used form of equation (1.1) is

$$R_{12}(u, v)T_1(u)T_2(v) = T_2(v)T_1(u)R_{12}(u, v). \tag{1.3}$$

Here the subscripts show in which of the two auxiliary spaces $\mathbb{C}^2$ the $T$-matrices act nontrivially. The $R$-matrix $R_{12}(u, v)$ acts in both spaces $\mathbb{C}^2$.

The $RTT$-relation immediately yields

$$[\operatorname{tr} T(u), \operatorname{tr} T(v)] = 0, \qquad \operatorname{tr} T(u) = A(u) + D(u). \tag{1.4}$$

Thus, the trace of the monodromy matrix in the auxiliary space (transfer matrix) is a generating function of commuting operators

$$\operatorname{tr} T(u) = \sum_k (u - u_0)^k I_k, \qquad [I_k, I_n] = 0. \tag{1.5}$$

Choosing one of $I_k$ as a Hamiltonian of a quantum model we automatically obtain many (generically, infinitely many) integrals of motion. Thus, we have a chance to build an integrable model.

It is assumed within the framework of the ABA that the Hilbert space $\mathcal{H}$ of the model contains a vacuum vector $|0\rangle$ with the following properties:

$$A(u)|0\rangle = a(u)|0\rangle, \qquad D(u)|0\rangle = d(u)|0\rangle, \qquad C(u)|0\rangle = 0. \tag{1.6}$$

Here $a(u)$ and $d(u)$ are some functions that depend on the particular model. Common eigenstates of the Hamiltonian and other integrals of motion are eigenstates of the transfer matrix for arbitrary complex $z$:

$$\operatorname{tr} T(z)|\Psi\rangle = \Lambda(z)|\Psi\rangle, \tag{1.7}$$

where $\Lambda(z)$ is the transfer matrix eigenvalue. Within the framework of the ABA they can be obtained by the successive action of the operators $B(u)$ on the vacuum vector:

$$|\Psi\rangle = B(u_1)\dots B(u_n)|0\rangle, \tag{1.8}$$

provided parameters $u_1, \dots, u_n$ satisfy a system of Bethe equations (see below). In this case, we call the vector (1.8) an *on-shell Bethe vector*. Otherwise, if parameters $u_1, \dots, u_n$ are arbitrary complex numbers, we call the vector (1.8) an *off-shell Bethe vector* or simply the *Bethe vector*.

## 1.2 Possible generalization

A question arises: can we generalize this construction to the case of the $N \times N$ monodromy matrix whose auxiliary space would be $\mathbb{C}^N$? Namely, we still want to have the $RTT$-relation (1.1). Then, the transfer matrix

$$\operatorname{tr} T(u) = \sum_{i=1}^N T_{ii}(u) \tag{1.9}$$

satisfies the commutation relation (1.4). Thus, we can obtain a Hamiltonian and other integrals of motion via (1.5). In order to construct the Hamiltonian eigenstates we assume that the Hilbert space of the model has a vacuum vector $|0\rangle$ with the properties analogous to (1.6):

$$
\begin{aligned}
T_{ii}(u)|0\rangle &= \lambda_i(u)|0\rangle, \qquad i = 1, \ldots, N, \\
T_{ij}(u)|0\rangle &= 0, \qquad i > j.
\end{aligned}
\tag{1.10}
$$

Here $\lambda_i(u)$ are some functions dependent on the particular model.

## 1.3 Examples of $R$-matrices

The first problem is to find an $R$-matrix acting in $\mathbb{C}^N \otimes \mathbb{C}^N$. The $R$-matrix should satisfy the Yang–Baxter equation

$$
R_{12}(u_1, u_2)R_{13}(u_1, u_3)R_{23}(u_2, u_3) = R_{23}(u_2, u_3)R_{13}(u_1, u_3)R_{12}(u_1, u_2),
\tag{1.11}
$$

in order to provide compatibility of the $RTT$-relation. The first example of the non-trivial $R$-matrix has exactly the same form as in the case of the $\mathbb{C}^2$ auxiliary space:

$$
R(u, v) = \mathbb{I} + g(u, v)P, \qquad g(u, v) = \frac{c}{u - v}.
\tag{1.12}
$$

Here $\mathbb{I}$ is the identity operator in $\mathbb{C}^N \otimes \mathbb{C}^N$, $P$ is the permutation operator in the same space, and $c$ is a constant. The permutation operator has the form

$$
P = \sum_{i,j=1}^{N} E_{ij} \otimes E_{ji},
\tag{1.13}
$$

where $(E_{ij})_{lk} = \delta_{il}\delta_{jk}$, $i, j, l, k = 1, \ldots, N$ are $N \times N$ matrices with unit at the intersection of $i$th row and $j$th column and zeros elsewhere (the standard basis matrices).

Another solution to the Yang–Baxter equation is given by the $q$-deformation of the $R$-matrix (1.12) [4, 30, 31]:

$$
\begin{aligned}
R^{(q)}(u, v) = f_q(u, v) \sum_{1 \le i \le N} E_{ii} \otimes E_{ii} \; &+ \sum_{1 \le i < j \le N} (E_{ii} \otimes E_{jj} + E_{jj} \otimes E_{ii}) \\
&+ \sum_{1 \le i < j \le N} \left( u g_q(u, v)E_{ij} \otimes E_{ji} + v g_q(u, v)E_{ji} \otimes E_{ij} \right),
\end{aligned}
\tag{1.14}
$$

where

$$
f_q(u, v) = \frac{qu - q^{-1}v}{u - v}, \qquad g_q(u, v) = \frac{q - q^{-1}}{u - v}.
\tag{1.15}
$$

Pay attention that this $R$-matrix is not a complete analog of the well known $4 \times 4$ trigonometric $R$-matrix acting in $\mathbb{C}^2 \otimes \mathbb{C}^2$. Indeed, the latter has the following form[1]:

$$
R^{\text{trig}}(u, v) = \begin{pmatrix} f_q(u, v) & 0 & 0 & 0 \\ 0 & 1 & \sqrt{uv}g_q(u, v) & 0 \\ 0 & \sqrt{uv}g_q(u, v) & 1 & 0 \\ 0 & 0 & 0 & f_q(u, v) \end{pmatrix}.
\tag{1.16}
$$

---

[1]For those who are used to write this matrix in terms of trigonometric (hyperbolic) functions, it is enough to substitute $u = e^{2x}$, $v = e^{2y}$, and $q = e^{\eta}$ in (1.15).

One would expect that an analog of $R^{\text{trig}}(u,v)$ in the case $\mathbb{C}^N \otimes \mathbb{C}^N$ is

$$
\begin{aligned}
R^{\text{trig}}(u,v) = f_q(u,v) \sum_{1\le i\le N} E_{ii}\otimes E_{ii} \;+\; & \sum_{1\le i<j\le N}(E_{ii}\otimes E_{jj}+E_{jj}\otimes E_{ii}) \\
+\; & \sum_{1\le i<j\le N}\sqrt{uv}\,g_q(u,v)\big(E_{ij}\otimes E_{ji}+E_{ji}\otimes E_{ij}\big).
\end{aligned}
\tag{1.17}
$$

However, the $R$-matrix (1.17) satisfies the Yang–Baxter equation for $N=2$ only. The point is that the $R$-matrix (1.14) can be transformed as follows:

$$
\tilde{R}^{(q)}_{12}(u,v) = K_1\!\left(\tfrac{u}{v}\right)R^{(q)}_{12}(u,v)K_1\!\left(\tfrac{v}{u}\right),
\tag{1.18}
$$

where

$$
K\!\left(\tfrac{u}{v}\right) = \sum_{j=1}^{N}\left(\tfrac{u}{v}\right)^{(N+1)/4-j/2} E_{jj}.
\tag{1.19}
$$

Then the new matrix $\tilde{R}^{(q)}(u,v)$ is also a solution of the Yang–Baxter equation (see appendix A). It is easy to check that for $N=2$ the matrix $\tilde{R}^{(q)}(u,v)$ (1.18) coincides with $R^{\text{trig}}(u,v)$ (1.16), but this is not true for $N>2$.

There exist, of course, other $R$-matrices acting in $\mathbb{C}^N \otimes \mathbb{C}^N$ and satisfying the Yang–Baxter equation, for example, Belavin elliptic $R$-matrix [32–35]. However, we will restrict our selves with consideration of the simplest $R$-matrix (1.12) only. Furthermore, the main part of these lectures will be devoted to the case $N=3$. We will see that even in this simplest case one should solve several non-trivial problems.

The $R$-matrix (1.12) is called $\mathfrak{gl}_N$-invariant due to the property

$$
[R_{12}(u,v),G_1G_2]=0,
\tag{1.20}
$$

for any $G\in\mathfrak{gl}_N$.

## 1.4 Examples of monodromy matrices

The first example of the monodromy matrix is completely analogous to the $\mathfrak{gl}_2$ case

$$
T(u) = R_{0L}(u,\xi_L)\dots R_{01}(u,\xi_1).
\tag{1.21}
$$

This is the monodromy matrix of the $SU(N)$-invariant inhomogeneous $XXX$ Heisenberg chain. The parameters $\xi_i$ are inhomogeneities. Each $R$-matrix $R_{0i}(u,\xi_i)$ acts in the tensor product $V_0\otimes V_i$, where every $V_i$ is $\mathbb{C}^N$. The auxiliary space of the monodromy matrix is $V_0\sim\mathbb{C}^N$. The quantum space is

$$
\mathcal{H}=V_1\otimes\cdots\otimes V_L=\underbrace{\mathbb{C}^N\otimes\cdots\otimes\mathbb{C}^N}_{L \quad \text{times}}.
\tag{1.22}
$$

This quantum space has a vacuum vector of the form

$$
|0\rangle = \underbrace{\begin{pmatrix}1\\0\\\vdots\\0\end{pmatrix}\otimes\cdots\otimes\begin{pmatrix}1\\0\\\vdots\\0\end{pmatrix}}_{L \quad \text{times}}.
\tag{1.23}
$$

Another example describes a system of bosons. For simplicity we give explicit formulas for the $\mathfrak{gl}_3$ case [4] (generalization to $\mathfrak{gl}_N$ is quite obvious). An $L$-operator of this system has the following form:

$$L^{(a)}(u) = u\mathbf{1} - c\mathcal{L}, \tag{1.24}$$

where $\mathbf{1}$ is the identity operator and

$$\mathcal{L} = \begin{pmatrix} a_1^\dagger a_1 & a_1^\dagger a_2 & ia_1^\dagger\sqrt{m+\rho} \\ a_2^\dagger a_1 & a_2^\dagger a_2 & ia_2^\dagger\sqrt{m+\rho} \\ i\sqrt{m+\rho}\,a_1 & i\sqrt{m+\rho}\,a_2 & -m-\rho \end{pmatrix}. \tag{1.25}$$

Here $m$ is a complex number and $\rho = a_1^\dagger a_1 + a_2^\dagger a_2$. The operators $a_k$ and $a_k^\dagger$ ($k = 1, 2$) act in a Fock space with the Fock vacuum $|0\rangle$: $a_k|0\rangle = 0$. They have standard commutation relations of the Heisenberg algebra $[a_i, a_k^\dagger] = \delta_{ik}$.

In order to construct the monodromy matrix we replace the original $a_k$ and $a_k^\dagger$ operators with $a_k(n)$ and $a_k^\dagger(n)$ so that $[a_i(n), a_k^\dagger(m)] = \delta_{ik}\delta_{nm}$. Then

$$T(u) = L_M(u)\ldots L_1(u), \quad \text{where} \quad L_i(u) = L^{(a)}(u)\Big|_{\substack{a_k=a_k(i) \\ a_k^\dagger=a_k^\dagger(i)}}. \tag{1.26}$$

This monodromy matrix describes a chain in each site of which there may be a particle of one of two sorts. The vacuum vector coincides with the Fock vacuum. This system admits the continuum limit. Then it turns into the system of the two-component Bose-gas with $\delta$-function interaction [7–9, 36].

## 1.5 Remark about $RTT$-algebra

$RTT$-algebra (1.1) with the $R$-matrix (1.12) is closely related to the concept of Yangian $Y(\mathfrak{gl}_N)$ (see [37, 38] and references therein). Sometimes in the literature it is called the Yangian. In fact, the $RTT$-algebra with the $\mathfrak{gl}_N$-invariant $R$-matrix is somewhat wider. In the case of the Yangian, we must impose an additional condition on the asymptotic behavior of the monodromy matrix elements $T_{ij}(u)$ at $u \to \infty$

$$T_{ij}(u) = \delta_{ij} + \sum_{k=0}^{\infty}\left(\frac{c}{u}\right)^{k+1} T_{ij}[k], \qquad u \to \infty, \tag{1.27}$$

where $T_{ij}[k]$ are generators of the Yangian $Y(\mathfrak{gl}_N)$. Note that the examples of the monodromy matrices considered in section 1.4 enjoy this condition after appropriate normalization. However, if we pass from the monodromy matrix $T(u)$ satisfying condition (1.27) to a matrix $KT(u)$, where $K$ is a diagonal $c$-number matrix, then the new matrix $KT(u)$ will also satisfy the $RTT$-relation. The properties of the vacuum vector will also be preserved. However, the $KT(u)$ matrix no longer has expansion (1.27), since it does not begin with the identity operator. This type of transformation (twist transformation) can be done with any of the $L$-operators entering the definition of the monodromy matrix. As a result, in some cases the monodromy matrix satisfying the $RTT$-relation may have essential singularity at infinity (see e.g. continuum models of one-dimensional Bose and Fermi gases [26, 36]).

Below we denote by $\mathcal{R}_N$ the $RTT$-algebra with the $\mathfrak{gl}_N$-invariant $R$-matrix (1.12), where $N$ indicates the size of the monodromy matrix. Starting from this point, we consider only such algebras, unless otherwise specified.

## 1.6 Automorphism

Let us define a linear mapping $\varphi : T \to \tilde{T}$ such that

$$
\tilde{T}_{ij} = \varphi\big(T_{ij}(u)\big) = T_{\tilde{j},\tilde{i}}(-u), \qquad \text{where} \qquad \tilde{i} = N + 1 - i,
$$
$$
\varphi\big(T_{ij}(u)T_{kl}(v)\big) = \varphi\big(T_{ij}(u)\big)\varphi\big(T_{kl}(v)\big).
$$
(1.28)

**Proposition 1.1.** *[38] Mapping* (1.28) *is an automorphism of the* $\mathcal{R}_N$ *algebra .*

To prove this proposition we need an auxiliary lemma.

**Lemma 1.1.** *The $RTT$-relation* (1.3) *with the R-matrix* (1.12) *implies*

$$
[T_{ij}(u), T_{kl}(v)] = g(u,v)\big(T_{kj}(v)T_{il}(u) - T_{kj}(u)T_{il}(v)\big),
$$
$$
= g(u,v)\big(T_{il}(u)T_{kj}(v) - T_{il}(v)T_{kj}(u)\big).
$$
(1.29)

*Proof.* Observe that the second equation (1.29) can be obtained from the first one via simultaneous replacements $i \leftrightarrow k$, $j \leftrightarrow l$, and $u \leftrightarrow v$. Thus, it is enough to prove the first equation (1.29).

Let us write down the $RTT$-relation (1.3) in the form

$$
[T_1(u), T_2(v)] = g(u,v)\big(T_2(v)T_1(u)P_{12} - P_{12}T_1(u)T_2(v)\big).
$$
(1.30)

Here we used representation (1.12) for the $R$-matrix.

The monodromy matrices $T_1(u)$ and $T_2(v)$ can be written as

$$
T_1(u) = \sum_{i,j=1}^{N} T_{ij}(u)E_1^{ij}, \qquad T_2(v) = \sum_{k,l=1}^{N} T_{kl}(v)E_2^{kl}.
$$
(1.31)

Here we used superscripts to denote different standard basis matrices, since the subscripts are already occupied for the designation of auxiliary spaces. The matrix elements $T_{ij}$ (or $T_{kl}$) act in the Hilbert space $\mathcal{H}$ only, while the standard basis matrices act in the auxiliary spaces $\mathbb{C}^N$. Recall also that

$$
P_{12} = \sum_{a,b=1}^{N} E_1^{ab}E_2^{ba}.
$$
(1.32)

Now we simply substitute (1.31) and (1.32) into (1.30). Then we obtain in the l.h.s.

$$
[T_1(u), T_2(v)] = \sum_{i,j,k,l=1}^{N} [T_{ij}(u), T_{kl}(v)]E_1^{ij}E_2^{kl}.
$$
(1.33)

In the r.h.s., we have

$$
g(u,v)\big(T_2(v)T_1(u)P_{12} - P_{12}T_1(u)T_2(v)\big)
$$
$$
= g(u,v)\sum_{i,j,k,l,a,b=1}^{N} \big(T_{kl}(v)T_{ij}(u)E_1^{ij}E_2^{kl}E_1^{ab}E_2^{ba} - T_{ij}(u)T_{kl}(v)E_1^{ab}E_2^{ba}E_1^{ij}E_2^{kl}\big).
$$
(1.34)

Multiplying the standard basis matrices via $E_\ell^{\lambda\mu}E_\ell^{\rho\sigma} = \delta_{\mu\rho}E_\ell^{\lambda\sigma}$ (for $\ell = 1, 2$) we obtain

$$
g(u,v)\big(T_2(v)T_1(u)P_{12} - P_{12}T_1(u)T_2(v)\big)
$$
$$
= g(u,v)\sum_{i,j,k,l,a,b=1}^{N} \big(T_{kl}(v)T_{ij}(u)E_1^{ib}E_2^{ka}\delta_{ja}\delta_{lb} - T_{ij}(u)T_{kl}(v)E_1^{aj}E_2^{bl}\delta_{bi}\delta_{ka}\big)
$$
$$
= g(u,v)\sum_{i,j,k,l=1}^{N} \big(T_{kl}(v)T_{ij}(u)E_1^{il}E_2^{kj} - T_{ij}(u)T_{kl}(v)E_1^{kj}E_2^{il}\big).
$$
(1.35)

Replacing the subscripts $l \leftrightarrow j$ in the first term and the subscripts $k \leftrightarrow i$ in the second term we arrive at

$$g(u,v)\big(T_2(v)T_1(u)P_{12} - P_{12}T_1(u)T_2(v)\big)$$

$$= g(u,v) \sum_{i,j,k,l=1}^{N} \big(T_{kj}(v)T_{il}(u) - T_{kj}(u)T_{il}(v)\big)E_1^{ij}E_2^{kl}. \quad (1.36)$$

Comparing the coefficients of $E_1^{ij}E_2^{kl}$ in (1.33) and (1.36) we immediately obtain (1.29). $\quad\square$

*Proof of proposition 1.1.* Consider commutation relations of the operators $\tilde{T}_{ij}$. We have

$$[\tilde{T}_{ij}(u), \tilde{T}_{kl}(v)] = [T_{\tilde{j},\tilde{i}}(-u), T_{\tilde{l},\tilde{k}}(-v)] = g(-u,-v)\big(T_{\tilde{l},\tilde{i}}(-v)T_{\tilde{j},\tilde{k}}(-u) - T_{\tilde{l},\tilde{i}}(-u)T_{\tilde{j},\tilde{k}}(-v)\big)$$

$$= g(v,u)\big(\tilde{T}_{il}(v)\tilde{T}_{kj}(u) - \tilde{T}_{il}(u)\tilde{T}_{kj}(v)\big) = g(u,v)\big(\tilde{T}_{il}(u)\tilde{T}_{kj}(v) - \tilde{T}_{il}(v)\tilde{T}_{kj}(u)\big). \quad (1.37)$$

Thus, the matrix elements $\tilde{T}_{ij}(u)$ satisfy the same commutation relations as $T_{ij}(u)$. $\quad\square$

## 1.7 Coloring

In physical models, the vectors of the space $\mathcal{H}$ describe states with different types of particles (excitations). We now introduce a notion of coloring, in which particles of different types also appear. To distinguish them from physical particles, we will call them quasiparticles, and their different types are colors.

The space $\mathcal{H}$ is generated by the states of the form

$$|\Psi\rangle = \prod_{p=1}^{n} T_{i_p,j_p}(u_p)|0\rangle, \quad (1.38)$$

where $i_p < j_p$ for $p = 1,\ldots,n$. This means that $T_{i_p,j_p}(u_p)$ are creation operators. We say that an operator $T_{ij}$ with $i < j$ creates quasiparticles with the colors $i,\ldots,j-1$, one quasiparticle of each color. In particular, the operator $T_{i,i+1}$ creates one quasiparticle of the color $i$, the operator $T_{1N}$ creates $N-1$ quasiparticles of $N-1$ different colors.

Thus, in $\mathfrak{gl}_N$ based models quasiparticles may have $N-1$ colors. Let $\{a_1,\ldots,a_{N-1}\}$ be a set of non-negative integers. We say that a state has coloring $\{a_k\} \equiv \{a_1,\ldots,a_{N-1}\}$, if it contains $a_k$ quasiparticles of the color $k$. In other words, we introduce a mapping $\mathrm{Col}(|\Psi\rangle)$ that maps $|\Psi\rangle$ to its coloring $\{a_k\}$:

$$\mathrm{Col}(|\Psi\rangle) = \{a_k\}, \qquad \text{where} \qquad a_k = \sum_{p=1}^{n}\big(\theta(j_p - k) - \theta(i_p - k)\big). \quad (1.39)$$

Here $\theta(k)$ is a step function of integer argument such that $\theta(k) = 1$ for $k > 0$ and $\theta(k) = 0$ otherwise. Let us give several examples. Consider $\mathfrak{gl}_4$ based models. Then we deal with three colors. We have

$$\begin{aligned}
\mathrm{Col}(|0\rangle) &= \{0,0,0\}, \qquad \text{(by definition)}, \\
\mathrm{Col}(T_{23}(u)|0\rangle) &= \{0,1,0\}, \\
\mathrm{Col}(T_{13}(u_1)T_{14}(u_2)|0\rangle) &= \{2,2,1\}, \\
\mathrm{Col}(T_{12}(u_1)T_{23}(v_1)T_{12}(u_2)T_{24}(v_2)T_{14}(w)|0\rangle) &= \{3,3,2\}.
\end{aligned} \quad (1.40)$$

Observe that the coloring does not depend on the arguments of the operators $T_{ij}(u)$ and on the order of these operators.

Assuming that the null-vector[2] has arbitrary coloring we extend the mapping (1.39) to some linear combinations of the states (1.38) as well as to the states containing neutral operators (i.e. $T_{ii}(u)$) and annihilation operators (i.e. $T_{ij}(u)$ with $i > j$). Namely, if $\mathrm{Col}(|\Psi_1\rangle) = \mathrm{Col}(|\Psi_2\rangle)$, then linear combinations of these states have the same coloring:

$$\mathrm{Col}(\alpha|\Psi_1\rangle + \beta|\Psi_2\rangle) = \mathrm{Col}(|\Psi_1\rangle), \qquad \text{if} \qquad \mathrm{Col}(|\Psi_1\rangle) = \mathrm{Col}(|\Psi_2\rangle), \qquad (1.41)$$

where $\alpha$ and $\beta$ are complex numbers. Then the states with the same coloring generate a subspace $\mathcal{H}_{\{a_k\}}$ of the space $\mathcal{H}$. The latter then can be presented as a direct sum of the subspaces with fixed coloring:

$$\mathcal{H} = \oplus \mathcal{H}_{\{a_k\}}. \qquad (1.42)$$

Let us consider the states of the form (1.38), but now suppose that among $T_{i_p,j_p}(u_p)$ there can be neutral operators and annihilation operators. The coloring of these states is defined by the same formula (1.39). Then the integers $a_k$ may take negative values.

**Proposition 1.2.** *Let* $\mathrm{Col}(|\Psi\rangle) = \{a_k\}$*, where at least one* $a_j < 0$*. Then* $|\Psi\rangle = 0$*.*

*Proof.* Suppose that $|\Psi\rangle \neq 0$. Then we can normal order all the operators $T_{ij}$, that is, we can move all neutral and annihilation operators to the extreme right position using commutation relations (1.29). Observe that the coloring mapping is compatible with these commutation relations. Thus, at any step of the normal ordering we deal with the state of the initial coloring. After the normal ordering is completed, the state $|\Psi\rangle$ depends on creation operators only. Then due to (1.39) $a_k \geq 0$ for all $k = 1, \dots, N-1$. We arrive at the contradiction, hence, $|\Psi\rangle = 0$. $\square$

Proposition 1.2 allows us in some cases to quickly calculate the action of annihilation operators on the states without use of commutation relations (1.29). For example, we can immediately say that

$$T_{41}(z)T_{13}(u_1)T_{13}(u_2)T_{12}(v_1)T_{13}(u_3)T_{12}(v_2)|0\rangle = 0, \qquad (1.43)$$

because

$$\mathrm{Col}\big(T_{41}(z)T_{13}(u_1)T_{13}(u_2)T_{12}(v_1)T_{13}(u_3)T_{12}(v_2)\big) = \{4, 2, -1, \dots\}. \qquad (1.44)$$

The reader can convince himself that the use of commutation relations (1.29) gives the same result, however it takes much more time and efforts. Generically, if $j - 1 < k < i$, and the annihilation operator $T_{ij}$ acts on a state in which there is no quasiparticles of the color $k$, then this action vanishes, like in (1.43).

**Proposition 1.3.** *Let a state* $|\Psi\rangle$ *do not contain quasiparticles of the color* 1*. Then*

$$T_{11}(z)|\Psi\rangle = \lambda_1(z)|\Psi\rangle. \qquad (1.45)$$

*Proof.* Obviously, it is enough to consider monomials (1.38) consisting of creation operators only. Otherwise, we always can get rid of the neutral and annihilation operators by normal ordering them. Since this monomial does not contain quasiparticles of the color 1, we conclude that $i_p > 1$ and $j_p > 1$. Then, due to commutation relations (1.29) we have

$$T_{11}(z)T_{i_p,j_p}(u_p) = T_{i_p,j_p}(u_p)T_{11}(z) + g(z,u_p)\big(T_{1,j_p}(z)T_{i_p,1}(u_p) - T_{1,j_p}(u_p)T_{i_p,1}(z)\big). \qquad (1.46)$$

---

[2]Do not confuse the null-vector 0 with the vacuum vector $|0\rangle$.

There are two terms here. In the first term, the operators $T_{11}(z)$ and $T_{i_p,j_p}(u_p)$ are simply rearranged. In the second term, we have the annihilation operator $T_{i_p,1}$ on the right. The action of the latter on a state without quasiparticles of the first color gives zero. Thus, the operator $T_{11}(z)$ goes through all the creation operators $T_{i_p,j_p}(u_p)$ to the extreme right position, where it acts on the vacuum vector and gives $\lambda_1(z)$. $\qquad\square$

Applying the automorphism (1.28) to (1.45) we find that in the $\mathcal{R}_N$ algebra

$$T_{NN}(z)|\Psi\rangle = \lambda_N(z)|\Psi\rangle, \tag{1.47}$$

provided the state $|\Psi\rangle$ does not contain quasiparticles of the last color $N-1$. This property also can be checked via direct calculation similar to (1.46). However, an analogous property is not valid for the quasiparticles of the intermediate colors $2,\dots,N-2$. For example, if $|\Psi\rangle = T_{12}(u_1)T_{34}(u_2)|0\rangle$, then this state does not contain quasiparticles of the color 2. At the same time,

$$\begin{aligned}
T_{22}(z)|\Psi\rangle = T_{22}(z)\, T_{12}(u_1)T_{34}(u_2)|0\rangle &= T_{12}(u_1)T_{22}(z)T_{34}(u_2)|0\rangle \\
&+ g(z,u_1)\big(T_{12}(u_1)T_{22}(z) - T_{12}(z)T_{22}(u_1)\big)T_{34}(u_2)|0\rangle \\
&= \lambda_2(z)\big(1 + g(z,u_1)\big)|\Psi\rangle + g(u_1,z)\lambda_2(u_1)T_{12}(z)T_{34}(u_2)|0\rangle. \tag{1.48}
\end{aligned}$$

In conclusion, we note that the coloring mapping can also be introduced for models with the $q$-deformed $R$-matrix (1.14).

# 2 Bethe vectors

We have already introduced in section 1.1 notions of Bethe vector and on-shell Bethe vector in $\mathfrak{gl}_2$ based models. Recall that on-shell Bethe vectors are eigenvectors of the transfer matrix. To solve the spectral problem, they are only needed. However, in computing the correlation functions, we also have to deal with off-shell Bethe vectors. Indeed, a typical problem arising in calculating correlation functions is to compute a matrix element of an operator $\hat{\mathcal{O}}$ of the following form:

$$\mathcal{O}_{\Psi'\Psi} = \langle\Psi'|\hat{\mathcal{O}}|\Psi\rangle. \tag{2.1}$$

Here $|\Psi\rangle$ is an on-shell Bethe vector, and $\langle\Psi'|$ is an on-shell Bethe vector in the dual space (dual on-shell Bethe vector). If the operator $\hat{\mathcal{O}}$ does not commute with the transfer matrix, then $\hat{\mathcal{O}}|\Psi\rangle = |\Phi\rangle$, where $|\Phi\rangle$ is no longer on-shell Bethe vector.

In many cases, it is possible to express this vector as a linear combination of off-shell Bethe vectors. In particular, if an explicit solution of the quantum inverse problem is known for the model under consideration, then we can express local operators in terms of the elements of the monodromy matrix [39–41]. Let us give an example. The solution of the quantum inverse problem in the models with the monodromy matrix (1.21) has the form

$$E_k^{ij} = \left(\prod_{\ell=1}^{k-1} \operatorname{tr} T(\xi_\ell)\right) T_{ji}(\xi_k) \left(\prod_{\ell=1}^{k} \operatorname{tr} T(\xi_\ell)\right)^{-1}. \tag{2.2}$$

Here $E_k^{ij}$ is the standard basis matrix acting in the local space $V_k$. Then the matrix element (2.1) of this local operator reduces to

$$\langle\Psi'|E_k^{ij}|\Psi\rangle = \frac{\prod_{\ell=1}^{k-1}\Lambda'(\xi_\ell)}{\prod_{\ell=1}^{k}\Lambda(\xi_\ell)} \langle\Psi'|T_{ji}(\xi_k)|\Psi\rangle. \tag{2.3}$$

Here $\Lambda'(\xi_\ell)$ and $\Lambda(\xi_\ell)$ respectively are the transfer matrix eigenvalues on the on-shell vectors $\langle\Psi'|$ and $|\Psi\rangle$. Thus, we have to calculate the action of the matrix element $T_{ji}(\xi_k)$ on the vector $|\Psi\rangle$ and then calculate the resulting scalar product. In $\mathfrak{gl}_2$ based models, such action obviously gives a linear combination of off-shell Bethe vectors. For instance, if

$$|\Psi\rangle = \prod_{\ell=1}^{n} B(u_\ell)|0\rangle, \tag{2.4}$$

then

$$T_{11}(\xi_k)|\Psi\rangle = a(\xi_k)\prod_{i=1}^{n} f(u_i,\xi_k)|\Psi\rangle + \sum_{j=1}^{n} a(u_j)g(u_j,\xi_k)\left(\prod_{\substack{i=1\\i\neq j}}^{n} f(u_i,u_j)\right)|\Phi_j\rangle, \tag{2.5}$$

where $f(u,v) = 1 + g(u,v)$, and

$$|\Phi_j\rangle = B(\xi_k)\prod_{\substack{\ell=1\\\ell\neq j}}^{n} B(u_\ell)|0\rangle, \tag{2.6}$$

(see [1, 26, 27, 29]). The set of variables $u_1,\ldots,u_n$ satisfies Bethe equations, because the original vector $|\Psi\rangle$ is on-shell. However, the new sets $\{\xi_k,u_1,\ldots,u_n\}\setminus u_j$, $j=1,\ldots,n$, are no longer solutions to these equations[3]. Therefore, we have a linear combination of off-shell Bethe vectors $|\Phi_j\rangle$ in (2.5).

Similarly, in models with higher rank symmetries, the actions of the monodromy matrix elements on Bethe vectors generate linear combinations of off-shell vectors [42–44]. As a result, matrix elements of local operators are reduced to scalar products of Bethe vectors, in which one of the vectors is on-shell, while another one, generally speaking, is off-shell. Compact determinant formulas for such scalar products are known in the case of the $\mathcal{R}_2$ algebra and its $q$-deformation [29, 39, 45]. Partially similar results were recently obtained in the case of the $\mathcal{R}_3$ algebra [46–50]. These determinant representations allow us to study correlation functions analytically and numerically [51–58].

Thus, despite the fact that the off-shell Bethe vectors themselves have no physical meaning, they play a very important role in calculating the correlation functions. It is for this reason that we pay so much attention to these vectors below.

## 2.1 Bethe vectors in $\mathfrak{gl}_2$ based models

In the $\mathfrak{gl}_2$ based models the on-shell Bethe vectors have the form (1.8), provided the parameters $\bar{u} = \{u_1,\ldots,u_n\}$ satisfy a system of Bethe equations [1, 2, 26–29]

$$\frac{a(u_j)}{d(u_j)} = \prod_{\substack{k=1\\k\neq j}}^{n} \frac{f(u_j,u_k)}{f(u_k,u_j)}, \qquad j=1,\ldots,n, \tag{2.7}$$

and we recall that

$$f(u,v) = 1 + g(u,v) = \frac{u-v+c}{u-v}. \tag{2.8}$$

---

[3]Recall that the inhomogeneities $\xi_k$ are arbitrary complex numbers.

Complex variables $\bar{u} = \{u_1, \ldots, u_n\}$ are called *Bethe parameters*. Generic Bethe vectors (off-shell Bethe vectors) also have the form (1.8), however, the Bethe parameters are arbitrary complex numbers.

Equation (1.8) can be taken as the definition of off-shell Bethe vectors. However, this is not the only possible definition. The point is that the main property of the off-shell Bethe vectors is that they become on-shell, if the system of Bethe equations is fulfilled. Then the original definition can be modified in various ways. Namely, we can add to the vector $B(u_1) \ldots B(u_n)|0\rangle$ any other vector that vanishes on the system of Bethe equations. For example, let

$$|\Psi\rangle = B(u_1) \ldots B(u_n)|0\rangle + \left( \prod_{j=1}^{n} a(u_j) - \prod_{j=1}^{n} d(u_j) \right) |\Phi\rangle, \qquad (2.9)$$

where $|\Phi\rangle$ is an arbitrary vector. It is easy to see that if the set $\bar{u}$ enjoys the Bethe equations (2.7), then

$$\prod_{j=1}^{n} a(u_j) = \prod_{j=1}^{n} d(u_j). \qquad (2.10)$$

Hence, the coefficient of the vector $|\Phi\rangle$ vanishes, and the vector (2.9) becomes on-shell.

Thus, the combination (2.9) can also be called the Bethe vector, because it turns into the on-shell Bethe vector as soon as the Bethe parameters satisfy Bethe equations. It is clear that we can invent plenty of combinations of this type. Therefore, strictly speaking, the definition of the off-shell Bethe vector is ambiguous.

Among possible definitions, equation (1.8) looks the most simple. However, there exist other presentations for the Bethe vectors, which also have rather simple form. For instance, let

$$\tilde{T} = KTK^{-1} = \begin{pmatrix} \tilde{A}(u) & \tilde{B}(u) \\ \tilde{C}(u) & \tilde{D}(u) \end{pmatrix}, \qquad (2.11)$$

where $K$ is a $2 \times 2$ $c$-number invertible matrix such that $K_{11} \neq 0$. It is clear that the new operator $\tilde{B}(u)$ is a linear combination of the original $A$, $B$, $C$, and $D$. Nevertheless, a state

$$|\widetilde{\Psi}\rangle = \tilde{B}(u_1) \ldots \tilde{B}(u_n)|0\rangle \qquad (2.12)$$

is the on-shell Bethe vector provided the system (2.7) is fulfilled [59, 60]. We suggest the reader to check this statement. Anyway, presentation (2.12) looks as simple as the original formula (1.8).

Thus, we have a big freedom in the definition of the Bethe vectors. Nevertheless, following the tradition we define them by equation (1.8), in particular, for the reasons of simplicity.

## 2.2 Bethe vectors in $\mathfrak{gl}_3$ based models

The problem of Bethe vectors in the $\mathfrak{gl}_3$ (and higher rank) based models is much more sophisticated than in the case considered above. The ambiguity of their definition still exists, however, now the form of the on-shell Bethe vectors is much more complex than (1.8). Therefore, we cannot use even the reasons of simplicity for choosing an appropriate definition. The matter is that there is only one creation operator in the $\mathfrak{gl}_2$ case ($T_{12}$), while there are three creation operators in the $\mathfrak{gl}_3$ case ($T_{12}$, $T_{13}$, $T_{23}$).

Consider a simple example that will allow us to feel the difference between the Bethe vectors in the $\mathfrak{gl}_2$ and $\mathfrak{gl}_3$ based models. For this we will try to construct simple on-shell Bethe vectors in these two cases.

Consider the commutation relations (1.29). We see that generically we have different operators in the l.h.s. ($T_{ij}$ and $T_{kl}$) and in the r.h.s. ($T_{il}$ and $T_{kj}$). However, if the operators in the l.h.s. belong to the same row (column), then we obtain the same operators in the r.h.s. In particular, we have

$$T_{11}(u)T_{12}(v) = f(v,u)T_{12}(v)T_{11}(u) + g(u,v)T_{12}(u)T_{11}(v),$$
$$T_{22}(u)T_{12}(v) = f(u,v)T_{12}(v)T_{22}(u) + g(v,u)T_{12}(u)T_{22}(v). \tag{2.13}$$

Let us try to find an on-shell Bethe vector in a model described by the $\mathcal{R}_2$ algebra. This means that we are looking for the eigenvectors of the transfer matrix $T_{11}(z) + T_{22}(z)$. Let us test a vector $T_{12}(u)|0\rangle$. Then due to (2.13) we obtain

$$\left(T_{11}(z) + T_{22}(z)\right)T_{12}(u)|0\rangle = T_{12}(u)\left(f(u,z)T_{11}(z) + f(z,u)T_{22}(z)\right)|0\rangle$$
$$+ g(z,u)T_{12}(z)\left(T_{11}(u) - T_{22}(u)\right)|0\rangle. \tag{2.14}$$

We see that we still deal with the vectors of the type $T_{12}(\cdot)|0\rangle$. Indeed, using $T_{ii}(u)|0\rangle = \lambda_i(u)|0\rangle$ (where $\lambda_1(u) = a(u)$ and $\lambda_2(u) = d(u)$ (1.6)) we obtain

$$\left(T_{11}(z) + T_{22}(z)\right)T_{12}(u)|0\rangle = \left(f(u,z)a(z) + f(z,u)d(z)\right)T_{12}(u)|0\rangle$$
$$+ g(z,u)\left(a(u) - d(u)\right)T_{12}(z)|0\rangle. \tag{2.15}$$

Thus, the result of the action of the transfer matrix $T_{11}(z) + T_{22}(z)$ gives two vectors: $T_{12}(u)|0\rangle$ and $T_{12}(z)|0\rangle$. The first one is the same as in the l.h.s., while the second is different. Traditionally this second term is called *unwanted term*. We will call it *unwanted term of the first type*. It is still given as the action of $T_{12}$ on the vacuum (as in the l.h.s.), but the operator $T_{12}$ has new argument. This unwanted term can be killed, if we choose an appropriate $u = u_0$, namely, such that $a(u_0) = d(u_0)$. Observe that this condition coincides with Bethe equations (2.7) at $n = 1$. Then the vector $T_{12}(u_0)|0\rangle$ becomes the eigenvector of the transfer matrix.

It is easy to see that generically, if we test a vector of the form $T_{12}(u_1)\dots T_{12}(u_n)|0\rangle$, then the action of the transfer matrix produces unwanted terms of the first type only: we still obtain the products of the operators $T_{12}$ applied to $|0\rangle$, but some of these operators may have new arguments.

Now let us consider the $\mathcal{R}_3$ algebra. Let us test the vector $T_{13}(u)|0\rangle$. We should act with the transfer matrix onto this vector

$$\left(T_{11}(z) + T_{22}(z) + T_{33}(z)\right)T_{13}(u)|0\rangle. \tag{2.16}$$

We see immediately a principle difference with the case considered above. Namely, the operators $T_{22}$ and $T_{13}$ do not belong to the same row or column. Due to the commutation relations we have

$$T_{22}(z)T_{13}(u)|0\rangle = \left(T_{13}(u)T_{22}(z) + g(z,u)\left(T_{12}(u)T_{23}(z) - T_{12}(z)T_{23}(u)\right)\right)|0\rangle$$
$$= \lambda_2(z)T_{13}(u)|0\rangle + g(z,u)\left(T_{12}(u)T_{23}(z) - T_{12}(z)T_{23}(u)\right)|0\rangle, \tag{2.17}$$

and the action of the transfer matrix is

$$\text{tr}\, T(z)T_{13}(u)|0\rangle = \left(f(u,z)\lambda_1(z) + \lambda_2(z) + f(z,u)\lambda_3(z)\right)T_{13}(u)|0\rangle$$
$$+ g(z,u)\left(\lambda_1(u) - \lambda_3(u)\right)T_{13}(z)|0\rangle$$
$$+ g(z,u)\left(T_{12}(u)T_{23}(z) - T_{12}(z)T_{23}(u)\right)|0\rangle. \tag{2.18}$$

We obtain the vectors of a new type $T_{12}(u)T_{23}(z)|0\rangle$ and $T_{12}(z)T_{23}(u)|0\rangle$, that we call *unwanted terms of the second type*. These unwanted terms generically cannot be killed by an appropriate choice of the original argument $u$.

*Remark 1.* Strictly speaking, the term in the third line of (2.18) vanishes at $u \to \infty$. In some models (for example, $SU(3)$-invariant $XXX$ chain) Bethe equations have infinite roots, and then the corresponding contribution in (2.18) vanishes.

*Remark 2.* In some models the operator $T_{23}(u)$ actually plays the role of the annihilation operator: $T_{23}(u)|0\rangle = 0$ for all $u$. A typical example of such a monodromy matrix is the matrix (1.21). We will consider this example in more detail in section 3.2. Then, for this type of models, the unwanted terms of the second type in (2.18) automatically vanish. The vector $T_{13}(u)|0\rangle$ thus becomes the on-shell Bethe vector for $u = u_0$ such that $\lambda_1(u_0) = \lambda_3(u_0)$. We emphasize, however, that this is true only for the special case of models with the $\mathcal{R}_3$ algebra. Generically $T_{23}(u)|0\rangle \neq 0$, therefore, we obtain the unwanted terms of the second type in (2.18).

Summarizing the above considerations we conclude that we deal with unwanted terms of two types:

- first type: the operators are the same as in the l.h.s., but some of them accept new arguments;

- second type: the operators in the r.h.s. are different form the ones in the l.h.s. In this case, they can either keep their original arguments or accept new arguments.

In the case of the $\mathcal{R}_2$ algebra we obtain the first type of unwanted terms only. For the $\mathcal{R}_3$ algebra (and higher) we necessarily obtain both types of unwanted terms. Therefore the structure of the Bethe vectors in the $\mathfrak{gl}_3$ case generically cannot be so simple as in the $\mathfrak{gl}_2$ case. The above example shows that not every combination of the creation operators applied to the vacuum has a chance to become an eigenvector of the transfer matrix even for some specific values of the Bethe parameters. In particular, in the general case, the vector $T_{13}(u)|0\rangle$ cannot be an on-shell Bethe vector for any values of $u$. We will see below, that in order to obtain a Bethe vector, one should take the following combination of the terms $T_{13}(u)|0\rangle$ and $T_{12}(u)T_{23}(v)|0\rangle$:

$$g(v,u)\lambda_2(v)T_{13}(u)|0\rangle + T_{12}(u)T_{23}(v)|0\rangle. \tag{2.19}$$

If the parameters $u$ and $v$ enjoy the system of equations

$$\frac{\lambda_1(u)}{\lambda_2(u)} = \frac{\lambda_3(v)}{\lambda_2(v)} = f(v,u), \tag{2.20}$$

then the state (2.19) is the on-shell Bethe vector. Thus, the state (2.19) can be called the off-shell Bethe vector, if $u$ and $v$ are generic complex numbers. We see, however, that even in this simple example, the form of the off-shell Bethe vector is highly non-trivial. This is a very special polynomial in the creation operators acting on the vacuum vector. Of course, the modification of the vector (2.19) in the same spirit as in (2.9) remains possible, and hence, representation (2.19) for this off-shell Bethe vector is not unique. It is not clear, however, which among these representations is the simplest.

There are several ways to construct on-shell Bethe vectors in the models with the $\mathfrak{gl}_N$-invariant $R$-matrix. In addition to the NABA, it is also worth mentioning the approach associated with the so called *trace formula* [10–12], as well as the method based on the use of a

special current algebra to describe the $RTT$-relation [61–64]. It is remarkable that all the three methods listed above give eventually the same expression, not only for on-shell, but also for off-shell Bethe vectors. And this is despite the initial ambiguity in defining the off-shell Bethe vectors. Therefore, we will adopt this formula as the definition of the Bethe vector. Details will be described later.

*Remark.* Recently a new presentation for on-shell Bethe vectors in terms of Sklyanin's $B$-operators [65] was conjectured in [59] for the models with $\mathfrak{gl}_N$-invariant $R$-matrix. The proof of this presentation was given in [66] in the framework of the quantum separation of variables developed in [67]. Within this approach, the on-shell Bethe vectors have the form (1.8) provided the Bethe parameters satisfy Bethe equations. However, it was shown in [68] that if the Bethe parameters remain arbitrary complex numbers, then at least in the case of the $\mathcal{R}_3$ algebra, the corresponding vectors coincide with the special case of off-shell Bethe vectors constructed within the NABA framework.

## 2.3 Notation

We now fix some notation and conventions that will be used throughout all the notes.

- *Rational functions.* We have introduced already two rational functions $g(x, y)$ and $f(x, y)$. Recall that

$$g(x, y) = \frac{c}{x - y}, \qquad f(x, y) = 1 + g(x, y) = \frac{x - y + c}{x - y}. \qquad (2.21)$$

Observe that $g(x, y) = -g(y, x)$. Below we will permanently use these functions.

- *Sets of variables.* We denote sets of variables by a bar: $\bar{x}$, $\bar{u}$, $\bar{v}$ etc. Individual elements of the sets are denoted by the subscripts: $v_j$, $u_k$ etc. A notation $\bar{u}_i$, means $\bar{u} \setminus u_i$ etc. Instead of the standard notation $\bar{u} \cup \bar{v}$ we use braces $\{\bar{u}, \bar{v}\}$ for the union of sets.

- *Shorthand notation for products.*

  In order to make formulas more compact we use a shorthand notation for the products of commuting operators or functions depending on one or two variables. Namely, if the functions $\lambda_i$, $g$, $f$, as well as the operators $T_{ij}$ depend on sets of variables, this means that one should take the product over the corresponding set. For example,

$$T_{ij}(\bar{u}) = \prod_{u_k \in \bar{u}} T_{ij}(u_k); \quad g(z, \bar{w}_i) = \prod_{\substack{w_j \in \bar{w} \\ w_j \neq w_i}} g(z, w_j); \quad f(\bar{u}, \bar{v}) = \prod_{u_j \in \bar{u}} \prod_{v_k \in \bar{v}} f(u_j, v_k). \quad (2.22)$$

  Observe that $[T_{ij}(u), T_{ij}(v)] = 0$ due to the commutation relations (1.29). Therefore, the product $T_{ij}(\bar{u})$ is well defined. By definition, any product over the empty set is equal to 1. A double product is equal to 1 if at least one of the sets is empty.

  The use of this shorthand notation is not a whim, but a necessity. Even within the framework of the usual ABA, we often have to deal with rather cumbersome formulas. In the NABA, this bulkiness drastically increases. The shorthand notation for products allows us to reduce the size of formulas to some extent. Therefore, we will constantly use them despite some disadvantages (for example, the lack of information about the cardinalities of the sets in which the product is taken).

We will extend this convention for new functions that will appear later on. For the moment, let us show how this convention works in particular examples. Equation (1.8) takes the following form

$$|\Psi\rangle = B(\bar{u})|0\rangle. \tag{2.23}$$

If necessary, we should add a special comment on the cardinality of the set $\bar{u}$. The system of Bethe equations (2.7) reads

$$\frac{a(u_j)}{d(u_j)} = \frac{f(u_j, \bar{u}_j)}{f(\bar{u}_j, u_j)}, \qquad j = 1, \ldots, n. \tag{2.24}$$

# 3 Nested algebraic Bethe ansatz

In this section, we directly proceed to the description of the NABA method. This method allows us to obtain a representation for on-shell and off-shell Bethe vectors in the models with the $\mathfrak{gl}_3$-invariant $R$-matrix. In addition, we obtain a system of Bethe equations, whose solutions determine the spectrum of the transfer matrix. The presentation follows the works [3–6].

## 3.1 Basic notions

Consider a model with the $\mathfrak{gl}_3$-invariant $R$-matrix (1.12)

$$R(u, v) = \mathbb{I} + g(u, v)P, \tag{3.1}$$

where the identity matrix $\mathbb{I}$ and the permutation matrix $P$ are $9 \times 9$ matrices:

$$\mathbb{I}_{ij}^{kl} = \delta_{ij}\delta_{kl}, \qquad P_{ij}^{kl} = \delta_{il}\delta_{jk}, \qquad i, j, k, l = 1, 2, 3. \tag{3.2}$$

Recall that the $R$-matrix acts in the tensor product $\mathbb{C}^3 \otimes \mathbb{C}^3$. The lower indices $i$ and $j$ in (3.2) refer to the first space, the upper indices $k$ and $l$ in refer to the second space.

We will need also the $\mathfrak{gl}_2$-invariant $R$-matrix, which we denote by $r(u, v)$:

$$r(u, v) = \mathbf{1} + g(u, v)\mathbf{p}. \tag{3.3}$$

Here the identity matrix $\mathbf{1}$ and the permutation matrix $\mathbf{p}$ are $4 \times 4$ matrices acting in $\mathbb{C}^2 \otimes \mathbb{C}^2$:

$$\mathbf{1}_{\alpha\beta}^{\rho\mu} = \delta_{\alpha\beta}\delta_{\rho\mu}, \qquad \mathbf{p}_{\alpha\beta}^{\rho\mu} = \delta_{\alpha\mu}\delta_{\beta\rho}, \qquad \alpha, \beta, \rho, \mu = 1, 2. \tag{3.4}$$

The indices of the matrices are arranged similarly to equation (3.2).

A monodromy matrix is

$$T(u) = \begin{pmatrix} T_{11}(u) & T_{12}(u) & T_{13}(u) \\ T_{21}(u) & T_{22}(u) & T_{23}(u) \\ T_{31}(u) & T_{32}(u) & T_{33}(u) \end{pmatrix}. \tag{3.5}$$

It satisfies $RTT$-relation (1.1). This relation implies the set of commutation relations (1.29) for the operators $T_{ij}$.

We will also use one more parametrization of the monodromy matrix

$$T(u) = \begin{pmatrix} A(u) & \mathbb{B}(u) \\ \mathbb{C}(u) & \mathbb{D}(u) \end{pmatrix} = \begin{pmatrix} A(u) & B_1(u) & B_2(u) \\ C_1(u) & D_{11}(u) & D_{12}(u) \\ C_2(u) & D_{21}(u) & D_{22}(u) \end{pmatrix}. \tag{3.6}$$

Here in the intermediate formula the $T$-matrix is presented as a $2 \times 2$ block-matrix. The block $A$ has the size $1 \times 1$, the block $\mathbb{B}$ has the size $1 \times 2$, the block $\mathbb{C}$ has the size $2 \times 1$, and the block $\mathbb{D}$ has the size $2 \times 2$.

*Remark.* One can also consider another embedding

$$T(u) = \begin{pmatrix} \mathbb{A}'(u) & \mathbb{B}'(u) \\ \mathbb{C}'(u) & D'(u) \end{pmatrix} = \begin{pmatrix} A'_{11}(u) & A'_{12}(u) & B'_1(u) \\ A'_{21}(u) & A'_{22}(u) & B'_2(u) \\ C'_1(u) & C'_2(u) & D'(u) \end{pmatrix}, \tag{3.7}$$

where now the block $\mathbb{A}'$ is a $2 \times 2$ matrix, while the block $D'$ has the size $1 \times 1$. These two embeddings are equivalent due to the automorphism (1.28). For definiteness, we consider in details parametrization (3.6) and give short comments about parametrization (3.7).

## 3.2 Particular case of $\mathfrak{gl}_3$ invariant models

In this section we consider a particular case of the Bethe vectors. It will give us an idea of their construction in the general case.

Generically, the operators $T_{ij}$ with $i < j$ are the creation operators. However, as we already mentioned, in some models the operator $T_{23}$ annihilates the vacuum vector[4]: $T_{23}(u)|0\rangle = 0$. Actually, we very often deal with this situation in the models of physical application ($SU(3)$-invariant Heisenberg chain, two-component Bose gas, t-J model). Therefore this particular case is rather important.

Consider the simplest example of such monodromy matrix: $T(u) = R(u, \xi)$, where $\xi$ is a fixed complex number. This is the monodromy matrix of the $XXX$ chain consisting of one site. Then $T_{23}(u) = g(u, \xi)E_{32}$. Obviously

$$T_{23}(u)|0\rangle = g(u, \xi)E_{32}|0\rangle = g(u, \xi)E_{32}\begin{pmatrix} 1 \\ 0 \\ 0 \end{pmatrix} = 0. \tag{3.8}$$

Consider now the chain with $L$ sites. Then the monodromy matrix is given by (1.21)

$$T^{(L)}(u) = T^{(L-1)}(u)T^{(1)}(u), \tag{3.9}$$

where

$$T^{(L-1)}(u) = R_{0L}(u, \xi_L)\dots R_{02}(u, \xi_2), \qquad T^{(1)}(u) = R_{01}(u, \xi_1). \tag{3.10}$$

Hence, the operator $T_{23}^{(L)}(u)$ of the whole chain has the following representation

$$T_{23}^{(L)}(u) = T_{21}^{(L-1)}(u)T_{13}^{(1)}(u) + T_{22}^{(L-1)}(u)T_{23}^{(1)}(u) + T_{23}^{(L-1)}(u)T_{33}^{(1)}(u), \tag{3.11}$$

where $T_{ij}^{(L-1)}(u)$ and $T_{ij}^{(1)}(u)$ are the entries of the monodromy matrices respectively corresponding to the sub-chains of the lengths $L-1$ and 1. Since $T_{ij}^{(L-1)}(u)$ and $T_{kl}^{(1)}(v)$ act in different spaces, they commute: $[T_{ij}^{(L-1)}(u), T_{kl}^{(1)}(v)] = 0$ for arbitrary subscripts and arbitrary arguments.

We know that $T_{23}^{(L-1)}(u)|0\rangle = 0$ for $L = 2$, since in this case we again deal with the monodromy matrix of the $XXX$ chain with one site. Assume that $T_{23}^{(L-1)}(u)|0\rangle = 0$ for some $L > 1$. Then it follows from (3.11) that $T_{23}^{(L)}(u)|0\rangle = 0$. Indeed, $T_{21}^{(L-1)}(u)|0\rangle = 0$ by definition, $T_{23}^{(1)}(u)|0\rangle = 0$ due to (3.8), and $T_{23}^{(L-1)}(u)|0\rangle = 0$ due to the induction assumption.

---

[4]Another possibility is that $T_{12}$ annihilates the vacuum. Due to the automorphism (1.28) the cases $T_{23}(u)|0\rangle = 0$ and $T_{12}(u)|0\rangle = 0$ are equivalent.

This method also allows us to find the vacuum eigenvalues $\lambda_i(u)$ of the diagonal entries $T_{ii}(u)$:

$$\lambda_1(u) = f(u, \bar{\xi}) = \prod_{k=1}^{L} f(u, \xi_k), \qquad \lambda_2(u) = \lambda_3(u) = 1. \tag{3.12}$$

Observe that we have $\lambda_2(u) = \lambda_3(u)$. This is the direct consequence of the property $T_{23}(u)|0\rangle = 0$. More precisely, if $T_{23}(u)|0\rangle = 0$, then $\lambda_2(u) = \kappa\lambda_3(u)$, where $\kappa$ is a constant. Indeed, it follows from (1.29) that

$$[T_{23}(u), T_{32}(v)] = g(u,v)\big(T_{22}(u)T_{33}(v) - T_{22}(v)T_{33}(u)\big). \tag{3.13}$$

Applying this equation to $|0\rangle$ we obtain

$$0 = g(u,v)\big(\lambda_2(u)\lambda_3(v) - \lambda_2(v)\lambda_3(u)\big)|0\rangle. \tag{3.14}$$

Hence, the ratio $\lambda_2(u)/\lambda_3(u)$ does not depend on $u$.

## 3.3 Action of the operators $D_{\xi\alpha}$

Consider a model in which $T_{23}(u)|0\rangle = 0$ without specifying the Hilbert space $\mathcal{H}$ in which the operators $T_{ij}(u)$ act. Let the monodromy matrix be normalized in such a way that $\lambda_3(u) = 1$. Then $\lambda_2(u)$ must be a constant. For simplicity we assume that $\lambda_2(u) = 1$, like in (3.12). More general case will be considered later.

In the case under consideration we have only two creation operators: $T_{12}(u) \equiv B_1(u)$ and $T_{13}(u) \equiv B_2(u)$ (see (3.6)). We can try to check a monomial

$$B_{\beta_1}(u_1)\ldots B_{\beta_a}(u_a)|0\rangle, \qquad a = 0, 1, \ldots, \tag{3.15}$$

as a candidate for the transfer matrix eigenvector. Here every $\beta_i$ is equal to either 1 or 2. We should act with the transfer matrix

$$\operatorname{tr} T(z) = A(z) + D_{11}(z) + D_{22}(z) \tag{3.16}$$

onto this vector. We start our consideration with the action of the operators $D_{\alpha\alpha}(z)$.

It follows from (1.29) that

$$D_{\xi\alpha}(z)B_\beta(u) = B_\beta(u)D_{\xi\alpha}(z) + g(z,u)B_\alpha(u)D_{\xi\beta}(z) + g(u,z)B_\alpha(z)D_{\xi\beta}(u). \tag{3.17}$$

We see that acting with $D_{\xi\alpha}(z)$ onto the vector (3.15) we may have unwanted terms of the second type. Indeed, the operator $D_{11}$ acting on $B_2$ gives contributions with $B_1$, and the operator $D_{22}$ acting on $B_1$ gives contributions with $B_2$. Thus, the operator structure in the vector (3.15) is not invariant under the action of $D_{11}(z) + D_{22}(z)$. At the same time the action of the operator $A(z)$ does not produce unwanted terms of the second type (see section 3.4).

Thus, the monomial (3.15) generically is not invariant under the action of $\operatorname{tr} T(z)$. Therefore, it is quite natural to replace the monomial (3.15) by a polynomial

$$|\Psi_a(\bar{u})\rangle = \sum_{\beta_1,\ldots,\beta_a} B_{\beta_1}(u_1)\ldots B_{\beta_a}(u_a)F_{\beta_1,\ldots,\beta_a}|0\rangle, \qquad a = 0, 1, \ldots. \tag{3.18}$$

Here $F_{\beta_1,\ldots,\beta_a}$ are some numerical coefficients. The sum is taken over every $\beta_i \in \{\beta_1, \ldots, \beta_a\}$. Each $\beta_i$ takes the values $\beta_i = 1, 2$.

Let us write down (3.18) in the form of a scalar product. For this we introduce a two-component vector-row $\mathbb{B}(u) = \big(B_1(u), B_2(u)\big)$ with operator-valued components. Consider a tensor product

$$\mathbb{B}_1(u_1)\mathbb{B}_2(u_2)\ldots\mathbb{B}_a(u_a) = \mathbb{B}(u_1) \otimes \mathbb{B}(u_2) \otimes \cdots \otimes \mathbb{B}(u_a). \tag{3.19}$$

This is a $2^a$-component vector-row. Then we can write down the vector $|\Psi_a(\bar{u})\rangle$ as the scalar product

$$|\Psi_a(\bar{u})\rangle = \mathbb{B}_1(u_1)\mathbb{B}_2(u_2)\ldots\mathbb{B}_a(u_a)\mathbb{F}(\bar{u})|0\rangle, \tag{3.20}$$

where $\mathbb{F}(\bar{u})$ is a vector belonging to the space

$$\underbrace{\mathbb{C}^2 \otimes \cdots \otimes \mathbb{C}^2}_{a \quad \text{times}}. \tag{3.21}$$

The commutation relations (3.17) can be written as follows

$$\mathbb{D}_0(z)\mathbb{B}_1(u) = \mathbb{B}_1(u)\mathbb{D}_0(z)r_{01}(z,u) + g(u,z)\mathbb{B}_1(z)\mathbb{D}_0(u)p_{01}, \tag{3.22}$$

where $p_{01}$ is $4 \times 4$ permutation matrix. Here the matrix $\mathbb{D}_0$ acts in the auxiliary space $V_0 \sim \mathbb{C}^2$ and the vector $\mathbb{B}$ belongs to the auxiliary space $V_1 \sim \mathbb{C}^2$. The $r$-matrix $r_{01}(z,u)$ and the matrix $p_{01}$ act in the tensor product $V_0 \otimes V_1$. We call the first term in the r.h.s. of (3.22) *the first scheme of commutation*. The second term in the r.h.s. of (3.22) is called *the second scheme of commutation*.

It follows from the commutation relations (1.29) that

$$[D_{\alpha\beta}(u), D_{\gamma\delta}(v)] = g(u,v)\big(D_{\gamma\beta}(v)D_{\alpha\delta}(u) - D_{\gamma\beta}(u)D_{\alpha\delta}(v)\big). \tag{3.23}$$

Hence, the matrix $\mathbb{D}(u)$ satisfies the $RTT$-relation with the $R$-matrix $r(u,v)$

$$r_{12}(u,v)\mathbb{D}_1(u)\mathbb{D}_2(v) = \mathbb{D}_2(v)\mathbb{D}_1(u)r_{12}(u,v). \tag{3.24}$$

Therefore, $\mathbb{D}(u)$ can be treated as the monodromy matrix of a model with the $\mathfrak{gl}_2$-invariant $R$-matrix.

Let us act with $\operatorname{tr}\mathbb{D}(z)$ onto $|\Psi_a(\bar{u})\rangle$. We have

$$\operatorname{tr}_0 \mathbb{D}_0(z)|\Psi_a(\bar{u})\rangle = \operatorname{tr}_0 \mathbb{D}_0(z)\mathbb{B}_1(u_1)\mathbb{B}_2(u_2)\ldots\mathbb{B}_a(u_a)\mathbb{F}(\bar{u})|0\rangle. \tag{3.25}$$

Here we have stressed that $\mathbb{D}(z)$ acts in $V_0$, which is different from $V_1, \ldots, V_a$. Permuting $\mathbb{D}_0(z)$ and $\mathbb{B}_1(u_1)$ we obtain

$$\operatorname{tr}_0 \mathbb{D}_0(z)|\Psi_a(\bar{u})\rangle = \operatorname{tr}_0\Big(\mathbb{B}_1(u_1)\mathbb{D}_0(z)r_{01}(z,u_1) + g(u_1,z)\mathbb{B}_1(z)\mathbb{D}_0(u_1)p_{01}\Big)$$
$$\times \mathbb{B}_2(u_2)\ldots\mathbb{B}_a(u_a)\mathbb{F}(\bar{u})|0\rangle. \tag{3.26}$$

We have two contributions. The second one definitely is unwanted, as it contains the operators $B_\beta(z)$ (in the vector-row $\mathbb{B}_1(z)$). Let us leave this term for some time and only deal with the wanted contributions. In other words, we use the first scheme of commutation only. In the case of the $\mathcal{R}_2$ algebra the use of the first scheme would necessarily give us wanted terms only. However, in the case of the $\mathcal{R}_3$ algebra we still may have unwanted terms of the second type. Our first goal is to get rid of them.

We have

$$\text{tr}_0\, \mathbb{D}_0(z)|\Psi_a(\bar{u})\rangle = \text{tr}_0\, \mathbb{B}_1(u_1)\mathbb{D}_0(z)r_{01}(z,u_1)\mathbb{B}_2(u_2)\ldots\mathbb{B}_a(u_a)\mathbb{F}(\bar{u})|0\rangle + \mathcal{Z}, \qquad (3.27)$$

where $\mathcal{Z}$ denotes unwanted contributions. Clearly, the $R$-matrix $r_{01}(z,u_1)$ can be moved to the right

$$\text{tr}_0\, \mathbb{D}_0(z)|\Psi_a(\bar{u})\rangle = \text{tr}_0\, \mathbb{B}_1(u_1)\mathbb{D}_0(z)\mathbb{B}_2(u_2)\ldots\mathbb{B}_a(u_a)r_{01}(z,u_1)\mathbb{F}(\bar{u})|0\rangle + \mathcal{Z}. \qquad (3.28)$$

Then we repeat the procedure and finally we arrive at

$$\text{tr}_0\, \mathbb{D}_0(z)|\Psi_a(\bar{u})\rangle = \text{tr}_0\, \mathbb{B}_1(u_1)\mathbb{B}_2(u_2)\ldots\mathbb{B}_a(u_a)\mathbb{D}_0(z)r_{0a}(z,u_a)\ldots r_{01}(z,u_1)\mathbb{F}(\bar{u})|0\rangle + \mathcal{Z}. \quad (3.29)$$

We have obtained a matrix $\widehat{\mathcal{T}}^{(a)}(z)$

$$\widehat{\mathcal{T}}_0^{(a)}(z) = \mathbb{D}_0(z)\mathcal{T}_0^{(a)}(z), \quad \text{where} \quad \mathcal{T}_0^{(a)}(z) = r_{0a}(z,u_a)\ldots r_{01}(z,u_1), \qquad (3.30)$$

and the subscript 0 stresses that the auxiliary space of this matrix is $V_0$. Recall that the matrix $\mathbb{D}_0(z)$ can be treated is the monodromy matrix satisfying the $\mathcal{R}_2$ algebra due to (3.24). Its matrix elements act in the original Hilbert space $\mathcal{H}$ as follows:

$$\begin{aligned} D_{11}(z)|0\rangle = T_{22}(z)|0\rangle = 1, \qquad D_{22}(z)|0\rangle = T_{33}(z)|0\rangle = 1, \\ D_{12}(z)|0\rangle = T_{23}(z)|0\rangle = 0, \qquad D_{21}(z)|0\rangle = T_{32}(z)|0\rangle = 0. \end{aligned} \qquad (3.31)$$

The matrix $\mathcal{T}_0^{(a)}(z)$ is the monodromy matrix of the inhomogeneous $\mathfrak{gl}_2$-invariant $XXX$ chain of the length $a$. The role of the inhomogeneity parameters is played by the parameters $\bar{u} = \{u_1,\ldots,u_a\}$. The quantum space $\mathcal{H}^{(a)}$ of this model is the tensor product

$$\mathcal{H}^{(a)} = V_1 \otimes \cdots \otimes V_a, \qquad \text{where} \qquad V_j \sim \mathbb{C}^2. \qquad (3.32)$$

This is exactly the space containing the vector $\mathbb{F}(\bar{u})$: $\mathbb{F}(\bar{u}) \in \mathcal{H}^{(a)}$.

A vacuum vector in the space $\mathcal{H}^{(a)}$ has the form

$$|\Omega^{(a)}\rangle = \underbrace{\left(\begin{smallmatrix}1\\0\end{smallmatrix}\right) \otimes \cdots \otimes \left(\begin{smallmatrix}1\\0\end{smallmatrix}\right)}_{a \quad \text{times}}. \qquad (3.33)$$

If we present $\mathcal{T}^{(a)}(z)$ as

$$\mathcal{T}^{(a)}(z) = \begin{pmatrix} \mathcal{A}^{(a)}(z) & \mathcal{B}^{(a)}(z) \\ \mathcal{C}^{(a)}(z) & \mathcal{D}^{(a)}(z) \end{pmatrix}, \qquad (3.34)$$

then

$$\mathcal{A}^{(a)}(z)|\Omega^{(a)}\rangle = f(z,\bar{u})|\Omega^{(a)}\rangle, \qquad \mathcal{D}^{(a)}(z)|\Omega^{(a)}\rangle = |\Omega^{(a)}\rangle. \qquad (3.35)$$

Thus, $\widehat{\mathcal{T}}_0^{(a)}(z)$ is the monodromy matrix of the $\mathcal{R}_2$ algebra, being the product of two monodromy matrices whose entries act in different spaces. It remains to act with $\text{tr}_0\, \widehat{\mathcal{T}}_0^{(a)}(z)$ on the vector $\mathbb{F}(\bar{u})|0\rangle$. Due to (3.31) we have

$$\text{tr}_0\, \widehat{\mathcal{T}}_0^{(a)}(z)\mathbb{F}(\bar{u})|0\rangle$$
$$= \left(D_{11}(z)\mathcal{A}^{(a)}(z) + D_{12}(z)\mathcal{C}^{(a)}(z) + D_{21}(z)\mathcal{B}^{(a)}(z) + D_{22}(z)\mathcal{D}^{(a)}(z)\right)\mathbb{F}(\bar{u})|0\rangle$$
$$= \left(\mathcal{A}^{(a)}(z) + \mathcal{D}^{(a)}(z)\right)\mathbb{F}(\bar{u})|0\rangle = \text{tr}\, \mathcal{T}^{(a)}(z)\mathbb{F}(\bar{u})|0\rangle. \quad (3.36)$$

Thus, if we do not want to have unwanted terms of the second type in the action of $\mathrm{tr}\,\mathbb{D}(z)$, we should require that $\mathbb{F}(\bar{u})$ be an eigenvector of the transfer matrix $\mathrm{tr}\,\mathcal{T}^{(a)}(z)$. Hence, $\mathbb{F}(\bar{u})$ has the form

$$\mathbb{F}(\bar{u}) = \mathcal{B}^{(a)}(v_1)\dots\mathcal{B}^{(a)}(v_b)|\Omega^{(a)}\rangle, \tag{3.37}$$

and the set $\bar{v} = \{v_1, \dots, v_b\}$ satisfies Bethe equations

$$f(v_j, \bar{u}) = \frac{f(v_j, \bar{v}_j)}{f(\bar{v}_j, v_j)}, \qquad j = 1, \dots, b. \tag{3.38}$$

Recall that here we use the shorthand notation (2.22) for the products of the $f$-functions. Observe also that the number of the operators $\mathcal{B}^{(a)}(v_b)$ cannot exceed the number of the sites of the chain $a$. Hence, $b \le a$.

Thus, we have conclude that the vector $\mathbb{F}(\bar{u})$ should be the eigenvector of the transfer matrix of the inhomogeneous $XXX$ chain with the inhomogeneities $\bar{u}$. This is the main idea of the nested algebraic Bethe ansatz. Namely, the on-shell Bethe vector of the model with the $\mathfrak{gl}_3$-invariant $R$-matrix is expressed in terms of the on-shell Bethe vector of the model with the $\mathfrak{gl}_2$-invariant $R$-matrix.

We see that $\mathbb{F}(\bar{u})$ depends on the set of auxiliary parameters $\bar{v}$, that is $\mathbb{F}(\bar{u}) = \mathbb{F}(\bar{u}; \bar{v})$. Hence, the vector $|\Psi_a(\bar{u})\rangle$ also depends on these parameters: $|\Psi_a(\bar{u})\rangle = |\Psi_{a,b}(\bar{u}; \bar{v})\rangle$. By construction, this vector is symmetric over the variables $\bar{v}$. It turns out that it is also symmetric over the variables $\bar{u}$, however, this symmetry is far from the evident. We postpone the corresponding proof till section 4.2. For the moment we assume that $|\Psi_{a,b}(\bar{u}; \bar{v})\rangle$ is symmetric over $\bar{u}$.

Thus, we obtain

$$\mathrm{tr}_0\,\mathbb{D}_0(z)|\Psi_{a,b}(\bar{u}; \bar{v})\rangle = \tau_D(z|\bar{u}; \bar{v})|\Psi_{a,b}(\bar{u}; \bar{v})\rangle + \mathcal{Z}, \tag{3.39}$$

where

$$\tau_D(z|\bar{u}; \bar{v}) = f(z, \bar{u})f(\bar{v}, z) + f(z, \bar{v}). \tag{3.40}$$

Up to now we used the first scheme of commutation only. Let us take into account the second scheme.

It is clear that if we use at least once the second scheme, then the operators $D_{ij}(z)$ and $B_{\beta_k}(u_l)$ exchange their arguments. Therefore, after moving the matrix $\mathbb{D}_0(z)$ through the product of $\mathbb{B}_j(u_j)$ to the right it will have an argument $u_k \in \bar{u}$. At the same time one of the operator-valued vectors $\mathbb{B}_{j_0}$ will have the argument $z$. Other operator-valued vectors will have arguments $u_j$ such that $u_j \ne u_k$.

Further arguments closely resemble those used in computing unwanted terms in $\mathcal{R}_2$ algebra. Due to the symmetry of $|\Psi_{a,b}(\bar{u}; \bar{v})\rangle$ over $\bar{u}$ it is enough to consider the case when $\mathbb{B}_1(u_1)$ looses its argument and absorbs the argument $z$, while $\mathbb{D}_0(z)$ arrives at the extreme right position with the argument $u_1$. Then at the first step we should use the second scheme in (3.26), otherwise $\mathbb{D}_0(z)$ never absorbs the argument $u_1$. We have

$$\mathrm{tr}_0\,\mathbb{D}_0(z)|\Psi_{a,b}(\bar{u}; \bar{v})\rangle = g(u_1, z)\,\mathrm{tr}_0\,\mathbb{B}_1(z)\mathbb{D}_0(u_1)p_{01}\mathbb{B}_2(u_2)\dots\mathbb{B}_a(u_a)\mathbb{F}(\bar{u}; \bar{v})|0\rangle + \mathcal{Z}, \tag{3.41}$$

where $\mathcal{Z}$ now denotes all the terms which do not give contributions to the desired result. Obviously, we can move $p_{01}$ to the right

$$\mathrm{tr}_0\,\mathbb{D}_0(z)|\Psi_{a,b}(\bar{u}; \bar{v})\rangle = g(u_1, z)\,\mathrm{tr}_0\,\mathbb{B}_1(z)\mathbb{D}_0(u_1)\mathbb{B}_2(u_2)\dots\mathbb{B}_a(u_a)p_{01}\mathbb{F}(\bar{u}; \bar{v})|0\rangle + \mathcal{Z}. \tag{3.42}$$

Moving further $\mathbb{D}_0(u_1)$ to the right we should keep its argument, therefore, now we can use only the first scheme. Then we obtain

$$\text{tr}_0 \mathbb{D}_0(z)|\Psi_{a,b}(\bar{u};\bar{v})\rangle = g(u_1,z)\,\text{tr}_0\,\mathbb{B}_1(z)\mathbb{B}_2(u_2)\ldots\mathbb{B}_a(u_a)$$
$$\times \mathbb{D}_0(u_1)r_{0a}(u_1,u_a)\ldots r_{02}(u_1,u_2)p_{01}\mathbb{F}(\bar{u};\bar{v})|0\rangle + \mathcal{Z}. \quad (3.43)$$

It is easy to see that

$$\mathbb{D}_0(u_1)r_{0a}(u_1,u_a)\ldots r_{02}(u_1,u_2)p_{01} = \frac{1}{c}\,\text{Res}\,\mathbb{D}_0(z)r_{0a}(z,u_a)\ldots r_{02}(z,u_2)r_{01}(z,u_1)\Big|_{z=u_1}$$
$$= \frac{1}{c}\,\text{Res}\,\mathbb{D}_0(z)\mathcal{T}_0^{(a)}(z)\Big|_{z=u_1} = \frac{1}{c}\,\text{Res}\,\widehat{\mathcal{T}}_0^{(a)}(z)\Big|_{z=u_1}. \quad (3.44)$$

Hence, we obtain

$$\text{tr}_0 \mathbb{D}_0(z)|\Psi_{a,b}(\bar{u};\bar{v})\rangle = g(u_1,z)\mathbb{B}_1(z)\mathbb{B}_2(u_2)\ldots\mathbb{B}_a(u_a)\frac{1}{c}\,\text{Res}\,\text{tr}_0\,\widehat{\mathcal{T}}_0^{(a)}(z)\mathbb{F}(\bar{u};\bar{v})|0\rangle\Big|_{z=u_1} + \mathcal{Z}$$
$$= g(u_1,z)\mathbb{B}_1(z)\mathbb{B}_2(u_2)\ldots\mathbb{B}_a(u_a)\frac{1}{c}\,\text{Res}\,\text{tr}_0\,\mathcal{T}_0^{(a)}(z)\mathbb{F}(\bar{u};\bar{v})|0\rangle\Big|_{z=u_1} + \mathcal{Z}, \quad (3.45)$$

where we used (3.36). Since $\mathbb{F}(\bar{u};\bar{v})$ is the eigenvector of $\text{tr}\,\mathcal{T}_0^{(a)}(z)$ for any $z$ we find

$$\text{tr}_0 \mathbb{D}_0(z)|\Psi_{a,b}(\bar{u};\bar{v})\rangle = g(u_1,z)\frac{1}{c}\,\text{Res}\,\tau_D(z|\bar{u};\bar{v})\Big|_{z=u_1}\mathbb{B}_1(z)\mathbb{B}_2(u_2)\ldots\mathbb{B}_a(u_a)\mathbb{F}(\bar{u};\bar{v})|0\rangle + \mathcal{Z}, \quad (3.46)$$

where the eigenvalue $\tau_D(z|\bar{u};\bar{v})$ is given by (3.40). Substituting this eigenvalue into (3.46) we eventually arrive at

$$\text{tr}_0 \mathbb{D}_0(z)|\Psi_{a,b}(\bar{u};\bar{v})\rangle = g(u_1,z)f(u_1,\bar{u}_1)f(\bar{v},u_1)|\Phi_{a,b}(z,u_1;\bar{u};\bar{v})\rangle + \mathcal{Z}, \quad (3.47)$$

where

$$|\Phi_{a,b}(z,u_1;\bar{u};\bar{v})\rangle = \mathbb{B}_1(z)\mathbb{B}_2(u_2)\ldots\mathbb{B}_a(u_a)\mathbb{F}(\bar{u};\bar{v})|0\rangle. \quad (3.48)$$

Thus, using the symmetry of $|\Psi_{a,b}(\bar{u};\bar{v})\rangle$ over $\bar{u}$ we find the total action of $\text{tr}\,\mathbb{D}(z) = T_{22}(z) + T_{33}(z)$ on the vector $|\Psi_{a,b}(\bar{u};\bar{v})\rangle$. It is given by

$$\text{tr}\,\mathbb{D}(z)|\Psi_{a,b}(\bar{u};\bar{v})\rangle = \tau_D(z|\bar{u};\bar{v})|\Psi_{a,b}(\bar{u};\bar{v})\rangle + \sum_{k=1}^{a} g(u_k,z)f(u_k,\bar{u}_k)f(\bar{v},u_k)|\Phi_{a,b}(z,u_k;\bar{u};\bar{v})\rangle, \quad (3.49)$$

where

$$|\Phi_{a,b}(z,u_k;\bar{u};\bar{v})\rangle = |\Phi_{a,b}(z,u_1;\bar{u};\bar{v})\rangle\Big|_{u_1\leftrightarrow u_k}. \quad (3.50)$$

### 3.4 Action of $A(z)$

We did not consider yet the action of the operator $T_{11}(z) = A(z)$ on the vector $|\Psi_{a,b}(\bar{u};\bar{v})\rangle$. This action is relatively simple and reminds the action of the operator $A$ in the case of the $\mathcal{R}_2$ algebra. Indeed, the commutation relation of $A(z)$ and $B_\beta(u)$ is

$$A(z)B_\beta(u) = f(u,z)B_\beta(u)A(z) + g(z,u)B_\beta(z)A(u), \qquad \beta = 1,2. \quad (3.51)$$

Therefore, the action of $A(z)$ does not produce unwanted terms of the second type. It is easy to see that the result should have the following form:

$$A(z)|\Psi_{a,b}(\bar{u};\bar{v})\rangle = \tau_A(z|\bar{u};\bar{v})|\Psi_{a,b}(\bar{u};\bar{v})\rangle + \sum_{k=1}^{a} \Lambda_k|\Phi_{a,b}(z,u_k;\bar{u};\bar{v})\rangle, \quad (3.52)$$

where $\tau_A(z|\bar{u};\bar{v})$ and $\Lambda_k$ are some numerical coefficients. They can be found exactly in the same manner as in the $\mathcal{R}_2$ case. Recall this procedure. As usual, let us call the first term in the r.h.s. of (3.51) *the first scheme of commutation*. Respectively, the second term in the r.h.s. of (3.51) is called *the second scheme of commutation*. In the first scheme both $A$ and $B_\beta$ keep their original arguments, while in the second scheme they exchange them.

Obviously, in order to obtain the contribution proportional to $\tau_A(z|\bar{u};\bar{v})$ we should use the first scheme of commutation only. We obtain

$$A(z)|\Psi_{a,b}(\bar{u};\bar{v})\rangle = f(\bar{u},z)\mathbb{B}_1(u_1)\mathbb{B}_2(u_2)\dots\mathbb{B}_a(u_a)A(z)\mathbb{F}(\bar{u};\bar{v})|0\rangle + \mathcal{Z}, \qquad (3.53)$$

where $\mathcal{Z}$ denotes all unwanted terms. Acting with $A(z)$ on $|0\rangle$ we immediately obtain

$$\tau_A(z|\bar{u};\bar{v}) = \lambda_1(z)f(\bar{u},z), \qquad (3.54)$$

and thus, this coefficient actually does not depend on the set $\bar{v}$.

In order to find the coefficients $\Lambda_k$ it is enough to find one of them, say, $\Lambda_1$. Here we use the symmetry of $|\Psi_{a,b}(\bar{u};\bar{v})\rangle$ over $\bar{u}$. Then at the first step we must use the second scheme of commutation, and after this we must use only the first scheme of commutation. This gives us

$$A(z)|\Psi_{a,b}(\bar{u};\bar{v})\rangle = g(z,u_1)f(\bar{u}_1,u_1)\mathbb{B}_1(z)\mathbb{B}_2(u_2)\dots\mathbb{B}_a(u_a)A(u_1)\mathbb{F}(\bar{u};\bar{v})|0\rangle + \mathcal{Z}, \qquad (3.55)$$

where now $\mathcal{Z}$ denotes all the terms that do not give contributions to the desired result. From this, we find

$$\Lambda_1 = \lambda_1(u_1)g(z,u_1)f(\bar{u}_1,u_1). \qquad (3.56)$$

Thus, we have computed the action of the transfer matrix $\operatorname{tr} T(z)$ on the vector $|\Psi_{a,b}(\bar{u};\bar{v})\rangle$. It is given by the sum of (3.49) and (3.52):

$$\operatorname{tr} T(z)|\Psi_{a,b}(\bar{u};\bar{v})\rangle = \tau(z|\bar{u};\bar{v})|\Psi_{a,b}(\bar{u};\bar{v})\rangle + \sum_{k=1}^{a} M_k|\Phi_{a,b}(z,u_k;\bar{u};\bar{v})\rangle. \qquad (3.57)$$

Here

$$\tau(z|\bar{u};\bar{v}) = \tau_A(z|\bar{u};\bar{v}) + \tau_D(z|\bar{u};\bar{v}) = \lambda_1(z)f(\bar{u},z) + f(z,\bar{u})f(\bar{v},z) + f(z,\bar{v}). \qquad (3.58)$$

The coefficients $M_k$ are

$$M_k = \lambda_1(u_k)g(z,u_k)f(\bar{u}_k,u_k) + g(u_k,z)f(u_k,\bar{u}_k)f(\bar{v},u_k). \qquad (3.59)$$

It is clear that $|\Psi_{a,b}(\bar{u};\bar{v})\rangle$ becomes on-shell Bethe vector, if we set $M_k = 0$ for $k = 1,\dots,a$, and for all complex $z$. This leads us to a new system of equations

$$\lambda_1(u_k) = \frac{f(u_k,\bar{u}_k)}{f(\bar{u}_k,u_k)}f(\bar{v},u_k), \qquad k = 1,\dots,a. \qquad (3.60)$$

Together with the already obtained equations (3.38)

$$f(v_j,\bar{u}) = \frac{f(v_j,\bar{v}_j)}{f(\bar{v}_j,v_j)}, \qquad j = 1,\dots,b, \qquad (3.61)$$

equations (3.60) form a system of $a + b$ equations on $a + b$ variables $\bar{u}$ and $\bar{v}$. This system is a particular case (corresponding to $\lambda_2(v) = \lambda_3(v)$) of Bethe equations for the models with the $\mathfrak{gl}_3$-invariant $R$-matrix.

Thus, the on-shell Bethe vectors have the form (3.20) with $\mathbb{F}(\bar{u};\bar{v})$ given by (3.37). The parameters $\bar{u}$ and $\bar{v}$ should satisfy the systems of equations (3.60) and (3.61).

### 3.5 General case

All the consideration above concerned the particular case $T_{23}(u)|0\rangle = 0$. What should be done in the general case? Remarkably, almost the entire scheme remains the same. We should only assume that the vector $\mathbb{F}(\bar{u}; \bar{v})$ in the expression

$$|\Psi_a(\bar{u}; \bar{v})\rangle = \mathbb{B}_1(u_1)\mathbb{B}_2(u_2)\ldots\mathbb{B}_a(u_a)\mathbb{F}(\bar{u}; \bar{v})|0\rangle \qquad (3.62)$$

has operator-valued components, which depend on the operators $D_{\alpha\beta}$ (in particular, on $T_{23}$). In other words, the vector $\mathbb{F}(\bar{u}; \bar{v})|0\rangle$ belongs to the tensor product $\mathcal{H} \otimes \mathcal{H}^{(a)}$ (because the action of $T_{23}$ on the vacuum $|0\rangle$ gives a vector in the space $\mathcal{H}$).

We also should take into account that

$$\begin{aligned} D_{11}(z)|0\rangle = T_{22}(z)|0\rangle = \lambda_2(z)|0\rangle, \\ D_{22}(z)|0\rangle = T_{33}(z)|0\rangle = \lambda_3(z)|0\rangle, \end{aligned} \qquad (3.63)$$

and now we have no the restriction $\lambda_2(z)/\lambda_3(z) = const$.

We start with representation (3.62) and act on this vector with $\operatorname{tr}\mathbb{D}(z)$. Using the first scheme of commutation only we arrive at (3.29):

$$\operatorname{tr}_0 \mathbb{D}_0(z)|\Psi_a(\bar{u})\rangle = \mathbb{B}_1(u_1)\mathbb{B}_2(u_2)\ldots\mathbb{B}_a(u_a)\operatorname{tr}_0 \widehat{\mathcal{T}}_0^{(a)}(z)\mathbb{F}(\bar{u}; \bar{v})|0\rangle + \mathcal{Z}. \qquad (3.64)$$

We see that $\mathbb{F}(\bar{u}; \bar{v})|0\rangle$ should be an eigenvector of the transfer matrix $\operatorname{tr}\widehat{\mathcal{T}}^{(a)}(z)$ (3.30).

The form of equation (3.64) coincides with the form of (3.29) in which the monodromy matrix $\widehat{\mathcal{T}}^{(a)}(z)$ (3.30) first appeared. The vector $\mathbb{F}(\bar{u}; \bar{v})|0\rangle$ also still belongs to the tensor product $\mathcal{H} \otimes \mathcal{H}^{(a)}$. However, in the case considered above, we had a factorization: the vacuum vector $|0\rangle$ belonged to the space $\mathcal{H}$, while the vector $\mathbb{F}(\bar{u}; \bar{v})$ belonged to the space $\mathcal{H}^{(a)}$. Now there is no such factorization, and we must consider the vector $\mathbb{F}(\bar{u}; \bar{v})|0\rangle$ as a whole.

Despite this difference, we can follow the same scheme as before. Let

$$\widehat{\mathcal{T}}^{(a)}(z) = \begin{pmatrix} \widehat{\mathcal{A}}^{(a)}(z) & \widehat{\mathcal{B}}^{(a)}(z) \\ \widehat{\mathcal{C}}^{(a)}(z) & \widehat{\mathcal{D}}^{(a)}(z) \end{pmatrix}. \qquad (3.65)$$

The entries of this matrix act in $\mathcal{H} \otimes \mathcal{H}^{(a)}$ with the vacuum vector $|0\rangle \otimes |\Omega^{(a)}\rangle$. The vacuum eigenvalues of the diagonal entries $\widehat{\mathcal{T}}_{ii}^{(a)}(z)$ (i.e. of $\widehat{\mathcal{A}}^{(a)}(z)$ and $\widehat{\mathcal{D}}^{(a)}(z)$) are given by the products of the vacuum eigenvalues of $D_{ii}(z)$ and $\mathcal{T}_{ii}^{(a)}(z)$:

$$\begin{aligned} \widehat{\mathcal{A}}^{(a)}(z)|0\rangle \otimes |\Omega^{(a)}\rangle = \lambda_2(z)f(z, \bar{u})|0\rangle \otimes |\Omega^{(a)}\rangle, \\ \widehat{\mathcal{D}}^{(a)}(z)|0\rangle \otimes |\Omega^{(a)}\rangle = \lambda_3(z)|0\rangle \otimes |\Omega^{(a)}\rangle. \end{aligned} \qquad (3.66)$$

Thus, the eigenvectors of $\operatorname{tr}\widehat{\mathcal{T}}^{(a)}(z)$ have the form similar to (3.37)

$$\mathbb{F}(\bar{u}; \bar{v}) = \widehat{\mathcal{B}}^{(a)}(v_1)\ldots\widehat{\mathcal{B}}^{(a)}(v_b)|0\rangle \otimes |\Omega^{(a)}\rangle, \qquad (3.67)$$

provided the set $\bar{v}$ satisfies Bethe equations

$$\frac{\lambda_2(v_j)}{\lambda_3(v_j)} = \frac{f(v_j, \bar{v}_j)}{f(\bar{v}_j, v_j)}\frac{1}{f(v_j, \bar{u})}, \qquad j = 1, \ldots, b. \qquad (3.68)$$

Observe that now the matrix $\widehat{\mathcal{T}}^{(a)}(z)$ is no longer the monodromy matrix of the $XXX$ chain. It is the product of two monodromy matrices $\mathbb{D}(z)$ and $\mathcal{T}^{(a)}(z)$. Therefore, there is no restriction on the number of the operators $\widehat{\mathcal{B}}^{(a)}$ in (3.67). Thus, we do not have the constraint $b \leq a$ as it was previously.

Thus, we obtain

$$\operatorname{tr} \mathbb{D}(z)|\Psi_{a,b}(\bar{u};\bar{v})\rangle = \widehat{\tau}_D(z|\bar{u};\bar{v})|\Psi_{a,b}(\bar{u};\bar{v})\rangle + \mathcal{Z}, \tag{3.69}$$

where now

$$\widehat{\tau}_D(z|\bar{u};\bar{v}) = \lambda_2(z)f(z,\bar{u})f(\bar{v},z) + \lambda_3(z)f(z,\bar{v}). \tag{3.70}$$

Consideration of the unwanted terms of the first type produced by the action of $\operatorname{tr} \mathbb{D}(z)$ can be done exactly in the same manner as before. This leads us to the analog of (3.47)

$$\operatorname{tr} \mathbb{D}(z)|\Psi_{a,b}(\bar{u};\bar{v})\rangle = \lambda_2(u_1)g(u_1,z)f(u_1,\bar{u}_1)f(\bar{v},u_1)|\Phi_{a,b}(z,u_1;\bar{u};\bar{v})\rangle + \mathcal{Z}, \tag{3.71}$$

where $|\Phi_{a,b}(z,u_1;\bar{u};\bar{v})\rangle$ is still given by (3.48). The only natural difference between (3.71) and (3.47) is that we have the additional factor $\lambda_2(u_1)$. Previously this factor was equal to 1. Thus, the total action of $\operatorname{tr} \mathbb{D}(z)$ on the vector $|\Psi_{a,b}(\bar{u};\bar{v})\rangle$ has the form

$$\operatorname{tr} \mathbb{D}(z)|\Psi_{a,b}(\bar{u};\bar{v})\rangle = \widehat{\tau}_D(z|\bar{u};\bar{v})|\Psi_{a,b}(\bar{u};\bar{v})\rangle$$
$$+ \sum_{k=1}^{a} \lambda_2(u_k)g(u_k,z)f(u_k,\bar{u}_k)f(\bar{v},u_k)|\Phi_{a,b}(z,u_k;\bar{u};\bar{v})\rangle, \tag{3.72}$$

where $|\Phi_{a,b}(z,u_k;\bar{u};\bar{v})\rangle$ is given by (3.50). Recall that this result is obtained under assumption that $|\Psi_{a,b}(\bar{u};\bar{v})\rangle$ is symmetric over the set $\bar{u}$. This symmetry still should be proved.

Considering the action of $A(z)$ on $|\Psi_{a,b}(\bar{u};\bar{v})\rangle$ we deal with only one new problem. Namely, we should prove that

$$A(z)\mathbb{F}(\bar{u};\bar{v})|0\rangle = \lambda_1(z)\mathbb{F}(\bar{u};\bar{v})|0\rangle. \tag{3.73}$$

This property was obvious in the previous case, because $\mathbb{F}(\bar{u};\bar{v})$ did not belong to the space $\mathcal{H}$, in which the operator $A(z)$ acted. Now $\mathbb{F}(\bar{u};\bar{v})|0\rangle \in \mathcal{H} \otimes \mathcal{H}^{(a)}$, therefore, the property (3.73) should be proved. However, the proof immediately follows from proposition 1.3. Indeed, since the components of $\mathbb{F}(\bar{u};\bar{v})|0\rangle$ depend on the operators $D_{\alpha\beta}$, they only contain quasiparticles of the second color. Thus, the action of $A(z)$ on each of these components reduces to multiplication by $\lambda_1(z)$.

In all other respects the action of $A(z)$ on $|\Psi_{a,b}(\bar{u};\bar{v})\rangle$ can be derived via the same lines leading eventually to equation (3.52), where $\tau_A(z|\bar{u};\bar{v})$ and $\Lambda_k$ respectively are given by (3.54) and (3.56).

Thus, the action of the transfer matrix $\operatorname{tr} T(z)$ on the vector $|\Psi_{a,b}(\bar{u};\bar{v})\rangle$ reads

$$\operatorname{tr} T(z)|\Psi_{a,b}(\bar{u};\bar{v})\rangle = \widehat{\tau}(z|\bar{u};\bar{v})|\Psi_{a,b}(\bar{u};\bar{v})\rangle + \sum_{k=1}^{a} \widehat{M}_k|\Phi_{a,b}(z,u_k;\bar{u};\bar{v})\rangle. \tag{3.74}$$

Here

$$\widehat{\tau}(z|\bar{u};\bar{v}) = \lambda_1(z)f(\bar{u},z) + \lambda_2(z)f(z,\bar{u})f(\bar{v},z) + \lambda_3(z)f(z,\bar{v}), \tag{3.75}$$

and the coefficients $\widehat{M}_k$ have the form

$$\widehat{M}_k = \lambda_1(u_k)g(z,u_k)f(\bar{u}_k,u_k) + \lambda_2(u_k)g(u_k,z)f(u_k,\bar{u}_k)f(\bar{v},u_k). \tag{3.76}$$

Setting $\widehat{M}_k = 0$ for $k = 1,\ldots,a$ we obtain a system of equations

$$\frac{\lambda_1(u_k)}{\lambda_2(u_k)} = \frac{f(u_k,\bar{u}_k)}{f(\bar{u}_k,u_k)}f(\bar{v},u_k), \qquad k = 1,\ldots,a. \tag{3.77}$$

If the sets $\bar{u}$ and $\bar{v}$ satisfy the systems (3.68) and (3.77), then $|\Psi_{a,b}(\bar{u};\bar{v})\rangle$ becomes the on-shell Bethe vector.

SciPost Phys. Lect. Notes 19 (2020)

## 3.6 Definition of Bethe vectors

At this point we can turn back to the problem of off-shell Bethe vectors (or simply Bethe vectors). Now we are able to give their definition at least for the $\mathfrak{gl}_3$ based models.

**Definition 3.1.** *We call a state $|\Psi_{a,b}(\bar{u};\bar{v})\rangle$ an off-shell Bethe vector of the $\mathcal{R}_3$ algebra, if it has the form (3.62), where the vector $\mathbb{F}(\bar{u};\bar{v})|0\rangle$ has the form (3.67). In other words,*

$$|\Psi_{a,b}(\bar{u};\bar{v})\rangle = \mathbb{B}_1(u_1)\mathbb{B}_2(u_2)\dots\mathbb{B}_a(u_a)\widehat{\mathcal{B}}^{(a)}(v_1)\dots\widehat{\mathcal{B}}^{(a)}(v_b)|0\rangle \otimes |\Omega^{(a)}\rangle. \tag{3.78}$$

*Here $\mathbb{B}(u_i)$ are the operator-valued vector rows of the original monodromy matrix (3.6), and $\widehat{\mathcal{B}}^{(a)}(v_k)$ are the creation operators of the auxiliary monodromy matrix (3.65). The Bethe parameters $\bar{u}$ and $\bar{v}$ are arbitrary complex numbers. The cardinalities of the sets $\bar{u}$ and $\bar{v}$ respectively are $\#\bar{u} = a$ and $\#\bar{v} = b$, where $a, b = 0, 1, \dots$.*

*Remark.* Note that when we looked for the on-shell Bethe vectors we required the vector $\mathbb{F}(\bar{u};\bar{v})|0\rangle$ to be the eigenvector of the transfer matrix $\operatorname{tr}\widehat{\mathcal{T}}^{(a)}(z)$ (3.65). This requirement led us to the set of equations (3.68). Now we do not impose this constraint. Thus, $\mathbb{F}(\bar{u};\bar{v})|0\rangle$ is not necessarily the eigenvector of $\operatorname{tr}\widehat{\mathcal{T}}^{(a)}(z)$, but it has the form (3.67).

If the parameters $\bar{u}$ and $\bar{v}$ satisfy the system of Bethe equations (3.68) and (3.77), that is

$$\begin{aligned}
\frac{\lambda_1(u_k)}{\lambda_2(u_k)} &= \frac{f(u_k,\bar{u}_k)}{f(\bar{u}_k,u_k)}f(\bar{v},u_k), & k &= 1,\dots,a, \\
\frac{\lambda_2(v_j)}{\lambda_3(v_j)} &= \frac{f(v_j,\bar{v}_j)}{f(\bar{v}_j,v_j)}\frac{1}{f(v_j,\bar{u})}, & j &= 1,\dots,b,
\end{aligned} \tag{3.79}$$

then the vector (3.78) becomes an on-shell Bethe vector.

Formally, definition 3.1 uniquely fixes the Bethe vector as a polynomial in the creation operators[5] $T_{ij}$ with $i < j$ acting on the vacuum vector. However, these operators, generally speaking, do not commute with each other. Therefore, their reordering leads to new representations for the Bethe vectors. Formula (3.78) is one of such representations.

Unfortunately, equation (3.78) does not give an explicit dependence of the Bethe vector on the creation operators. We will derive such explicit dependence later. In the meantime, as a example, consider a couple of the simplest cases.

The most simple case is $a = 0$, that is $\bar{u} = \emptyset$. Then $\widehat{\mathcal{T}}^{(a)}(z) = \mathbb{D}(z)$, and hence, $\widehat{\mathcal{B}}^{(a)}(z) = T_{23}(z)$. We obtain

$$|\Psi_{0,b}(\emptyset;\bar{v})\rangle = T_{23}(\bar{v})|0\rangle. \tag{3.80}$$

We see that in this case the $\mathcal{R}_3$ Bethe vector reduces to the $\mathcal{R}_2$ Bethe vector. This is not surprising, because the state $|\Psi_{0,b}(\emptyset;\bar{v})\rangle$ has quasiparticles of the color 2 only. Thus, it should coincide with the Bethe vector of the $\mathfrak{gl}_2$ based models.

Let now $a = b = 1$. Then

$$|\Psi_{1,1}(u;v)\rangle = \mathbb{B}(u)\widehat{\mathcal{B}}^{(1)}(v)|0\rangle \otimes |\Omega^{(1)}\rangle, \tag{3.81}$$

where $|\Omega^{(1)}\rangle = \begin{pmatrix} 1 \\ 0 \end{pmatrix}$. The matrix $\widehat{\mathcal{T}}^{(1)}(v)$ is given by (3.30) at $z = v$ and $a = 1$, that is

$$\widehat{\mathcal{T}}_0^{(1)}(v) = \mathbb{D}_0(v)\mathcal{T}_0^{(1)}(v), \tag{3.82}$$

---

[5]Strictly speaking, equation (3.78) also contains the neutral operators $T_{ii}$. However, their action on the vacuum vector can be replaced by the corresponding eigenvalues.

where

$$\mathcal{T}_0^{(1)}(v) = r_{01}(v, u) = \begin{pmatrix} \mathbf{1} & 0 \\ 0 & \mathbf{1} \end{pmatrix}_0 + g(v, u) \begin{pmatrix} E_{11} & E_{21} \\ E_{12} & E_{22} \end{pmatrix}_0, \tag{3.83}$$

and we stressed by the subscript 0 that the auxiliary space of this matrix is $V_0$. Thus,

$$\widehat{\mathcal{B}}^{(1)}(v) = g(v, u)D_{11}(v)E_{21} + D_{12}(v)\bigl(1 + g(v, u)E_{22}\bigr), \tag{3.84}$$

and hence, the action of $\widehat{\mathcal{B}}^{(1)}(v)$ on $|\Omega^{(1)}\rangle$ is given by

$$\widehat{\mathcal{B}}^{(1)}(v)\left(\begin{smallmatrix} 1 \\ 0 \end{smallmatrix}\right) = g(v, u)D_{11}(v)\left(\begin{smallmatrix} 0 \\ 1 \end{smallmatrix}\right) + D_{12}(v)\left(\begin{smallmatrix} 1 \\ 0 \end{smallmatrix}\right). \tag{3.85}$$

Equation (3.85) gives us the components of the vector $\mathbb{F}(u; v)|0\rangle$ in the space $\mathcal{H}^{(1)}$:

$$\begin{aligned} F_1(u; v)|0\rangle &= D_{12}(v)|0\rangle = T_{23}(v)|0\rangle, \\ F_2(u; v)|0\rangle &= g(v, u)D_{11}(v)|0\rangle = g(v, u)T_{22}(v)|0\rangle = g(v, u)\lambda_2(v)|0\rangle. \end{aligned} \tag{3.86}$$

Thus, we obtain the explicit expression for the Bethe vector $|\Psi_{1,1}(u; v)\rangle$:

$$\begin{aligned} |\Psi_{1,1}(u; v)\rangle &= B_1(u)F_1(u; v)|0\rangle + B_2(u)F_2(u; v)|0\rangle \\ &= T_{12}(u)T_{23}(v)|0\rangle + g(v, u)\lambda_2(v)T_{13}(u)|0\rangle, \end{aligned} \tag{3.87}$$

what coincides with (2.19). This Bethe vector becomes on-shell, if $u$ and $v$ satisfy the system (3.79). It is easy to see that for $a = b = 1$ this system turns into (2.20).

## 3.7 Remarks about different embedding

Let us say few words about parametrization (3.7). This parametrization also can be used for constructing the Bethe vectors. The general strategy is the same in this case, however, several minor details are different. We recommend that the reader obtain himself the formula for the Bethe vector using parametrization (3.7). We restrict ourselves with several comments.

In the case of embedding (3.7) we deal with a $2 \times 2$ matrix $\mathbb{A}'$ and a two-component vector-column

$$\mathbb{B}'(v) = \begin{pmatrix} B_1'(v) \\ B_2'(v) \end{pmatrix} = \begin{pmatrix} T_{13}(v) \\ T_{23}(v) \end{pmatrix}. \tag{3.88}$$

Instead of (3.62) we use the following ansatz for the Bethe vectors:

$$|\Psi_{a,b}(\bar{u}; \bar{v})\rangle = \Bigl(\mathbb{B}_1'(v_1)\mathbb{B}_2'(v_2)\ldots\mathbb{B}_b'(v_b)\Bigr)^T \mathbb{F}'(\bar{u}; \bar{v})|0\rangle. \tag{3.89}$$

Here the superscript $T$ means transposition in the space $V_1 \otimes \cdots \otimes V_b$ (where each $V_k \sim \mathbb{C}^2$). The commutation relations between the operator-valued matrix $\mathbb{A}'$ and the operator-valued vector-column $\mathbb{B}'$ have the form

$$\mathbb{A}_0'(z)\mathbb{B}_1'(v) = r_{01}(v, z)\mathbb{B}_1'(v)\mathbb{A}_0'(z) + g(z, v)p_{01}\mathbb{B}_1'(z)\mathbb{A}_0'(v). \tag{3.90}$$

Here, in distinction of (3.22), the $R$-matrix $r_{01}(v, z)$ and the permutation matrix $p_{01}$ act on other matrices from the left. Therefore, moving $\mathbb{A}_0'(z)$ through the product of $\mathbb{B}_j'(v_j)$ we obtain the product of the $R$-matrices in the extreme left position. However, after the transposition we obtain the product of the $R$-matrices to the right from the product of the $\mathbb{B}_j'(v_j)$-operators.

Then we require that the vector $\mathbb{F}'(\bar{u};\bar{v})|0\rangle$ would be an eigenvector of the monodromy matrix $\widetilde{\mathcal{T}}^{(b)}(z)$:

$$\widetilde{\mathcal{T}}_0^{(b)}(z) = r'_{01}(z,v_1)\ldots r'_{0b}(z,v_b)\mathbb{A}'_0(z). \tag{3.91}$$

Here $r'_{0k}(u,v) = r_{0k}^{t_k}(-u,-v)$, where $t_k$ means the transposition in the space $V_k$. These matrices appear when we take the transposition of the product $r_{01}\ldots r_{0b}$ in the space $V_1 \otimes \cdots \otimes V_b$. We leave to the reader to prove that $r'(u,v)$ satisfies the $RTT$-relation with the $R$-matrix $r(u,v)$:

$$r_{12}(u,v)r'_{13}(u,w)r'_{23}(v,w) = r'_{23}(v,w)r'_{13}(u,w)r_{12}(u,v). \tag{3.92}$$

Thus, the product $r'_{01}(z,v_1)\ldots r'_{0b}(z,v_b)$ also satisfies the $RTT$-relation with the $R$-matrix $r(u,v)$.

The entries of $\widetilde{\mathcal{T}}^{(b)}(z)$ act in the space $\mathcal{H}^{(b)} \otimes \mathcal{H}$, where $\mathcal{H}^{(b)} = V_1 \otimes \cdots \otimes V_b$ has the following vacuum vector:

$$|\widetilde{\Omega}^{(b)}\rangle = \underbrace{\begin{pmatrix} 0 \\ 1 \end{pmatrix} \otimes \cdots \otimes \begin{pmatrix} 0 \\ 1 \end{pmatrix}}_{b \quad \text{times}}. \tag{3.93}$$

If we set

$$\widetilde{\mathcal{T}}^{(b)}(z) = \begin{pmatrix} \widetilde{\mathcal{A}}^{(b)}(z) & \widetilde{\mathcal{B}}^{(b)}(z) \\ \widetilde{\mathcal{C}}^{(b)}(z) & \widetilde{\mathcal{D}}^{(b)}(z) \end{pmatrix}, \tag{3.94}$$

then the vector $\mathbb{F}'(\bar{u};\bar{v})|0\rangle$ has the following form:

$$\mathbb{F}'(\bar{u};\bar{v})|0\rangle = \widetilde{\mathcal{B}}^{(b)}(u_1)\ldots\widetilde{\mathcal{B}}^{(b)}(u_a)|\widetilde{\Omega}^{(b)}\rangle \otimes |0\rangle. \tag{3.95}$$

We will see below that formulas for the Bethe vectors based on the embeddings (3.6) and (3.7) look very different. First, they have different ordering of the creation operators. Second, some of those operators have different arguments. Nevertheless, these different representations describe the same Bethe vector $|\Psi_{a,b}(\bar{u};\bar{v})\rangle$.

## 3.8 Remarks about $\mathfrak{gl}_N$ Bethe vectors

The scheme described above does not change in the case of the models with $\mathfrak{gl}_N$-invariant $R$-matrix or its $q$-deformed analog (1.14). We present the $N \times N$ monodromy matrix as a $2 \times 2$ block-matrix

$$T(u) = \begin{pmatrix} A(u) & \mathbb{B}(u) \\ \mathbb{C}(u) & \mathbb{D}(u) \end{pmatrix}. \tag{3.96}$$

Now the block $\mathbb{D}$ has the size $(N-1) \times (N-1)$. Respectively, $\mathbb{B}$ is the operator-valued vector-row with $N-1$ components

$$\mathbb{B}(u) = (B_1(u),\ldots,B_{N-1}(u)) = (T_{12}(u),\ldots,T_{1,N}(u)). \tag{3.97}$$

Then we look for the on-shell Bethe vectors in the form similar to (3.62)

$$|\Psi\rangle = \mathbb{B}_1(u_1)\ldots\mathbb{B}_a(u_a)\mathbb{F}|0\rangle. \tag{3.98}$$

The vector $\mathbb{F}|0\rangle$ belongs to the space $\mathcal{H} \otimes \mathcal{H}^{(a)}$, where $\mathcal{H}^{(a)}$ is the tensor product

$$\mathcal{H}^{(a)} = V_1 \otimes \cdots \otimes V_a, \qquad \text{where} \qquad V_j \sim \mathbb{C}^{N-1}. \tag{3.99}$$

Otherwise, all the arguments remain unchanged. They lead us to the conclusion that the vector $\mathbb{F}|0\rangle$ must be an eigenvector of the transfer matrix of the model with $\mathfrak{gl}_{N-1}$-invariant $R$-matrix.

Of course, an explicit expression for this vector is no longer given by (3.67), but is much more complicated.

It is clear that using this method we obtain Bethe vectors depending on $N-1$ sets of variables[6] $\bar{t} = \{\bar{t}^1, \ldots, \bar{t}^{N-1}\}$. In its turn, every set $\bar{t}^k$ consists of individual elements $\bar{t}^k = \{t_1^k, \ldots, t_{a_k}^k\}$, where $a_k = \#\bar{t}^k$. Then we can refine formula (3.98) as

$$|\Psi_{\bar{a}}(\bar{t})\rangle = \mathbb{B}_1(t_1^1)\ldots\mathbb{B}_{a_1}(t_{a_1}^1)\mathbb{F}(\bar{t})|0\rangle, \tag{3.100}$$

where $\bar{a}$ is a multi-index consisting of the cardinalities $\bar{a} = \{a_1, \ldots, a_{N-1}\}$. In this formula, $\mathbb{F}(\bar{t})|0\rangle$ is the Bethe vector of the monodromy matrix

$$\widehat{\mathcal{T}}_0^{(a_1)}(z) = \mathbb{D}_0(z)r_{0,a_1}(z, t_{a_1}^1)\ldots r_{01}(z, t_1^1), \tag{3.101}$$

where now the $R$-matrix $r(u,v)$ acts in $\mathbb{C}^{N-1} \otimes \mathbb{C}^{N-1}$. Equation (3.100) can be taken as the definition of the off-shell Bethe vector.

This vector becomes on-shell if the Bethe parameters $\bar{t}$ satisfy a system of Bethe equations. It has the following form:

$$\frac{\lambda_k(t_j^k)}{\lambda_{k+1}(t_j^k)} = \frac{f(t_j^k, \bar{t}^k)f(\bar{t}^{k+1}, t_j^k)}{f(\bar{t}^k, t_j^k)f(t_j^k, \bar{t}^{k-1})}, \qquad \begin{array}{l} k = 1, \ldots, N-1, \\ j = 1, \ldots, a_k. \end{array} \tag{3.102}$$

Here we set by definition $\bar{t}^0 = \bar{t}^N = \emptyset$ and used the shorthand notation for the products of the $f$-functions over the sets $\bar{t}^k$.

# 4 Trace formula

## 4.1 Bethe vector via trace formula

The resulting formulas for Bethe vectors depend to a large extent on the embedding of the $\mathcal{R}_2$ algebra in the $\mathcal{R}_3$ algebra. Depending on the embedding the roles of the parameters $\bar{u}$ and $\bar{v}$ also are very different. In this section, we consider one more representation for the Bethe vectors [10–12]. The main advantage of this representation is that it can be easily generalized to the case of models with $\mathfrak{gl}_N$-invariant $R$-matrix (although we will still restrict ourselves to the case $\mathfrak{gl}_3$). Besides, the Bethe parameters $\bar{u}$ and $\bar{v}$ are included in this representation in a more symmetric way. Finally, the new formula for Bethe vectors will allow us to prove the symmetry of these vectors with respect to the parameters $\bar{u}$, which has not yet been proved.

Consider a tensor product $V_{k_1} \otimes \cdots \otimes V_{k_a} \otimes V_{n_1} \otimes \cdots \otimes V_{n_b}$, where each $V_j \sim \mathbb{C}^3$. Let

$$\mathbb{T}_{\bar{k}}(\bar{u}) = T_{k_1}(u_1)\ldots T_{k_a}(u_a), \qquad \mathbb{T}_{\bar{n}}(\bar{v}) = T_{n_1}(v_1)\ldots T_{n_b}(v_b), \tag{4.1}$$

and

$$\mathbb{R}_{\bar{n},\bar{k}}(\bar{v}, \bar{u}) = \prod_{i=1}^{b}\prod_{j=a}^{1} R_{n_i, k_j}(v_i, u_j). \tag{4.2}$$

Here every $T_j$ acts in $V_j \otimes \mathcal{H}$. Each $R$-matrix $R_{i,j}$ acts in $V_i \otimes V_j$. We would like to draw attention of the reader to the ordering of the $R$-matrices in the double product (4.2). There the index

[6]In the $\mathfrak{gl}_3$-based models we have two sets of variables $\bar{u} = \bar{t}^1$ and $\bar{v} = \bar{t}^2$.

$i$ changes in the standard increasing direction, while the index $j$ changes in the decreasing direction. For example, for $a = b = 2$, the product (4.2) reads

$$\mathbb{R}_{\bar{n},\bar{k}}(\bar{v},\bar{u}) = R_{n_1,k_2}(v_1,u_2)R_{n_1,k_1}(v_1,u_1)R_{n_2,k_2}(v_2,u_2)R_{n_2,k_1}(v_2,u_1). \tag{4.3}$$

**Proposition 4.1.** *The off-shell Bethe vectors of the $\mathfrak{gl}_3$-invariant models have the following form:*

$$|\Psi_{a,b}(\bar{u};\bar{v})\rangle = \operatorname{tr}_{\bar{k},\bar{n}}\Big(\mathbb{T}_{\bar{k}}(\bar{u})\mathbb{T}_{\bar{n}}(\bar{v})\mathbb{R}_{\bar{n},\bar{k}}(\bar{v},\bar{u})E_{k_1}^{21}\dots E_{k_a}^{21}E_{n_1}^{32}\dots E_{n_b}^{32}\Big)|0\rangle. \tag{4.4}$$

*The trace is taken over all the spaces $V_{k_1},\dots,V_{k_a}, V_{n_1},\dots,V_{n_b}$. The matrices $E_{k_j}^{21}$ and $E_{n_j}^{32}$ are the standard basis matrices that respectively act in the spaces $V_{k_j}$ and $V_{n_j}$. In distinction of the previous section we use here superscripts for the different standard basis matrices.*

Equation (4.4) is known as a *trace formula*. We will prove that the trace formula is equivalent to the representation obtained in the previous section.

*Proof.* Let us present all the monodromy matrices in (4.4) as

$$T_{k_s}(u_s) = \sum_{i,j=1}^{3} T_{i,j}(u_s)E_{k_s}^{i,j}, \qquad T_{n_p}(v_p) = \sum_{\alpha,\beta=1}^{3} T_{\alpha,\beta}(v_p)E_{n_p}^{\alpha,\beta}. \tag{4.5}$$

Substituting this into the trace formula we obtain

$$|\Psi_{a,b}(\bar{u};\bar{v})\rangle = \sum_{\bar{i},\bar{j}=1}^{3}\sum_{\bar{\alpha},\bar{\beta}=1}^{3} T_{i_1,j_1}(u_1)\dots T_{i_a,j_a}(u_a)T_{\alpha_1,\beta_1}(v_1)\dots T_{\alpha_b,\beta_b}(v_b)$$
$$\times \operatorname{tr}_{\bar{k},\bar{n}}\Big(\mathbb{R}_{\bar{n},\bar{k}}(\bar{v},\bar{u})E_{k_1}^{21}\dots E_{k_a}^{21}E_{n_1}^{32}\dots E_{n_b}^{32}\,E_{k_1}^{i_1,j_1}\dots E_{k_a}^{i_a,j_a}E_{n_1}^{\alpha_1,\beta_1}\dots E_{n_b}^{\alpha_b,\beta_b}\Big)|0\rangle, \tag{4.6}$$

where we have used cyclicity of the trace. The sum is taken over all $i_s$, $j_s$ (with $s = 1,\dots,a$) and all $\alpha_p$, $\beta_p$ (with $p = 1,\dots,b$). Taking the product of the $E$-matrices via $E^{ab}E^{cd} = \delta_{bc}E^{ad}$ we obtain that all $i_s = 1$ and all $\alpha_p = 2$. Then

$$|\Psi_{a,b}(\bar{u};\bar{v})\rangle = \sum_{\bar{j}=1}^{3}\sum_{\bar{\beta}=1}^{3} T_{1,j_1}(u_1)\dots T_{1,j_a}(u_a)T_{2,\beta_1}(v_1)\dots T_{2,\beta_b}(v_b)$$
$$\times \operatorname{tr}_{\bar{k},\bar{n}}\Big(\mathbb{R}_{\bar{n},\bar{k}}(\bar{v},\bar{u})E_{k_1}^{2,j_1}\dots E_{k_a}^{2,j_a}E_{n_1}^{3,\beta_1}\dots E_{n_b}^{3,\beta_b}\Big)|0\rangle. \tag{4.7}$$

To calculate the remaining trace we present the product of the $R$-matrices $\mathbb{R}_{\bar{n},\bar{k}}(\bar{v},\bar{u})$ as

$$\mathbb{R}_{\bar{n},\bar{k}}(\bar{v},\bar{u}) = \sum_{\bar{\lambda},\bar{\mu},\bar{p},\bar{q}=1}^{3} r^{\lambda_1\mu_1,\dots,\lambda_b\mu_b;p_1,q_1,\dots,p_a,q_a}(\bar{v},\bar{u})E_{n_1}^{\lambda_1,\mu_1}\dots E_{n_b}^{\lambda_b,\mu_b}E_{k_1}^{p_1,q_1}\dots E_{k_a}^{p_a,q_a}, \tag{4.8}$$

where $r^{\lambda_1\mu_1,\dots,p_a,q_a}(\bar{v},\bar{u})$ are numeric coefficients, and the sum is taken over all $\bar{\lambda} = \{\lambda_1,\dots,\lambda_b\}$, $\bar{\mu} = \{\mu_1,\dots,\mu_b\}$, $\bar{p} = \{p_1,\dots,p_a\}$, and $\bar{q} = \{q_1,\dots,q_a\}$. Then we obtain (see appendix B for more details)

$$\operatorname{tr}_{\bar{k},\bar{n}}\Big(\mathbb{R}_{\bar{n},\bar{k}}(\bar{v},\bar{u})E_{k_1}^{2,j_1}\dots E_{k_a}^{2,j_a}E_{n_1}^{3,\beta_1}\dots E_{n_b}^{3,\beta_b}\Big) = r^{\beta_1 3,\dots,\beta_b 3;j_1,2,\dots,j_a 2}(\bar{v},\bar{u}) \equiv r^{\bar{\beta},\bar{j}}(\bar{v},\bar{u}). \tag{4.9}$$

Now we should compute the coefficients $r^{\bar{\beta},\bar{j}}(\bar{v},\bar{u})$. For this, it is convenient to use a diagram technique [29, 69]. We present a single $R$-matrix as a vertex (see Fig. 1). Observe that $R^{\alpha\beta;ij}(v,u) \neq 0$, if either $\alpha = \beta$ and $i = j$ or $\alpha = j$ and $i = \beta$. Thus, the index, which

SciPost Phys. Lect. Notes 19 (2020)

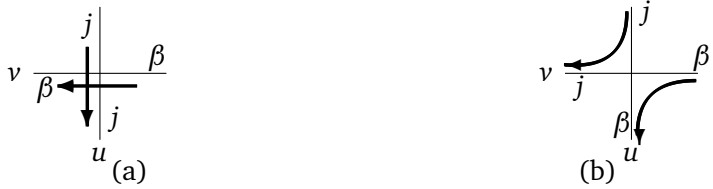

Figure 1: $R$-matrix as a vertex. Horizontal edge is associated with the parameter $v$ and the space $V_{n_p}$. Vertical edge is associated with the parameter $u$ and the space $V_{k_s}$.

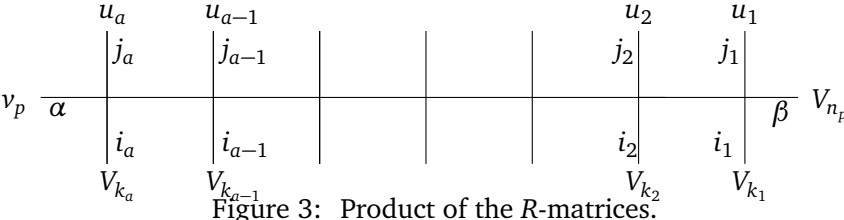

Figure 2: Two moves of the indices.

enters the vertex from the north, can go further to the south or turn to the west. Respectively, the index, which enters the vertex from the east, can go further to the west or turn to the south (see Fig. 2).

The product of the $R$-matrices

$$R_{n_p,k_a}(v_p,u_a)\ldots R_{n_p,k_1}(v_p,u_1) \tag{4.10}$$

is given by the horizontal line (see Fig. 3). Respectively, the total product $\mathbb{R}_{\bar{n},\bar{k}}(\bar{v},\bar{u})$ looks as

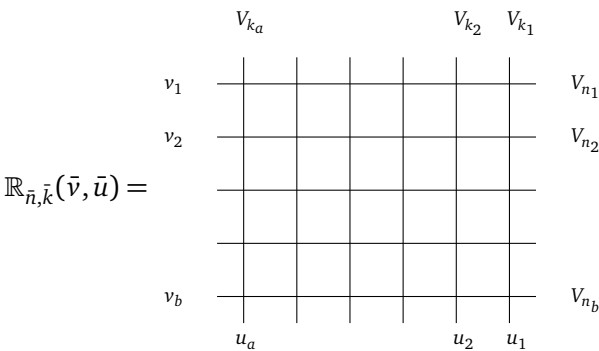

Figure 3: Product of the $R$-matrices.

it is shown on the Fig 4. Finally, the matrix element $r^{\bar{\beta},\bar{j}}(\bar{v},\bar{u})$ has a graphical representation

$$\mathbb{R}_{\bar{n},\bar{k}}(\bar{v},\bar{u}) =$$

Figure 4: Graphical interpretation of $\mathbb{R}_{\bar{n},\bar{k}}(\bar{v},\bar{u})$

shown on Fig. 5.

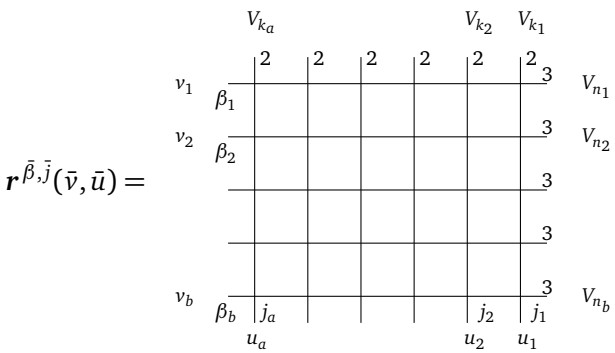

Figure 5: Graphical interpretation of the matrix element $r^{\bar{\beta},\bar{j}}(\bar{v},\bar{u})$.

Generically, the indices on the edges of the lattice on Fig. 4 can take three values: $1, 2, 3$. However, in the case of the lattice on Fig. 5 the value 1 is forbidden. Indeed, we have seen that moving through any vertex, every index goes in the direction from the north-east to the south-west. Thus, any index of an arbitrary edge has its source either on the northern or eastern

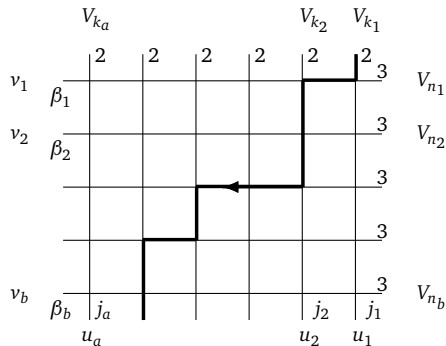

Figure 6: Line of the constant index

lattice boundary. But all the indices on those boundaries take the values 2 or 3. Thus, there is no the index 1 on the edges of the lattice on the Fig. 6.

The above consideration shows that the original $\mathfrak{gl}_3$-invariant $R$-matrix $R^{\alpha\beta;ij}(u,v)$ turns into the $\mathfrak{gl}_2$-invariant $R$-matrix $r^{\alpha\beta;ij}(u,v)$, where all the indices take values 2 and 3.

Let

$$\mathcal{T}^{(a)}(v_p|\bar{u}) = r_{n_p,k_a}(v_p,u_a)\dots r_{n_p,k_1}(v_p,u_1). \tag{4.11}$$

It is easy to see that $\mathcal{T}^{(a)}(v_p|\bar{u})$ coincides with the monodromy matrix introduced by (3.30). Then

$$r^{\bar{\beta},\bar{j}}(\bar{v},\bar{u}) = \prod_{p=1}^{b} \mathcal{T}^{(a)}_{\beta_p,3}(v_p|\bar{u}), \tag{4.12}$$

and we obtain

$$|\Psi_{a,b}(\bar{u};\bar{v})\rangle = \sum_{\bar{j}=2}^{3}\sum_{\bar{\beta}=2}^{3} T_{1,j_1}(u_1)\dots T_{1,j_a}(u_a)T_{2,\beta_1}(v_1)\dots T_{2,\beta_b}(v_b)\prod_{p=1}^{b} \mathcal{T}^{(a)}_{\beta_p,3}(v_p|\bar{u})|0\rangle. \tag{4.13}$$

Observe that here we have changed the summation limits for $\bar{j}$ and $\bar{\beta}$. When taking the sum over $\bar{j}$ one should remember that the monodromy matrix $\mathcal{T}^{(a)}_{\beta_p,3}(v_p|\bar{u})$ acts in the tensor product of $V_{k_s}$, $s = 1,\dots,a$, where it has indices $j_s$. These indices are not shown explicitly in (4.13).

If we introduce

$$\mathbb{D}(v) = \begin{pmatrix} T_{22}(v) & T_{23}(v) \\ T_{32}(v) & T_{33}(v) \end{pmatrix}, \tag{4.14}$$

and

$$\widehat{\mathcal{T}}^{(a)}(v_p|\bar{u}) = \mathbb{D}(v_p)\mathcal{T}^{(a)}(v_p|\bar{u}), \tag{4.15}$$

then (4.13) takes the form

$$|\Psi_{a,b}(\bar{u};\bar{v})\rangle = \sum_{\bar{j}=2}^{3} T_{1,j_1}(u_1)\dots T_{1,j_a}(u_a)\prod_{p=1}^{b}\widehat{\mathcal{T}}_{2,3}^{(a)}(v_p|\bar{u})|0\rangle. \tag{4.16}$$

Now it becomes obvious that this formula coincides with (3.62), where $\mathbb{F}(\bar{u};\bar{v})|0\rangle$ is given by (3.67).

Concluding this section, we note that the trace formula has a fairly obvious generalization to the case of models with $\mathfrak{gl}_N$-invariant $R$-matrix. We do not give the explicit formula, since this would require the introduction of a large number of new notations. The reader, however, can find this formula in [11].

## 4.2 Symmetry over $\bar{u}$

Trace formula (4.4) allows us to prove the long standing problem of the symmetry of the Bethe vectors over the set $\bar{u}$. For this we first consider some properties of the matrix $R(u,v)P$:

$$R(u,v)P = \sum_{a,b=1}^{3} E^{a,b} \otimes E^{b,a} + g(u,v)\mathbb{I}. \tag{4.17}$$

Let $j < 3$. Then

$$f(u,v)E^{j+1,j} \otimes E^{j+1,j} = R(u,v)PE^{j+1,j} \otimes E^{j+1,j} = E^{j+1,j} \otimes E^{j+1,j}R(u,v)P. \tag{4.18}$$

Indeed, using (4.17) we have, for example,

$$E^{j+1,j} \otimes E^{j+1,j}R(u,v)P = \sum_{a,b=1}^{3} (E^{j+1,j} \otimes E^{j+1,j})(E^{a,b} \otimes E^{b,a}) + g(u,v)E^{j+1,j} \otimes E^{j+1,j}$$

$$= g(u,v)E^{j+1,j} \otimes E^{j+1,j} + \sum_{a,b=1}^{3} E^{j+1,b} \otimes E^{j+1,a}\delta_{ja}\delta_{jb} = f(u,v)E^{j+1,j} \otimes E^{j+1,j}. \tag{4.19}$$

Consider now the right action of the matrix $R_{2,1}(u_2,u_1)P_{2,1}$ on the product of the $R$-matrices $R_{n,2}(v,u_2)R_{n,1}(v,u_1)$, where the subscript $n$ refers to some space $V_n$ which is different from $V_1$ and $V_2$. Due to the Yang–Baxter equation we have

$$R_{n,2}(v,u_2)R_{n,1}(v,u_1)\,R_{2,1}(u_2,u_1)P_{2,1} = R_{2,1}(u_2,u_1)\,R_{n,1}(v,u_1)R_{n,2}(v,u_2)\,P_{2,1}. \tag{4.20}$$

Then moving the permutation matrix to the left we exchange the spaces $V_1$ and $V_2$:

$$R_{n,2}(v,u_2)R_{n,1}(v,u_1)\,R_{2,1}(u_2,u_1)P_{2,1} = R_{2,1}(u_2,u_1)P_{2,1}\,R_{n,2}(v,u_1)R_{n,1}(v,u_2). \tag{4.21}$$

Thus, acting form the right on $R_{n,2}(v,u_2)R_{n,1}(v,u_1)$, the matrix $R_{2,1}(u_2,u_1)P_{2,1}$ in fact makes the replacement $u_1 \leftrightarrow u_2$.

Similarly $R_{2,1}(u_2, u_1)P_{2,1}$ acts on the product $T_1(u_1)T_2(u_2)$. For this we first write down the $RTT$-relation

$$R_{1,2}(u_1, u_2)T_1(u_1)T_2(u_2) = T_2(u_2)T_1(u_1)R_{1,2}(u_1, u_2), \tag{4.22}$$

and multiply it from both sides by $R_{2,1}(u_2, u_1)$:

$$T_1(u_1)T_2(u_2)R_{2,1}(u_2, u_1) = R_{2,1}(u_2, u_1)T_2(u_2)T_1(u_1). \tag{4.23}$$

Here we used the fact that $R_{1,2}(u_1, u_2)R_{2,1}(u_2, u_1) = f(u_1, u_2)f(u_2, u_1)\mathbb{I}$.

Consider now how the permutation matrix acts on $T_1(u_1)T_2(u_2)$. We have

$$T_1(u_1)T_2(u_2) = \sum_{i,j,k,l} T_{ij}(u_1)T_{kl}(u_2)E_1^{ij}E_2^{kl}. \tag{4.24}$$

Then

$$
\begin{aligned}
P_{1,2}T_1(u_1)T_2(u_2)P_{1,2} &= \sum_{a,b,c,d}\sum_{i,j,k,l} T_{ij}(u_1)T_{kl}(u_2)E_1^{ab}E_2^{ba}E_1^{ij}E_2^{kl}E_1^{cd}E_2^{dc} \\
&= \sum_{a,b,c,d}\sum_{i,j,k,l} T_{ij}(u_1)T_{kl}(u_2)E_1^{ab}E_1^{id}\delta_{jc}E_2^{ba}E_2^{kc}\delta_{ld} \\
&= \sum_{a,b,c,d}\sum_{i,j,k,l} T_{ij}(u_1)T_{kl}(u_2)E_1^{ad}\delta_{bi}\delta_{jc}E_2^{bc}\delta_{ak}\delta_{ld} \\
&= \sum_{i,j,k,l} T_{ij}(u_1)T_{kl}(u_2)E_1^{kl}E_2^{ij} = T_2(u_1)T_1(u_2). \tag{4.25}
\end{aligned}
$$

Thus, using (4.23) and (4.25) we obtain

$$T_1(u_1)T_2(u_2)\,R_{2,1}(u_2, u_1)P_{1,2} = R_{2,1}(u_2, u_1)\,T_2(u_2)T_1(u_1)\,P_{1,2} = R_{2,1}(u_2, u_1)P_{1,2}\,T_1(u_2)T_2(u_1). \tag{4.26}$$

Hence, here we also deal with the replacement $u_1 \leftrightarrow u_2$.

Now everything is ready for the proof of the symmetry of the Bethe vector $|\Psi_{a,b}(\bar{u}; \bar{v})\rangle$ over the parameters $\bar{u}$. Consider a vector

$$f(u_{i+1}, u_i)|\Psi_{a,b}(\bar{u}; \bar{v})\rangle = f(u_{i+1}, u_i)\,\mathrm{tr}_{\bar{k}, \bar{n}}\Big(\mathbb{T}_{\bar{k}}(\bar{u})\mathbb{T}_{\bar{n}}(\bar{v})\mathbb{R}_{\bar{n}, \bar{k}}(\bar{v}, \bar{u})E_{k_1}^{21}\ldots E_{k_a}^{21}E_{n_1}^{32}\ldots E_{n_b}^{32}\Big)|0\rangle, \tag{4.27}$$

for some $i = 1, \ldots, a-1$. Due to (4.18) we have

$$
\begin{aligned}
f(u_{i+1}, u_i)|\Psi_{a,b}(\bar{u}; \bar{v})\rangle = \mathrm{tr}_{\bar{k}, \bar{n}}\Big(&\mathbb{T}_{\bar{k}}(\bar{u})\mathbb{T}_{\bar{n}}(\bar{v})\mathbb{R}_{\bar{n}, \bar{k}}(\bar{v}, \bar{u})E_{k_1}^{21}\ldots E_{k_a}^{21}E_{n_1}^{32}\ldots E_{n_b}^{32} \\
&\times R_{k_{i+1}, k_i}(u_{i+1}, u_i)P_{k_{i+1}, k_i}\Big)|0\rangle. \tag{4.28}
\end{aligned}
$$

The matrix $R_{k_{i+1}, k_i}(u_{i+1}, u_i)P_{k_{i+1}, k_i}$ first can be moved to the left through the products of the matrices $E_{n_1}^{32}\ldots E_{n_b}^{32}$ and $E_{k_1}^{21}\ldots E_{k_a}^{21}$. Then, moving through the $R$-matrices $\mathbb{R}_{\bar{n}, \bar{k}}$ we should exchange it with the combinations $R_{n_s, k_{i+1}}(v_{n_s}, u_{i+1})R_{n_s, k_i}(v_{n_s}, u_i)$. Due to (4.21) this leads to the replacement $u_i \leftrightarrow u_{i+1}$ in $\mathbb{R}_{\bar{n}, \bar{k}}$. After this we should move $R_{k_{i+1}, k_i}(u_{i+1}, u_i)P_{k_{i+1}, k_i}$ through the product of the $T$-matrices $\mathbb{T}_{\bar{k}}(\bar{u})$. Here we meat a combination $T_{k_i}(u_i)T_{k_{i+1}}(u_{i+1})$. And again we obtain the replacement $u_i \leftrightarrow u_{i+1}$ in $\mathbb{T}_{\bar{k}}(\bar{u})$. Thus, we arrive at

$$
\begin{aligned}
f(u_{i+1}, u_i)|\Psi_{a,b}(\bar{u}; \bar{v})\rangle = \mathrm{tr}_{\bar{k}, \bar{n}}\Big(&R_{k_{i+1}, k_i}(u_{i+1}, u_i)P_{k_{i+1}, k_i}\mathbb{T}_{\bar{k}}(\bar{u})\Big|_{u_i \leftrightarrow u_{i+1}} \\
&\times \mathbb{T}_{\bar{n}}(\bar{v})\mathbb{R}_{\bar{n}, \bar{k}}(\bar{v}, \bar{u})\Big|_{u_i \leftrightarrow u_{i+1}} E_{k_1}^{21}\ldots E_{k_a}^{21}E_{n_1}^{32}\ldots E_{n_b}^{32}\Big)|0\rangle. \tag{4.29}
\end{aligned}
$$

Using cyclicity of the trace we move $R_{k_{i+1},k_i}(u_{i+1},u_i)P_{k_{i+1},k_i}$ to its original position

$$
\begin{aligned}
f(u_{i+1},u_i)|\Psi_{a,b}(\bar{u};\bar{v})\rangle &= \mathrm{tr}_{\bar{k},\bar{n}}\Big(\mathbb{T}_{\bar{k}}(\bar{u})\Big|_{u_i\leftrightarrow u_{i+1}}\mathbb{T}_{\bar{n}}(\bar{v})\mathbb{R}_{\bar{n},\bar{k}}(\bar{v},\bar{u})\Big|_{u_i\leftrightarrow u_{i+1}} \\
&\quad \times E^{21}_{k_1}\dots E^{21}_{k_a}E^{32}_{n_1}\dots E^{32}_{n_b}R_{k_{i+1},k_i}(u_{i+1},u_i)P_{k_{i+1},k_i}\Big)|0\rangle \\
&= f(u_{i+1},u_i)\,\mathrm{tr}_{\bar{k},\bar{n}}\Big(\mathbb{T}_{\bar{k}}(\bar{u})\Big|_{u_i\leftrightarrow u_{i+1}}\mathbb{T}_{\bar{n}}(\bar{v})\mathbb{R}_{\bar{n},\bar{k}}(\bar{v},\bar{u})\Big|_{u_i\leftrightarrow u_{i+1}}E^{21}_{k_1}\dots E^{21}_{k_a}E^{32}_{n_1}\dots E^{32}_{n_b}\Big)|0\rangle. \quad (4.30)
\end{aligned}
$$

Thus, we obtain

$$
|\Psi_{a,b}(\bar{u};\bar{v})\rangle = |\Psi_{a,b}(\bar{u};\bar{v})\rangle\Big|_{u_i\leftrightarrow u_{i+1}}. \tag{4.31}
$$

Formally, one can prove the symmetry over the set $\bar{v}$ using exactly the same way. However, this symmetry is obvious in the representations (3.62), (3.67).

# 5 Recursion for the Bethe vectors

If we restrict ourselves to the problem of the spectrum of the transfer matrix (and hence, the Hamiltonian and other integrals of motion), then this problem is solved. We constructed eigenvectors of the transfer matrix and found the corresponding eigenvalues. However, if we expect to use the NABA for calculating correlation functions, the results obtained are still insufficient. As we have already mentioned, in this case, one has to calculate scalar products containing off-shell Bethe vectors. The representation (3.78) and the trace formula (4.4) are not very convenient for these purposes. Therefore, we need to get other representations for Bethe vectors, with a view to their further application to the calculation of scalar products. One of the steps on this way is recursive formulas. These formulas, in particular, make it possible to prove by induction many statements concerning scalar products.

In this section we derive a relation between Bethe vectors $|\Psi_{a,b}\rangle$, $|\Psi_{a-1,b}\rangle$, and $|\Psi_{a-1,b-1}\rangle$. To do this, we will use some formulas for the composite model in the $\mathcal{R}_2$ algebra. Therefore, we begin this section with a brief description of the composite model. More details can be found in [26, 28, 29, 70].

## 5.1 Composite model in $\mathcal{R}_2$ algebra

Let we have two $2\times 2$ monodromy matrices $T^{(1)}(v)$ and $T^{(2)}(v)$:

$$
T^{(j)}(v)=\begin{pmatrix} A^{(j)}(v) & B^{(j)}(v) \\ C^{(j)}(v) & D^{(j)}(v) \end{pmatrix}, \qquad j=1,2. \tag{5.1}
$$

We assume that both of them satisfy the $RTT$-relation (1.1) with the $R$-matrix (1.12). We also assume the entries of $T^{(1)}(v)$ and $T^{(2)}(v)$ act in different Hilbert spaces and each of these matrices possesses a vacuum vector $|0\rangle^{(j)}$ with the standard properties

$$
A^{(j)}(v)|0\rangle^{(j)}=a^{(j)}(v)|0\rangle^{(j)}, \qquad D^{(j)}(v)|0\rangle^{(j)}=d^{(j)}(v)|0\rangle^{(j)}, \qquad C^{(j)}(v)|0\rangle^{(j)}=0, \tag{5.2}
$$

where $a^{(j)}(v)$ and $d^{(j)}(v)$ are some complex valued functions. Then we can define off-shell Bethe vectors

$$
B^{(j)}(\bar{v})|0\rangle^{(j)}=B^{(j)}(v_1)\dots B^{(j)}(v_n)|0\rangle^{(j)}, \tag{5.3}
$$

for each $T^{(j)}(v)$. Here we used the shorthand notation for the product of the operators $B^{(j)}$.

Obviously, under these conditions, a matrix $T(v)$

$$T(v) = T^{(2)}(v)T^{(1)}(v) = \begin{pmatrix} A(v) & B(v) \\ C(v) & D(v) \end{pmatrix} \tag{5.4}$$

satisfies the $RTT$-relation and has a vacuum vector $|0\rangle = |0\rangle^{(2)} \otimes |0\rangle^{(1)}$. We also can define off-shell Bethe vectors corresponding to the matrix $T(v)$:

$$B(\bar{v})|0\rangle = B(v_1)\ldots B(v_n)|0\rangle. \tag{5.5}$$

A model in which the monodromy matrix is defined in the form (5.4) is called composite. The matrix $T(u)$ is called a full monodromy matrix, and the matrices $T^{(j)}(v)$ are called partial monodromy matrices. Similarly, vectors (5.5) are called full Bethe vectors, and vectors (5.3) are called partial.

In fact, we have already dealt with the composite model in section 3. Indeed, the matrix $\widehat{\mathcal{T}}^{(a)}(z)$ (3.30) is the product of two partial monodromy matrices $\mathbb{D}(z)$ and $\mathcal{T}_0^{(a)}(z)$. However, so far we have needed fairly simple statements about the composite model. For example, we used the fact that the functions $a(v)$ and $d(v)$ of the full matrix $T(v)$ are respectively products of the functions $a^{(j)}(v)$ and $d^{(j)}(v)$. This statement immediately follows from representations

$$\begin{aligned} A(v) &= A^{(2)}(v)A^{(1)}(v) + B^{(2)}(v)C^{(1)}(v), \\ D(v) &= D^{(2)}(v)D^{(1)}(v) + C^{(2)}(v)B^{(1)}(v). \end{aligned} \tag{5.6}$$

Now we need to express the full Bethe vectors in terms of the partial ones. Using

$$B(v) = A^{(2)}(v)B^{(1)}(v) + B^{(2)}(v)D^{(1)}(v), \tag{5.7}$$

we easily find

$$B(v)|0\rangle = a^{(2)}(v)|0\rangle^{(2)} \otimes B^{(1)}(v)|0\rangle^{(1)} + d^{(1)}(v)B^{(2)}(v)|0\rangle^{(2)} \otimes |0\rangle^{(1)}. \tag{5.8}$$

However, if the cardinality of the set $\bar{v}$ is more than 1, then the problem to express $B(\bar{v})|0\rangle$ in terms of $B^{(j)}(\bar{v})|0\rangle^{(j)}$ becomes more sophisticated. It was solved in [70]:

$$B(\bar{v})|0\rangle = \sum_{\bar{v} \mapsto \{\bar{v}_{\text{I}}, \bar{v}_{\text{II}}\}} a^{(2)}(\bar{v}_{\text{I}})d^{(1)}(\bar{v}_{\text{II}})B^{(2)}(\bar{v}_{\text{II}})|0\rangle^{(2)} \otimes B^{(1)}(\bar{v}_{\text{I}})|0\rangle^{(1)} \cdot f(\bar{v}_{\text{II}}, \bar{v}_{\text{I}}). \tag{5.9}$$

Here the sum is taken over all possible partitions of the set $\bar{v}$ into two subsets $\bar{v}_{\text{I}}$ and $\bar{v}_{\text{II}}$. We have also extended our convention on the shorthand notation to the products of the functions $a^{(j)}$ and $d^{(j)}$. For example, $d^{(1)}(\bar{v}_{\text{II}})$ means the product of $d^{(1)}(v_i)$ over $v_i \in \bar{v}_{\text{II}}$. A detailed proof of (5.9) can be found in [29].

## 5.2 First recursion for Bethe vectors

We now proceed to the derivation of recursion for Bethe vectors in the $\mathcal{R}_3$ algebra. For this, we use representation (3.62)

$$|\Psi_{a,b}(\bar{u}; \bar{v})\rangle = \mathbb{B}_1(u_1)\mathbb{B}_2(u_2)\ldots\mathbb{B}_a(u_a)\mathbb{F}^{(a)}(\bar{u}; \bar{v})|0\rangle, \tag{5.10}$$

where we equipped the vector $\mathbb{F}^{(a)}$ with the additional superscript $(a)$. This superscript stresses that $\mathbb{F}^{(a)}|0\rangle$ belongs to the space $\mathcal{H} \otimes \mathcal{H}^{(a)}$, where $\mathcal{H}^{(a)}$ is the tensor product of $a$ spaces $\mathbb{C}^2$

(see (3.32)). Our goal is to express this vector in terms of vectors $\mathbb{F}^{(a-1)}|0\rangle$, which belong to the space $\mathcal{H} \otimes \mathcal{H}^{(a-1)}$, where $\mathcal{H}^{(a-1)}$ is the tensor product of $a-1$ spaces $\mathbb{C}^2$. This is a typical problem of a composite model [26, 28, 29, 70].

Recall that representation (5.10) is equivalent to the following sum:

$$|\Psi_{a,b}(\bar{u};\bar{v})\rangle = \sum_{\beta_1,\ldots,\beta_a} B_{\beta_1}(u_1)B_{\beta_2}(u_2)\ldots B_{\beta_a}(u_a)F^{(a)}_{\beta_1,\ldots,\beta_a}(\bar{u};\bar{v})|0\rangle, \tag{5.11}$$

where every $\beta_i \in \{\beta_1,\ldots,\beta_a\}$ takes two values $\beta_i = 1, 2$. Let us write explicitly the sum over $\beta_1$:

$$|\Psi_{a,b}(\bar{u};\bar{v})\rangle = B_1(u_1) \sum_{\beta_2,\ldots,\beta_a} B_{\beta_2}(u_2)\ldots B_{\beta_a}(u_a)F^{(a)}_{1,\beta_2,\ldots,\beta_a}(\bar{u};\bar{v})|0\rangle$$
$$+ B_2(u_1) \sum_{\beta_2,\ldots,\beta_a} B_{\beta_2}(u_2)\ldots B_{\beta_a}(u_a)F^{(a)}_{2,\beta_2,\ldots,\beta_a}(\bar{u};\bar{v})|0\rangle, \tag{5.12}$$

where the remaining sums are taken over $\{\beta_2,\ldots,\beta_a\}$. Thus, we should find explicit representations for the components $F^{(a)}_{1,\beta_2,\ldots,\beta_a}(\bar{u};\bar{v})|0\rangle$ and $F^{(a)}_{2,\beta_2,\ldots,\beta_a}(\bar{u};\bar{v})|0\rangle$ in terms of vectors belonging to the space of lower dimension.

We know that the vector $\mathbb{F}^{(a)}|0\rangle$ has the form (3.67), where the operator $\widehat{\mathcal{B}}^{(a)}$ is the matrix element of the monodromy matrix $\widehat{\mathcal{T}}^{(a)}$ (3.65). We can present this monodromy matrix as follows:

$$\widehat{\mathcal{T}}^{(a)}(z) = \widehat{\mathcal{T}}^{(a-1)}(z)r_{01}(z,u_1), \tag{5.13}$$

where

$$\widehat{\mathcal{T}}^{(a-1)}_0(z) = \mathbb{D}_0(z)r_{0a}(z,u_a)\ldots r_{02}(z,u_2) = \begin{pmatrix} \widehat{\mathcal{A}}^{(a-1)}(z) & \widehat{\mathcal{B}}^{(a-1)}(z) \\ \widehat{\mathcal{C}}^{(a-1)}(z) & \widehat{\mathcal{D}}^{(a-1)}(z) \end{pmatrix}_0. \tag{5.14}$$

The entries of $\widehat{\mathcal{T}}^{(a-1)}_{ij}(z)$ act in the space $\mathcal{H} \otimes \mathcal{H}^{(a-1)}$. The space $\mathcal{H}^{(a-1)}$ has a natural vacuum $|\tilde{0}\rangle = |0\rangle \otimes |\Omega^{(a-1)}\rangle$, where

$$|\Omega^{(a-1)}\rangle = \underbrace{\left(\begin{smallmatrix}1\\0\end{smallmatrix}\right) \otimes \cdots \otimes \left(\begin{smallmatrix}1\\0\end{smallmatrix}\right)}_{a-1 \quad \text{times}}. \tag{5.15}$$

It is easy to see that

$$\widehat{\mathcal{A}}^{(a-1)}(z)|\tilde{0}\rangle = \lambda_2(z)f(z,\bar{u}_1)|\tilde{0}\rangle,$$
$$\widehat{\mathcal{D}}^{(a-1)}(z)|\tilde{0}\rangle = \lambda_3(z)|\tilde{0}\rangle, \tag{5.16}$$

and we recall that $\bar{u}_1 = \bar{u} \setminus u_1$.

In its turn, the matrix $r_{01}(z,u_1)$ can be considered as the monodromy matrix of the $XXX$ chain consisting of one site. We dealt already with this monodromy matrix in section 3.6 (see (3.83)). Recall that the auxiliary space of this matrix is $V_0 \sim \mathbb{C}^2$, the quantum space is $V_1 \sim \mathbb{C}^2$. It can be presented as a $2 \times 2$ matrix in the space $V_0$

$$r_{01}(z,u_1) = \begin{pmatrix} a(z) & b(z) \\ c(z) & d(z) \end{pmatrix} = \begin{pmatrix} 1 & 0 \\ 0 & 1 \end{pmatrix} + g(z,u_1)\begin{pmatrix} E^{11}_1 & E^{21}_1 \\ E^{12}_1 & E^{22}_1 \end{pmatrix}, \tag{5.17}$$

where the entries act in the space $V_1 \sim \mathbb{C}^2$ with the vacuum vector $\left(\begin{smallmatrix}1\\0\end{smallmatrix}\right)$. Obviously

$$a(z)\left(\begin{smallmatrix}1\\0\end{smallmatrix}\right) = f(z,u_1)\left(\begin{smallmatrix}1\\0\end{smallmatrix}\right),$$
$$d(z)\left(\begin{smallmatrix}1\\0\end{smallmatrix}\right) = \left(\begin{smallmatrix}1\\0\end{smallmatrix}\right). \tag{5.18}$$

A peculiarity of this monodromy matrix is that $\boldsymbol{b}^n(z) = 0$ for $n > 1$, because $\boldsymbol{b}(z) = g(z, u_1)E_1^{21}$.

We can treat $\widehat{\mathcal{T}}^{(a)}(z)$ as the monodromy matrix of the composite model with $T^{(2)}(z) = \widehat{\mathcal{T}}^{(a-1)}(z)$ and $T^{(1)}(z) = r_{01}(z, u_1)$. In this model

$$a^{(2)}(z) = \lambda_2(z)f(z, \bar{u}_1), \qquad d^{(1)}(z) = 1, \tag{5.19}$$

due to (5.16) and (5.18). Hence, in the case under consideration equation (5.9) takes the form

$$\widehat{\mathcal{B}}^{(a)}(\bar{v})|0\rangle = \sum_{\bar{v} \mapsto \{\bar{v}_{\mathrm{I}}, \bar{v}_{\mathrm{II}}\}} \lambda_2(\bar{v}_{\mathrm{I}})f(\bar{v}_{\mathrm{I}}, \bar{u}_1)f(\bar{v}_{\mathrm{II}}, \bar{v}_{\mathrm{I}})\widehat{\mathcal{B}}^{(a-1)}(\bar{v}_{\mathrm{II}})|\tilde{0}\rangle \otimes \boldsymbol{b}(\bar{v}_{\mathrm{I}})\left(\begin{smallmatrix}1\\0\end{smallmatrix}\right). \tag{5.20}$$

Here we have extended the convention on the shorthand notation (2.22) to the products of the commuting operators $\widehat{\mathcal{B}}^{(a)}(\bar{v})$, $\widehat{\mathcal{B}}^{(a-1)}(\bar{v})$, and $\boldsymbol{b}(\bar{v})$.

Formally, the sum in (5.20) is taken over all possible partitions of the set $\bar{v}$ into two disjoint subsets $\bar{v}_{\mathrm{I}}$ and $\bar{v}_{\mathrm{II}}$. However, due to the property $\boldsymbol{b}^n(z) = 0$ for $n > 1$ we conclude that either $\#\bar{v}_{\mathrm{I}} = 0$ or $\#\bar{v}_{\mathrm{I}} = 1$. In the first case we have $\bar{v}_{\mathrm{I}} = \emptyset$ and $\bar{v}_{\mathrm{II}} = \bar{v}$. In the second case we can set $\bar{v}_{\mathrm{I}} = v_\ell$ and $\bar{v}_{\mathrm{II}} = \bar{v}_\ell$, where $\ell = 1, \ldots, b$. Thus, we find

$$\widehat{\mathcal{B}}^{(a)}(\bar{v})|0\rangle = \widehat{\mathcal{B}}^{(a-1)}(\bar{v})|\tilde{0}\rangle \otimes \left(\begin{smallmatrix}1\\0\end{smallmatrix}\right) + \sum_{\ell=1}^{b} \lambda_2(v_\ell)g(v_\ell, u_1)f(v_\ell, \bar{u}_1)f(\bar{v}_\ell, v_\ell)\widehat{\mathcal{B}}^{(a-1)}(\bar{v}_\ell)|\tilde{0}\rangle \otimes \left(\begin{smallmatrix}0\\1\end{smallmatrix}\right), \tag{5.21}$$

where the first term corresponds to $\bar{v}_{\mathrm{I}} = \emptyset$ and the sum over $\ell$ corresponds to the partitions $\bar{v}_{\mathrm{I}} = v_\ell$. From this equation we find the components $F^{(a)}_{1,\beta_2,\ldots,\beta_a}$ and $F^{(a)}_{2,\beta_2,\ldots,\beta_a}$ in terms of components $F^{(a-1)}_{\beta_2,\ldots,\beta_a}$:

$$F^{(a)}_{1,\beta_2,\ldots,\beta_a}(\bar{u}; \bar{v})|0\rangle = F^{(a-1)}_{\beta_2,\ldots,\beta_a}(\bar{u}_1; \bar{v})|\tilde{0}\rangle,$$

$$F^{(a)}_{2,\beta_2,\ldots,\beta_a}(\bar{u}; \bar{v})|0\rangle = \sum_{\ell=1}^{b} \lambda_2(v_\ell)g(v_\ell, u_1)f(v_\ell, \bar{u}_1)f(\bar{v}_\ell, v_\ell)F^{(a-1)}_{\beta_2,\ldots,\beta_a}(\bar{u}_1; \bar{v}_\ell)|\tilde{0}\rangle. \tag{5.22}$$

Substituting this into (5.12) we arrive at

$$|\Psi_{a,b}(\bar{u}; \bar{v})\rangle = B_1(u_1) \sum_{\beta_2,\ldots,\beta_a} B_{\beta_2}(u_2)\ldots B_{\beta_a}(u_a)F^{(a-1)}_{\beta_2,\ldots,\beta_a}(\bar{u}_1; \bar{v})|\tilde{0}\rangle$$

$$+ B_2(u_1) \sum_{\ell=1}^{b} \lambda_2(v_\ell)g(v_\ell, u_1)f(v_\ell, \bar{u}_1)f(\bar{v}_\ell, v_\ell) \sum_{\beta_2,\ldots,\beta_a} B_{\beta_2}(u_2)\ldots B_{\beta_a}(u_a)F^{(a-1)}_{\beta_2,\ldots,\beta_a}(\bar{u}_1; \bar{v}_\ell)|\tilde{0}\rangle. \tag{5.23}$$

Then we recognise the Bethe vector $|\Psi_{a-1,b}(\bar{u}_1; \bar{v})\rangle$ in the first line of (5.23) and the sum of the Bethe vectors $|\Psi_{a-1,b-1}(\bar{u}_1; \bar{v}_\ell)\rangle$ in the second line:

$$|\Psi_{a,b}(\bar{u}; \bar{v})\rangle = B_1(u_1)|\Psi_{a-1,b}(\bar{u}_1; \bar{v})\rangle$$

$$+ B_2(u_1) \sum_{\ell=1}^{b} \lambda_2(v_\ell)g(v_\ell, u_1)f(v_\ell, \bar{u}_1)f(\bar{v}_\ell, v_\ell)|\Psi_{a-1,b-1}(\bar{u}_1; \bar{v}_\ell)\rangle. \tag{5.24}$$

Since $B_1(u) = T_{12}(u)$ and $B_2(u) = T_{13}(u)$, we recast (5.24) as follows:

$$|\Psi_{a,b}(\bar{u}; \bar{v})\rangle = T_{12}(u_1)|\Psi_{a-1,b}(\bar{u}_1; \bar{v})\rangle$$

$$+ T_{13}(u_1) \sum_{\ell=1}^{b} \lambda_2(v_\ell)g(v_\ell, u_1)f(v_\ell, \bar{u}_1)f(\bar{v}_\ell, v_\ell)|\Psi_{a-1,b-1}(\bar{u}_1; \bar{v}_\ell)\rangle. \tag{5.25}$$

Recursion (5.25) allows one to build the Bethe vectors successively, starting from the case $a = 0$. Indeed, for $a = 0$ we have $|\Psi_{0,b}(\emptyset; \bar{v})\rangle = T_{23}(\bar{v})|0\rangle$ (see (3.80)). Then we immediately find an explicit expression for the Bethe vector $|\Psi_{1,b}(u; \bar{v})\rangle$:

$$|\Psi_{1,b}(u; \bar{v})\rangle = T_{12}(u)T_{23}(\bar{v})|0\rangle + \sum_{\ell=1}^{b} \lambda_2(v_\ell)g(v_\ell, u)f(\bar{v}_\ell, v_\ell)T_{13}(u)T_{23}(\bar{v}_\ell)|0\rangle. \qquad (5.26)$$

To conclude this section we note that it follows from recursion (5.25) and initial condition (3.80) that the Bethe vectors are the states of fixed coloring

$$\mathrm{Col}\big(|\Psi_{a,b}(\bar{u}; \bar{v})\rangle\big) = \{a, b\}. \qquad (5.27)$$

This statement can be easily proved via induction over $a$.

## 5.3 Second recursion for Bethe vectors

Using representation for the Bethe vectors described in section 3.7 one can obtain another recursion

$$|\Psi_{a,b}(\bar{u}; \bar{v})\rangle = T_{23}(v_b)|\Psi_{a,b-1}(\bar{u}; \bar{v}_b)\rangle$$
$$+ T_{13}(v_b)\sum_{j=1}^{a} \lambda_2(u_j)g(v_b, u_j)f(\bar{v}_b, u_j)f(u_j, \bar{u}_j)|\Psi_{a-1,b-1}(\bar{u}_j; \bar{v}_b)\rangle. \qquad (5.28)$$

This recursion is derived via the composite model in the same way as recursion (5.25). We provide the readers with the opportunity to do this themselves.

Recursion (5.28) allows one to build the Bethe vectors starting from

$$|\Psi_{a,0}(\bar{u}; \emptyset)\rangle = T_{12}(\bar{u})|0\rangle. \qquad (5.29)$$

# 6 Explicit form of Bethe vector

Successive application of recursion (5.25) allows us to guess a general explicit formula for the Bethe vector. In other words, we can now represent the off-shell Bethe vector as a polynomial in the creation operators $T_{12}$, $T_{13}$, and $T_{23}$ applied to the vacuum vector $0\rangle$ and explicitly specify the coefficients of this polynomial. The proof of this formula relies on the recursion (5.25), but it is rather long, therefore, we do not give it here. Nevertheless, we consider it necessary to give this explicit representation, since it plays an important role in calculating scalar products.

In the explicit formula for the Bethe vector, a partition function of the six-vertex model with domain wall boundary conditions (DWPF) appears. Therefore, we give its brief description (see [26, 28, 29, 71] for more details).

## 6.1 Partition function with domain wall boundary conditions

We denote the DWPF by $K_n(\bar{v}|\bar{u})$. It depends on two sets of variables $\bar{v}$ and $\bar{u}$; the subscript indicates that $\#\bar{v} = \#\bar{u} = n$. The function $K_n$ has the following determinant representation

$$K_n(\bar{v}|\bar{u}) = \Delta'_n(\bar{v})\Delta_n(\bar{u}) \frac{f(\bar{v}, \bar{u})}{g(\bar{v}, \bar{u})} \det_n\left(\frac{g^2(v_j, u_k)}{f(v_j, u_k)}\right), \qquad (6.1)$$

where $\Delta'_n(\bar{v})$ and $\Delta_n(\bar{u})$ are

$$\Delta'_n(\bar{v}) = \prod_{j<k}^n g(v_j, v_k), \qquad \Delta_n(\bar{u}) = \prod_{j>k}^n g(u_j, u_k). \tag{6.2}$$

More explicitly

$$K_n(\bar{v}|\bar{u}) = \frac{\prod_{j,k=1}^n (v_j - u_k + c)}{\prod_{j<k}^n (v_j - v_k)(u_k - u_j)} \det_n \left( \frac{c}{(v_j - u_k)(v_j - u_k + c)} \right). \tag{6.3}$$

Obviously, $K_n(\bar{v}|\bar{u})$ is a symmetric function of $\bar{v}$ and a symmetric function of $\bar{u}$. It decreases as $1/v_1$ (resp. as $1/u_1$), if $v_1 \to \infty$ (resp. $u_1 \to \infty$) and all other variables are fixed. It has simple poles at $v_j = u_k$, $j, k = 1, \dots, n$. The residues in the poles are proportional to $K_{n-1}$:

$$K_n(\bar{v}|\bar{u})\Big|_{u_n \to v_n} = g(v_n, u_n) f(\bar{v}_n, v_n) f(v_n, \bar{u}_n) K_{n-1}(\bar{v}_n|\bar{u}_n) + reg, \tag{6.4}$$

where $reg$ stays for the terms which remain regular at $u_n \to v_n$. To prove (6.4) it is enough to observe that the determinant in (6.3) becomes singular at $v_n \to u_n$ due to the pole of the matrix element at the intersection of the $n$th row and the $n$th column. Then the determinant (6.3) reduces to the product of this matrix element and the corresponding minor, leading eventually to (6.4).

Using this property one can decompose $K_n(\bar{v}|\bar{u})$ over the poles at $u_n = v_i$, $i = 1, \dots, n$, as follows:

$$K_n(\bar{v}|\bar{u}) = \sum_{i=1}^n g(v_i, u_n) f(\bar{v}_i, v_i) f(v_i, \bar{u}_n) K_{n-1}(\bar{v}_i|\bar{u}_n). \tag{6.5}$$

In particular,

$$K_2(\bar{v}|\bar{u}) = g(v_1, u_2) g(v_2, u_1) f(v_2, v_1) f(v_1, u_1) + g(v_2, u_2) g(v_1, u_1) f(v_1, v_2) f(v_2, u_1), \tag{6.6}$$

where we used

$$K_1(v|u) = g(v, u). \tag{6.7}$$

## 6.2 Bethe vector as a sum over partitions

**Proposition 6.1.** *[42] Bethe vectors of the $\mathcal{R}_3$ algebra have the following form:*

$$|\Psi_{a,b}(\bar{u}; \bar{v})\rangle = \sum_{\substack{\bar{u} \mapsto \{\bar{u}_I, \bar{u}_{II}\} \\ \bar{v} \mapsto \{\bar{v}_I, \bar{v}_{II}\}}} K_n(\bar{v}_I|\bar{u}_I) f(\bar{u}_I, \bar{u}_{II}) f(\bar{v}_{II}, \bar{v}_I) T_{13}(\bar{u}_I) T_{12}(\bar{u}_{II}) T_{23}(\bar{v}_{II}) \lambda_2(\bar{v}_I) |0\rangle. \tag{6.8}$$

*Here $K_n(\bar{v}_I|\bar{u}_I)$ is the DWPF (6.1). The sum is taken over partitions $\bar{u} \mapsto \{\bar{u}_I, \bar{u}_{II}\}$ and $\bar{v} \mapsto \{\bar{v}_I, \bar{v}_{II}\}$ such that $\#\bar{u}_I = \#\bar{v}_I = n$ and $n = 0, 1, \dots, \min(a, b)$. Everywhere the convention on the shorthand notation (2.22) is used.*

*Remark.* Note that the r.h.s. of (6.8) is obviously symmetric over $\bar{u}$ and symmetric over $\bar{v}$.

The proof of proposition 6.1 can be found in [42]. Actually, it is enough to show that (6.8) satisfies recursion (5.25). However, as we have mentioned in the beginning of this section, this proof is rather bulky, and we do not give it here. Instead we illustrate representation (6.8) by several examples for $a$ small and $b$ arbitrary.

First of all, it follows form (6.8) that for $a = 0$ we have

$$|\Psi_{0,b}(\emptyset; \bar{v})\rangle = T_{23}(\bar{v})|0\rangle, \tag{6.9}$$

what coincides with (3.80).

Let $a = 1$. Then either $\#\bar{u}_I = \#\bar{v}_I = 0$ or $\#\bar{u}_I = \#\bar{v}_I = 1$. In both cases the product $f(\bar{u}_I, \bar{u}_{II})$ disappears from equation (6.8), because one of the subsets of $\bar{u}$ is empty. We obtain

$$|\Psi_{1,b}(u; \bar{v})\rangle = T_{12}(u)T_{23}(\bar{v})|0\rangle + \sum_{\bar{v} \mapsto \{\bar{v}_I, \bar{v}_{II}\}} g(\bar{v}_I, u)f(\bar{v}_{II}, \bar{v}_I)T_{13}(u)T_{23}(\bar{v}_{II})\lambda_2(\bar{v}_I)|0\rangle, \tag{6.10}$$

where we used (6.7), and the sum over partitions of the set $\bar{v}$ is taken under restriction $\#\bar{v}_I = 1$. Setting in this sum $\bar{v}_I = v_\ell$ and $\bar{v}_{II} = \bar{v}_\ell$, $\ell = 1, \ldots, b$, we immediately arrive at (5.26).

Let now $a = 2$. Then either $\#\bar{u}_I = \#\bar{v}_I = 0$, or $\#\bar{u}_I = \#\bar{v}_I = 1$, or $\#\bar{u}_I = \#\bar{v}_I = 2$. Respectively, we have three contributions to the Bethe vector (6.8):

$$|\Psi_{2,b}(\bar{u}; \bar{v})\rangle = |\Psi^{(0)}\rangle + |\Psi^{(1)}\rangle + |\Psi^{(2)}\rangle. \tag{6.11}$$

The contribution $|\Psi^{(0)}\rangle$ corresponds to the case $\#\bar{u}_I = \#\bar{v}_I = 0$. It is quite obvious that

$$|\Psi^{(0)}\rangle = T_{12}(\bar{u})T_{23}(\bar{v})|0\rangle. \tag{6.12}$$

The next contribution $|\Psi^{(1)}\rangle$ has the form

$$|\Psi^{(1)}\rangle = \sum_{\substack{\bar{u} \mapsto \{\bar{u}_I, \bar{u}_{II}\} \\ \bar{v} \mapsto \{\bar{v}_I, \bar{v}_{II}\}}} g(\bar{v}_I, \bar{u}_I)f(\bar{u}_I, \bar{u}_{II})f(\bar{v}_{II}, \bar{v}_I)T_{13}(\bar{u}_I)T_{12}(\bar{u}_{II})T_{23}(\bar{v}_{II})\lambda_2(\bar{v}_I)|0\rangle, \tag{6.13}$$

where the sum over partitions is taken under restriction $\#\bar{u}_I = \#\bar{v}_I = 1$. Taking explicitly the sum over partitions $\bar{u} \mapsto \{\bar{u}_I, \bar{u}_{II}\}$ we obtain

$$|\Psi^{(1)}\rangle = \sum_{\bar{v} \mapsto \{\bar{v}_I, \bar{v}_{II}\}} \Big( g(\bar{v}_I, u_1)f(u_1, u_2)T_{13}(u_1)T_{12}(u_2) + g(\bar{v}_I, u_2)f(u_2, u_1)T_{13}(u_2)T_{12}(u_1) \Big)$$
$$\times \lambda_2(\bar{v}_I)f(\bar{v}_{II}, \bar{v}_I)T_{23}(\bar{v}_{II})|0\rangle. \tag{6.14}$$

Finally, setting here $\bar{v}_I = v_\ell$ and $\bar{v}_{II} = \bar{v}_\ell$, $\ell = 1, \ldots, b$, we arrive at

$$|\Psi^{(1)}\rangle = \sum_{\ell=1}^{b} \Big( g(v_\ell, u_1)f(u_1, u_2)T_{13}(u_1)T_{12}(u_2) + g(v_\ell, u_2)f(u_2, u_1)T_{13}(u_2)T_{12}(u_1) \Big)$$
$$\times \lambda_2(v_\ell)f(\bar{v}_\ell, v_\ell)T_{23}(\bar{v}_\ell)|0\rangle. \tag{6.15}$$

The last contribution $|\Psi^{(2)}\rangle$ in (6.11) has the form:

$$|\Psi^{(2)}\rangle = \sum_{\bar{v} \mapsto \{\bar{v}_I, \bar{v}_{II}\}} K_2(\bar{v}_I|\bar{u})f(\bar{v}_{II}, \bar{v}_I)T_{13}(\bar{u})T_{23}(\bar{v}_{II})\lambda_2(\bar{v}_I)|0\rangle, \tag{6.16}$$

where $\#\bar{v}_I = 2$. Setting here $\bar{v}_I = \{v_k, v_\ell\}$ and $\bar{v}_{II} = \bar{v}_{k,\ell} \equiv \bar{v} \setminus \{v_k, v_\ell\}$ we find

$$|\Psi^{(2)}\rangle = \sum_{1 \leq \ell < k \leq b} \lambda_2(v_k)\lambda_2(v_\ell)K_2(\{v_k, v_\ell\}|\bar{u})f(\bar{v}_{k,\ell}, v_\ell)f(\bar{v}_{k,\ell}, v_k)T_{13}(\bar{u})T_{23}(\bar{v}_{k,\ell})|0\rangle. \tag{6.17}$$

Let us reproduce all the contributions $|\Psi^{(j)}\rangle$ for $j = 0, 1, 2$ via recursion (5.25). For $a = 2$, recursion (5.25) has the form

$$|\Psi_{2,b}(\bar{u};\bar{v})\rangle = T_{12}(u_2)|\Psi_{1,b}(u_1;\bar{v})\rangle$$
$$+ T_{13}(u_2)\sum_{\ell=1}^{b}\lambda_2(v_\ell)g(v_\ell,u_2)f(v_\ell,u_1)f(\bar{v}_\ell,v_\ell)|\Psi_{1,b-1}(u_1;\bar{v}_\ell)\rangle, \quad (6.18)$$

where we replaced $u_1 \leftrightarrow u_2$ due to the symmetry of the Bethe vector over the set $\bar{u}$. Using (5.26) for $|\Psi_{1,b}(u_1;\bar{v})\rangle$ and $|\Psi_{1,b-1}(u_1;\bar{v}_\ell)\rangle$ we find

$$|\Psi_{2,b}(\bar{u};\bar{v})\rangle = T_{12}(u_2)\Big(T_{12}(u_1)T_{23}(\bar{v})|0\rangle + \sum_{\ell=1}^{b}\lambda_2(v_\ell)g(v_\ell,u_1)f(\bar{v}_\ell,v_\ell)T_{13}(u_1)T_{23}(\bar{v}_\ell)|0\rangle\Big)$$
$$+ T_{13}(u_2)\sum_{k=1}^{b}\lambda_2(v_k)g(v_k,u_2)f(v_k,u_1)f(\bar{v}_k,v_k)\Big(T_{12}(u_1)T_{23}(\bar{v}_k)|0\rangle$$
$$+ \sum_{\substack{\ell=1\\\ell\neq k}}^{b}\lambda_2(v_\ell)g(v_\ell,u_1)f(\bar{v}_{\ell,k},v_\ell)T_{13}(u_1)T_{23}(\bar{v}_{\ell,k})|0\rangle\Big). \quad (6.19)$$

One can easily see the contribution $|\Psi^{(0)}\rangle = T_{12}(\bar{u})T_{23}(\bar{v})|0\rangle$. The contribution $|\Psi^{(2)}\rangle$ comes from the double sum

$$|\Psi^{(2)}\rangle = T_{13}(\bar{u})\sum_{k=1}^{b}\sum_{\substack{\ell=1\\\ell\neq k}}^{b}\lambda_2(v_k)\lambda_2(v_\ell)g(v_k,u_2)g(v_\ell,u_1)$$
$$\times f(v_k,u_1)f(\bar{v}_k,v_k)f(\bar{v}_{\ell,k},v_\ell)T_{23}(\bar{v}_{\ell,k})|0\rangle. \quad (6.20)$$

Indeed, using

$$\sum_{k=1}^{b}\sum_{\substack{\ell=1\\\ell\neq k}}^{b}X_{k\ell} = \sum_{1\leq\ell<k\leq b}(X_{k\ell}+X_{\ell k}), \quad (6.21)$$

we recast (6.20) as follows:

$$|\Psi^{(2)}\rangle = T_{13}(\bar{u})\sum_{1\leq\ell<k\leq b}\lambda_2(v_k)\lambda_2(v_\ell)f(\bar{v}_{k,\ell},v_k)f(\bar{v}_{\ell,k},v_\ell)T_{23}(\bar{v}_{\ell,k})|0\rangle$$
$$\times\Big\{g(v_k,u_2)g(v_\ell,u_1)f(v_k,u_1)f(v_\ell,v_k) + g(v_\ell,u_2)g(v_k,u_1)f(v_\ell,u_1)f(v_k,v_\ell)\Big\}. \quad (6.22)$$

Comparing the expression in braces with (6.6) we see that

$$g(v_k,u_2)g(v_\ell,u_1)f(v_\ell,v_k)f(v_k,u_1) + g(v_k,u_2)g(v_k,u_1)f(v_k,v_\ell)f(v_\ell,u_1) = K_2(\{v_k,v_\ell\}|\bar{u}),$$
$$(6.23)$$

and we do reproduce (6.17).

It remains to check that

$$|\Psi^{(1)}\rangle = \sum_{\ell=1}^{b}W_\ell\, T_{23}(\bar{v}_\ell)|0\rangle, \quad (6.24)$$

where

$$W_\ell = \lambda_2(v_\ell)f(\bar{v}_\ell,v_\ell)\Big(T_{12}(u_2)T_{13}(u_1)g(v_\ell,u_1) + T_{13}(u_2)T_{12}(u_1)g(v_\ell,u_2)f(v_\ell,u_1)\Big). \quad (6.25)$$

Using commutation relations (1.29) we find

$$T_{12}(u_2)T_{13}(u_1) = f(u_1,u_2)T_{13}(u_1)T_{12}(u_2) + g(u_2,u_1)T_{13}(u_2)T_{12}(u_1). \tag{6.26}$$

Substituting this into (6.25) we arrive at

$$W_\ell = \lambda_2(v_\ell)f(\bar{v}_\ell,v_\ell)\Big(T_{13}(u_1)T_{12}(u_2)f(u_1,u_2)g(v_\ell,u_1)$$
$$+ T_{13}(u_2)T_{12}(u_1)\big[g(v_\ell,u_2)f(v_\ell,u_1) + g(u_2,u_1)g(v_\ell,u_1)\big]\Big). \tag{6.27}$$

Simple straightforward calculation shows that

$$g(v_\ell,u_2)f(v_\ell,u_1) + g(u_2,u_1)g(v_\ell,u_1) = f(u_2,u_1)g(v_\ell,u_2). \tag{6.28}$$

Substituting this into (6.27) we obtain (6.15). Thus, we have convinced our selves that recursion (5.25) does give the Bethe vector (6.8) for $a = 2$.

In concluding this section, we note that explicit formulas for the $\mathcal{R}_N$ Bethe vectors in terms of polynomials in the creation operators acting on the vacuum vector were obtained in [43].

## 6.3 Alternative representation for Bethe vectors

Along with the representation (6.8), there is another representation for the Bethe vectors:

$$|\Psi_{a,b}(\bar{u};\bar{v})\rangle = \sum_{\substack{\bar{u}\mapsto\{\bar{u}_{\mathrm{I}},\bar{u}_{\mathrm{II}}\} \\ \bar{v}\mapsto\{\bar{v}_{\mathrm{I}},\bar{v}_{\mathrm{II}}\}}} K_n(\bar{v}_{\mathrm{I}}|\bar{u}_{\mathrm{I}})f(\bar{u}_{\mathrm{I}},\bar{u}_{\mathrm{II}})f(\bar{v}_{\mathrm{II}},\bar{v}_{\mathrm{I}})T_{13}(\bar{v}_{\mathrm{I}})T_{23}(\bar{v}_{\mathrm{II}})T_{12}(\bar{u}_{\mathrm{II}})\lambda_2(\bar{u}_{\mathrm{I}})|0\rangle. \tag{6.29}$$

Here the sum is taken over partitions $\bar{u}\mapsto\{\bar{u}_{\mathrm{I}},\bar{u}_{\mathrm{II}}\}$ and $\bar{v}\mapsto\{\bar{v}_{\mathrm{I}},\bar{v}_{\mathrm{II}}\}$ like in (6.8).

In order to prove (6.29) we first prove the following proposition.

**Proposition 6.2.** *Let us extend the action of the automorphism* (1.28) *on the vectors by*

$$\varphi\big(|0\rangle\big) = |0\rangle,$$
$$\varphi\big(T_{ij}(u)|0\rangle\big) = \varphi\big(T_{ij}(u)\big)|0\rangle, \quad \varphi\big(\lambda_i(u)\big) = \lambda_{4-i}(-u). \tag{6.30}$$

*Then*

$$\varphi\big(|\Psi_{a,b}(\bar{u};\bar{v})\rangle\big) = |\widetilde{\Psi}_{b,a}(-\bar{v};-\bar{u})\rangle, \tag{6.31}$$

*where* $|\widetilde{\Psi}_{b,a}(-\bar{v};-\bar{u})\rangle$ *is the Bethe vector corresponding to the monodromy matrix* $\tilde{T}(z)$.

*Proof.* We use induction over $a$. We have for $a = 0$ and $b$ arbitrary

$$\varphi\big(|\Psi_{0,b}(\emptyset;\bar{v})\rangle\big) = \varphi\big(T_{23}(\bar{v})|0\rangle\big) = \tilde{T}_{12}(-\bar{v})|0\rangle = |\widetilde{\Psi}_{b,0}(-\bar{v};\emptyset)\rangle, \tag{6.32}$$

where we used (3.80) and (5.29).

Assume that (6.31) holds for some $a-1 \geq 0$ and $b$ arbitrary. Applying the automorphism $\varphi$ to the recursion (5.25) we obtain

$$\varphi\big(|\Psi_{a,b}(\bar{u};\bar{v})\rangle\big) = \varphi\big(T_{12}(u_1)\big)\varphi\big(|\Psi_{a-1,b}(\bar{u}_1;\bar{v})\rangle\big)$$
$$+ \varphi\big(T_{13}(u_1)\big)\sum_{\ell=1}^{b}\varphi\big(\lambda_2(v_\ell)\big)g(v_\ell,u_1)f(v_\ell,\bar{u}_1)f(\bar{v}_\ell,v_\ell)\varphi\big(|\Psi_{a-1,b-1}(\bar{u}_1;\bar{v}_\ell)\rangle\big). \tag{6.33}$$

Due to induction assumption we find

$$\varphi\big(|\Psi_{a,b}(\bar{u};\bar{v})\rangle\big) = T_{23}(-u_1)|\widetilde{\Psi}_{b,a-1}(-\bar{v},-\bar{u}_1)\rangle$$
$$+ T_{13}(-u_1)\sum_{\ell=1}^{b}\lambda_2(-v_\ell)g(v_\ell,u_1)f(v_\ell,\bar{u}_1)f(\bar{v}_\ell,v_\ell)|\widetilde{\Psi}_{b-1,a-1}(-\bar{v}_\ell,-\bar{u}_1)\rangle. \quad (6.34)$$

Setting $a' = b$, $b' = a$, $\bar{u}' = -\bar{v}$, and $\bar{v}' = -\bar{u}$, one obtains

$$\varphi\big(|\Psi_{a,b}(\bar{u};\bar{v})\rangle\big) = T_{23}(v_1')|\widetilde{\Psi}_{a',b'-1}(\bar{u}',\bar{v}_1')\rangle$$
$$+ T_{13}(v_1')\sum_{\ell=1}^{a'}\lambda_2(u_\ell')g(-u_\ell',-v_1')f(-u_\ell',-\bar{v}_1')f(-\bar{u}_\ell',-u_\ell')|\widetilde{\Psi}_{a'-1,b'-1}(\bar{u}_\ell',\bar{v}_1')\rangle. \quad (6.35)$$

Using $g(-v,-u) = g(u,v)$ and $f(-v,-u) = f(u,v)$ we arrive at

$$\varphi\big(|\Psi_{a,b}(\bar{u};\bar{v})\rangle\big) = T_{23}(v_1')|\widetilde{\Psi}_{a',b'-1}(\bar{u}',\bar{v}_1')\rangle$$
$$+ T_{13}(v_1')\sum_{\ell=1}^{a'}\lambda_2(u_\ell')g(v_1',u_\ell')f(\bar{v}_1',u_\ell')f(u_\ell',\bar{u}_\ell')|\widetilde{\Psi}_{a'-1,b'-1}(\bar{u}_\ell',\bar{v}_1')\rangle. \quad (6.36)$$

One recognizes in the r.h.s. of (6.36) the recursion formula[7] (5.28) for $|\widetilde{\Psi}_{a',b'}(\bar{u}';\bar{v}')\rangle$. Hence,

$$\varphi\big(|\Psi_{a,b}(\bar{u};\bar{v})\rangle\big) = |\widetilde{\Psi}_{a',b'}(\bar{u}';\bar{v}')\rangle = |\widetilde{\Psi}_{b,a}(-\bar{v};-\bar{u})\rangle. \quad (6.37)$$

$\square$

Now we are ready to prove representation (6.29). We start with equation (6.8) for the Bethe vector $|\widetilde{\Psi}_{b,a}(\bar{v};\bar{u})\rangle$:

$$|\widetilde{\Psi}_{b,a}(\bar{v};\bar{u})\rangle = \sum_{\substack{\bar{u}\mapsto\{\bar{u}_{\mathrm{I}},\bar{u}_{\mathrm{II}}\}\\\bar{v}\mapsto\{\bar{v}_{\mathrm{I}},\bar{v}_{\mathrm{II}}\}}} K_n(\bar{u}_{\mathrm{I}}|\bar{v}_{\mathrm{I}})f(\bar{v}_{\mathrm{I}},\bar{v}_{\mathrm{II}})f(\bar{u}_{\mathrm{II}},\bar{u}_{\mathrm{I}})\tilde{T}_{13}(\bar{v}_{\mathrm{I}})\tilde{T}_{12}(\bar{v}_{\mathrm{II}})\tilde{T}_{23}(\bar{u}_{\mathrm{II}})\tilde{\lambda}_2(\bar{u}_{\mathrm{I}})|0\rangle. \quad (6.38)$$

It is easy to see that equations (1.28), (6.30) imply $\varphi^2 = 1$, and hence,

$$\varphi(|\widetilde{\Psi}_{b,a}(\bar{v};\bar{u})\rangle) = \varphi^2(|\Psi_{a,b}(-\bar{u};-\bar{v})\rangle) = |\Psi_{a,b}(-\bar{u};-\bar{v})\rangle. \quad (6.39)$$

Thus, acting with $\varphi$ on (6.38) we obtain

$$|\Psi_{a,b}(-\bar{u};-\bar{v})\rangle = \sum_{\substack{\bar{u}\mapsto\{\bar{u}_{\mathrm{I}},\bar{u}_{\mathrm{II}}\}\\\bar{v}\mapsto\{\bar{v}_{\mathrm{I}},\bar{v}_{\mathrm{II}}\}}} K_n(\bar{u}_{\mathrm{I}}|\bar{v}_{\mathrm{I}})f(\bar{v}_{\mathrm{I}},\bar{v}_{\mathrm{II}})f(\bar{u}_{\mathrm{II}},\bar{u}_{\mathrm{I}})T_{13}(-\bar{v}_{\mathrm{I}})T_{23}(-\bar{v}_{\mathrm{II}})T_{12}(-\bar{u}_{\mathrm{II}})\lambda_2(-\bar{u}_{\mathrm{I}})|0\rangle.$$
$$(6.40)$$

It follows form $g(-u,-v) = g(v,u)$ and $f(-u,-v) = f(v,u)$ that $K_n(-\bar{v}|-\bar{u}) = K_n(\bar{u}|\bar{v})$. Thus, changing $\bar{u} \to -\bar{u}$ and $\bar{v} \to -\bar{v}$ in (6.40) we immediately arrive at (6.29).

## 6.4 Commutation relations and Bethe vectors

Comparing representations (6.8) and (6.29) we see that we deal with different ordering of the operators in these formulas. Nevertheless, these formulas are equivalent.

---

[7]One can replace $v_b$ in (5.28) by any other $v_k$ due to the symmetry of Bethe vectors over $\bar{v}$.

Let us consider a simple but non-trivial example of the Bethe vector $|\Psi_{1,1}(u;v)\rangle$. Using (6.8) and (6.29) we respectively obtain

$$|\Psi_{1,1}(u;v)\rangle = T_{12}(u)T_{23}(v)|0\rangle + g(v,u)T_{13}(u)\lambda_2(v)|0\rangle, \qquad (6.41)$$

and

$$|\Psi_{1,1}(u;v)\rangle = T_{23}(v)T_{12}(u)|0\rangle + g(v,u)T_{13}(v)\lambda_2(u)|0\rangle. \qquad (6.42)$$

It is easy to see that (6.41) and (6.42) do coincide. Indeed, due to the commutation relations (1.29) we have

$$[T_{12}(u), T_{23}(v)] = g(u,v)\big(T_{13}(u)T_{22}(v) - T_{13}(v)T_{22}(u)\big), \qquad (6.43)$$

or equivalently

$$T_{12}(u)T_{23}(v) + g(v,u)T_{13}(u)T_{22}(v) = T_{23}(v)T_{12}(u) + g(v,u)T_{13}(v)T_{22}(u). \qquad (6.44)$$

Applying (6.44) to the vacuum vector we obtain

$$T_{12}(u)T_{23}(v)|0\rangle + g(v,u)T_{13}(u)\lambda_2(v)|0\rangle = T_{23}(v)T_{12}(u)|0\rangle + g(v,u)T_{13}(v)\lambda_2(u)|0\rangle. \qquad (6.45)$$

A similar effect takes place in the case of general Bethe vectors $|\Psi_{a,b}(\bar{u};\bar{v})\rangle$. Namely, one can prove [72] that

$$\sum_{\substack{\bar{u}\mapsto\{\bar{u}_{\mathrm{I}},\bar{u}_{\mathrm{II}}\} \\ \bar{v}\mapsto\{\bar{v}_{\mathrm{I}},\bar{v}_{\mathrm{II}}\}}} K_n(\bar{v}_{\mathrm{I}}|\bar{u}_{\mathrm{I}})f(\bar{u}_{\mathrm{I}},\bar{u}_{\mathrm{II}})f(\bar{v}_{\mathrm{II}},\bar{v}_{\mathrm{I}})$$

$$\times \Big(T_{13}(\bar{u}_{\mathrm{I}})\,T_{12}(\bar{u}_{\mathrm{II}})\,T_{23}(\bar{v}_{\mathrm{II}})\,T_{22}(\bar{v}_{\mathrm{I}}) - T_{13}(\bar{v}_{\mathrm{I}})\,T_{23}(\bar{v}_{\mathrm{II}})\,T_{12}(\bar{u}_{\mathrm{II}})\,T_{22}(\bar{u}_{\mathrm{I}})\Big) = 0. \qquad (6.46)$$

Here the sum is taken over partitions $\bar{u} \mapsto \{\bar{u}_{\mathrm{I}}, \bar{u}_{\mathrm{II}}\}$ and $\bar{v} \mapsto \{\bar{v}_{\mathrm{I}}, \bar{v}_{\mathrm{II}}\}$ such that $\#\bar{u}_{\mathrm{I}} = \#\bar{v}_{\mathrm{I}} = n$ and $n = 0, 1, \ldots, \min(a,b)$, where $a = \#\bar{u}$ and $b = \#\bar{v}$.

On the one hand, acting with (6.46) on $|0\rangle$ we immediately prove the equivalence of the representations (6.8) and (6.29). On the other hand, (6.46) can be considered as a multiple commutation relation between the products $T_{23}(\bar{v})$ and $T_{12}(\bar{u})$. Indeed, extracting explicitly the term $n = 0$ we recast (6.46) as follows:

$$[T_{23}(\bar{v}), T_{12}(\bar{u})] = \sum_{\substack{\bar{u}\mapsto\{\bar{u}_{\mathrm{I}},\bar{u}_{\mathrm{II}}\} \\ \bar{v}\mapsto\{\bar{v}_{\mathrm{I}},\bar{v}_{\mathrm{II}}\} \\ n>0}} K_n(\bar{v}_{\mathrm{I}}|\bar{u}_{\mathrm{I}})f(\bar{u}_{\mathrm{I}},\bar{u}_{\mathrm{II}})f(\bar{v}_{\mathrm{II}},\bar{v}_{\mathrm{I}})$$

$$\times \Big(T_{13}(\bar{u}_{\mathrm{I}})\,T_{12}(\bar{u}_{\mathrm{II}})\,T_{23}(\bar{v}_{\mathrm{II}})\,T_{22}(\bar{v}_{\mathrm{I}}) - T_{13}(\bar{v}_{\mathrm{I}})\,T_{23}(\bar{v}_{\mathrm{II}})\,T_{12}(\bar{u}_{\mathrm{II}})\,T_{22}(\bar{u}_{\mathrm{I}})\Big). \qquad (6.47)$$

Thus, we see that the explicit representations for the Bethe vectors in the $\mathfrak{gl}_3$ based models are closely related to the multiple commutation relations. Whether this correspondence exists in the models with higher rank of symmetry is an open question.

## Summary

We have considered the basic principles of NABA. The main example for us were models with the $\mathfrak{gl}_3$-invariant $R$-matrix. However, in more general cases, the principle scheme for obtaining

off-shell and on-shell Bethe vectors persists. In particular, this scheme works in models with the $q$-deformed $R$-matrix (1.14), as well as in the models based on graded algebras [12, 43, 73, 74].

Further development of NABA is associated with the application of this method to the calculation of the matrix elements of local operators and correlation functions. In these matters, however, there are still many unsolved problems. Basically these problems are of a technical nature and involve very non-trivial representations for the Bethe vectors. Partially these problems are solved in the models with the $\mathfrak{gl}_3$-invariant $R$-matrix, as well as its graded and $q$-deformed analogues [50, 75, 76].

# Acknowledgements

I am grateful to the organizers of Les Houches summer school 2018 *Integrability in Atomic and Condensed Matter Physics* for hospitality and beautiful scientific atmosphere. I also thank all listeners of the School for their attention, patience, and questions that have allowed me to improve my notes.

# A   Transformation of $R$-matrices

**Proposition A.1.** *Let $K\left(\frac{u}{v}\right)$ be given by (1.19) and let $R(u,v)$ be a solution to the Yang–Baxter equation such that $[R_{12}(u,v), K_1\left(\frac{u}{v}\right)K_2\left(\frac{u}{v}\right)] = 0$. Then matrices*

$$\tilde{R}_{12}(u,v) = \begin{cases} K_1\left(\frac{u}{v}\right)R_{12}(u,v)K_1\left(\frac{v}{u}\right), \\ K_1\left(\frac{v}{u}\right)R_{12}(u,v)K_1\left(\frac{u}{v}\right), \end{cases} \tag{A.1}$$

*also solve the Yang–Baxter equation.*

*Proof.* Define

$$\tilde{R}_{12}(u,v) = K_1\left(\frac{u}{v}\right)R_{12}(u,v)K_1\left(\frac{v}{u}\right). \tag{A.2}$$

Let us show that $\tilde{R}_{12}(u,v)$ satisfies the Yang–Baxter equation. Let

$$\begin{aligned} \Lambda &\equiv \tilde{R}_{12}(u,v)\tilde{R}_{13}(u,w)\tilde{R}_{23}(v,w) \\ &= K_1\left(\frac{u}{v}\right)R_{12}(u,v)K_1\left(\frac{v}{u}\right)K_1\left(\frac{u}{w}\right)R_{13}(u,w)K_1\left(\frac{w}{u}\right)K_2\left(\frac{v}{w}\right)R_{23}(v,w)K_2\left(\frac{w}{v}\right). \end{aligned} \tag{A.3}$$

Obviously $K_1\left(\frac{v}{u}\right)K_1\left(\frac{u}{w}\right) = K_1\left(\frac{v}{w}\right)$. We also have

$$\begin{aligned} &[K_1\left(\frac{w}{u}\right), K_2\left(\frac{v}{w}\right)] = 0, \\ &[K_1\left(\frac{w}{u}\right), R_{23}(v,w)] = 0, \\ &[R_{13}(u,w), K_2\left(\frac{v}{w}\right)] = 0, \end{aligned} \tag{A.4}$$

since in all these commutation relations we deal with matrices acting in different spaces. Then we obtain

$$\Lambda = K_1\left(\frac{u}{v}\right)R_{12}(u,v)K_1\left(\frac{v}{w}\right)K_2\left(\frac{v}{w}\right)R_{13}(u,w)R_{23}(v,w)K_1\left(\frac{w}{u}\right)K_2\left(\frac{w}{v}\right). \tag{A.5}$$

The product $K_1\left(\frac{v}{w}\right)K_2\left(\frac{v}{w}\right)$ can be moved further to the left due to

$$\left[R_{12}(u,v), K_1\left(\frac{v}{w}\right)K_2\left(\frac{v}{w}\right)\right] = 0, \tag{A.6}$$

and we arrive at

$$\Lambda = K_1\left(\tfrac{u}{w}\right)K_2\left(\tfrac{v}{w}\right)R_{12}(u,v)R_{13}(u,w)R_{23}(v,w)K_1\left(\tfrac{w}{u}\right)K_2\left(\tfrac{w}{v}\right). \tag{A.7}$$

Now we use the Yang–Baxter equation for $R$:

$$\Lambda = K_1\left(\tfrac{u}{w}\right)K_2\left(\tfrac{v}{w}\right)R_{23}(v,w)R_{13}(u,w)R_{12}(u,v)K_1\left(\tfrac{w}{u}\right)K_2\left(\tfrac{w}{v}\right). \tag{A.8}$$

We present $K_1\left(\tfrac{w}{u}\right)=K_1\left(\tfrac{w}{v}\right)K_1\left(\tfrac{v}{u}\right)$ and move the combination $K_1\left(\tfrac{w}{v}\right)K_2\left(\tfrac{w}{v}\right)$ to the left

$$\Lambda = K_1\left(\tfrac{u}{w}\right)K_2\left(\tfrac{v}{w}\right)R_{23}(v,w)R_{13}(u,w)K_1\left(\tfrac{w}{v}\right)K_2\left(\tfrac{w}{v}\right)R_{12}(u,v)K_1\left(\tfrac{v}{u}\right). \tag{A.9}$$

Moving $K_1\left(\tfrac{u}{w}\right)$ to the right and $K_2\left(\tfrac{w}{v}\right)$ to the left we obtain

$$\Lambda = K_2\left(\tfrac{v}{w}\right)R_{23}(v,w)K_2\left(\tfrac{w}{v}\right)K_1\left(\tfrac{u}{w}\right)R_{13}(u,w)K_1\left(\tfrac{w}{v}\right)R_{12}(u,v)K_1\left(\tfrac{v}{u}\right). \tag{A.10}$$

It remains to present $K_1\left(\tfrac{w}{v}\right)=K_1\left(\tfrac{w}{u}\right)K_1\left(\tfrac{u}{v}\right)$ and we finally arrive at

$$\Lambda = \tilde{R}_{23}(v,w)\tilde{R}_{13}(u,w)\tilde{R}_{12}(u,v). \tag{A.11}$$

Thus, $\tilde{R}(u,v)$ solves the Yang–Baxter equation. Using the fact that

$$K^{-1}\left(\tfrac{u}{v}\right)=K\left(\tfrac{v}{u}\right), \tag{A.12}$$

we obtain that

$$\hat{R}_{12}(u,v)=K_1\left(\tfrac{v}{u}\right)R_{12}(u,v)K_1\left(\tfrac{u}{v}\right) \tag{A.13}$$

also satisfies the Yang–Baxter equation. $\qquad\square$

# B  Calculation of traces

Let $X$ be a matrix acting in $\mathbb{C}^n$. Then

$$\operatorname{tr}(XE^{i,j})=\sum_{k,l=1}^{n}X^{k,l}\operatorname{tr}(E^{k,l}E^{i,j})=\sum_{k,l=1}^{n}X^{k,l}\operatorname{tr}(\delta_{il}E^{k,j})=\sum_{k=1}^{n}X^{k,i}\operatorname{tr}(E^{k,j})=X^{j,i}, \tag{B.1}$$

because $\operatorname{tr}E^{k,j}=\delta_{kj}$.

Now let $X$ be a matrix acting in the tensor product $V_1\otimes\cdots\otimes V_m$, where $V_j\sim\mathbb{C}^n$. Then

$$\operatorname{tr}_{1,\ldots,m}(XE_1^{i_1,j_1}\ldots E_m^{i_m,j_m})$$

$$=\sum_{\substack{k_1,\ldots,k_m=1\\ l_1,\ldots,l_m=1}}^{n}X^{k_1,l_1\ldots k_m,l_m}\operatorname{tr}_{1,\ldots,m}(E_1^{k_1,l_1}\ldots E_m^{k_m,l_m}E_1^{i_1,j_1}\ldots E_m^{i_m,j_m})$$

$$=\sum_{\substack{k_1,\ldots,k_m=1\\ l_1,\ldots,l_m=1}}^{n}X^{k_1,l_1\ldots k_m,l_m}\operatorname{tr}_{1,\ldots,m}(E_1^{k_1,j_1}\ldots E_m^{k_m,j_m})\delta_{i_1,l_1}\ldots\delta_{i_m,l_m}$$

$$=\sum_{k_1,\ldots,k_m=1}^{n}X^{k_1,i_1\ldots k_m,i_m}\operatorname{tr}_{1,\ldots,m}(E_1^{k_1,j_1}\ldots E_m^{k_m,j_m})=X^{j_1,i_1\ldots j_m,i_m}. \tag{B.2}$$

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
