# Peer review of "Introduction to the nested algebraic Bethe ansatz"

_SciPost Physics Lecture Notes, doi:SciPost Phys. Lect. Notes 19 (2020)_

## Round 1 · Referee Report · Anonymous (Referee 1) · 2020-5-25

Strengths

  1. The most complete detailed and rigorous presentation of the Nested Bethe Ansatz construction of the Bethe states.
  2. Focus on the notion of on-shell and off-shell Bethe states

Weaknesses

The presentation in its current form is more suitable for a review article than lecture notes

Report

This is the most complete and the most detailed presentation of the nested algebraic Bethe ansatz construction, and for this reason it is perfectly suitable for the ``les Houches series''. The lecture course is built around the concepts of on-shell and off-shell Bethe states. This point of view is the main originality and an evident strength of this lecture course as it opens a way for further study of form factors and correlation functions.

The only important improvement I would recommend is to add more explanation in the beginning of each section. In the current form the manuscript ressembles more to a review article than lecture notes. It is important to better explain the motivation and logic of each section (specially for the sections 4, 5 and 6). This motivation is clear for an experienced researcher working in the field, however it is much less evident for a new PhD student studying the subject and it should be kept in mind that it is the target audience of the ``les Houches series''. I would also suggest to move the appendices into the main body of the lectures, as it seems to me that this paper structure with 4 appendices once again is more suitable for a research paper than lecture notes (taking into account the fact that all the four appendices are not too long and perfectly fit each into corresponding section). In particular I would suggest to do it for the appendix B as it is crucial for all the following constructions.

There are also few minor remarks and typos listed below

Requested changes

1- Typo, line after (1.13): unit AT the intersection 2- In the first line of (1.29), there should be no tilde over T. 3- Also in (1.29) the bar notation for the indices can lead to a confusion with the set notations used throughout the manuscript 4- The phrase after (1.40) can lead to a confusion (using of either and or) as both terms always appear. 5- The fact stated between (2.1) and (2.2) that a local operator acting on an on-shell Bethe state produces a linear combination of off-shell states is not that evident, it should be better explained and appropriate citations added. 6- Typo in the phrase before (2.6), should be probably: there exist OTHER representation 7- Remark 2 in the section 2.2 is very important for the future presentation, maybe it can be better illustrated with examples (1.22) and (1.26) already introduced in the first section? 8- Beginning of 3.3 "Let we" should be replaced 9- May be it is better to reproduce (3.38) after (3.60) to have all the nested Bethe equations together 10- Before (3.65): may be a little bit more comments on the difference with the restricted case here would be useful. 11- Definition 3.1, typo: AN off-shell vector 12- After (3.78), typo: becomes AN on-shell state 13- Last phrase before the Conclusion, typo: OPEN question 14- Careful proof-reading would be welcome

  • validity: top
  • significance: high
  • originality: high
  • clarity: good
  • formatting: excellent
  • grammar: reasonable

Author:  Nikita Slavnov  on 2020-06-19  [id 857]

(in reply to Report 1 on 2020-05-25)

I am grateful to Referee for careful reading of my notes and valuable remarks. I believe that Referee found serious disadvantages in my manuscript. This primarily concerns the lack of motivation and explanation in the beginning of the sections. A very important remark is the fact that the action of operators on the on-shell Bethe vectors, generally speaking, is not necessarily expressed through the off-shell Bethe vectors. The suggestion to move the appendices to the main text is not obvious for me. I agree that such a construction of the text of the article is more consistent with the lecture notes. However, sometimes this strongly violates the presentation logic. I think that appendices B and D can be moved to the text. At the same time, I prefer to keep appendices A and C in their present form (that is, as appendices).

---

## Round 1 · Referee Report · Anonymous (Referee 2) · 2020-6-5

Strengths

1- Comprehensive review of the mathematics of the NABA by an expert in the field.

2- Reference for the analysis of integrable quantum chains with higher rank symmetries

Weaknesses

1- Better motivation could ease access to the topic.

Report

The manuscript is a comprehensive presentation of the nested algebraic Bethe ansatz method (NABA) mostly using the $gl_3$-invariant (rational) $R$-matrix as an example. Both the 'classical' scheme of Kulish and Reshetikhin and the representation of off-shell Bethe states in terms of trace formulae are covered. Explicit forms of Bethe states are obtained based on recursion relations satisfied between Bethe vectors in different color sectors. This provides the full set of tools used in the recent approaches to the computation of form factors correlation functions for quantum chains with higher rank symmetries.

The paper's focus is on mathematical aspects of the NABA (as stated in the introduction) and it certainly provides a reference for anybody working in the field. However, it would be more accessible to those trying to learn the methods if more motivation e.g. by relating to the mathematical challenges in obtaining insights into physical problems would be given.
  • validity: top
  • significance: high
  • originality: high
  • clarity: high
  • formatting: good
  • grammar: reasonable

Author:  Nikita Slavnov  on 2020-06-19  [id 858]

(in reply to Report 2 on 2020-06-05)

I am grateful to Referee for careful reading of my notes and valuable remarks. I have somewhat expanded the physical interpretation of the NABA in Introduction and added some references in a new version of the manuscript. However, I prefer not to touch this topic too deeply, because I am not an expert in this field.

---

## Round 1 · Referee Report · Anonymous (Referee 3) · 2020-6-13

Strengths

1.) Detailed pedagogical introduction, accessible by students and beginners in the field.
2.) Written by an expert who himself contributes significantly to the development of the method.

Weaknesses

1.) No obvious weakness.

Report

The manuscript comprises lecture notes from a Les Houches summer school in 2018. The author lectured about the nested algebraic Bethe ansatz for models with rational gl(n) invariant R-matrices. For pedagogical reasons he mostly restricted himself to the simplest nested case n = 3. The focus of the lecture was on the construction of Bethe vectors. Questions concerning the spectrum of physical models were not considered.

Being a productive contributor to the presented subject for the last ten years, the author possesses considerable insight into the details, which he tries to share with his readers. The choice of the material is rather on the mathematical side as the Bethe vectors themselves have little physical meaning. But their knowledge may be a prerequisite for calculating more physical quantities like form factors or correlation functions. Some more comments on the latter might have rendered the manuscript more readable for the general audience. A proofreading by a native speaker might also add to the readability of the manuscript.

Besides these general remarks I would only suggest very minor corrections
listed below.

1.) It should be said below (1.7) that $\Lambda (z)$ is the transfer matrix eigenvalue

2.) There should be no tilde on the right hand side of the first equation (1.29)

3.) Below (1.31) the argument of $\tilde T$ should probably rather be $u$ instead of $-u$

4.) Above (1.32): as generated $\rightarrow$ is generated

4.) I guess in more mathematical terms the coloring in section 1.7 is a grading

5.) Above (2.6): over presentations $\rightarrow$ other presentations

6.) Are the indices in (3.4) correct? They look strange to me, but perhaps I simply do not understand the conventions

7.) Section 3.3: Let we have $\rightarrow$ Let us consider

8.) Above (3.18): polylinear $\rightarrow$ linear

9.) Below (3.26): get read $\rightarrow$ get rid

10.) I guess the indices in (4.1) refer to mutually distinct spaces. If yes, it may add clarity to say this

11.) Summary section: vector Beta

12.) Right hand side of (A.11): $K_1$

Requested changes

see above

  • validity: top
  • significance: high
  • originality: -
  • clarity: high
  • formatting: excellent
  • grammar: reasonable

Author:  Nikita Slavnov  on 2020-06-19  [id 859]

(in reply to Report 3 on 2020-06-13)

I am grateful to Referee for careful reading of my notes and valuable remarks. I would only like to give a comment on the second remark 4. The coloring in the Yangian of the gl_n algebra is related to the eigenvalues of the Cartan generators of the gl_n algebra (see page 6 of https://arxiv.org/pdf/1704.08173.pdf). This has nothing to do with what is commonly called grading (see the beginning of section 2 of the same paper).

---

## Round 1 · Referee Report · Anonymous (Referee 4) · 2020-6-25

Strengths

These lecture notes provide a detailed and accessible description of the method of the Nested Algebraic Bethe Ansatz for higher rank integrable quantum chains, where different representations of Bethe vectors are shown to be equivalent.

Weaknesses

The introduction is in a draft status, some account of the mathematical and physical context should be detailed in order to motivate and ease the topic for a general audience.

Report

The manuscript contains a detailed description of the method of the Nested Algebraic Bethe Ansatz (NABA) for higher rank integrable quantum chains. Despite the algebraic complexity of the subject, I found it very well-structured and easily accessible. This is not a surprise being the author one key player in the development of several subjects here presented. The author mainly focus his presentation of NABA method for the models associated to the fundamental representation of the rational gl(3) invariant R-matrix also allowing for non-trivial gl(3) gauge transformations. The on-shell and off-shell Bethe vectors are constructed and different representations of them are shown to be equivalent and their specific interest pointed out. An interesting example is the trace form of Bethe vectors used to show their symmetry over the corresponding arguments. The gl(3) case is then used to explain the similarities or further complexity emerging for the representations of the Bethe vectors for the higher rank gl(n) cases.

Aside this very positive judgment on the manuscript (in particular on its potential to be accessible to non-experts), I find that it misses some important explanations of the mathematical and physical context. The author has been probably motivated by the aim to keep the presentation concise. However, I found that this results in the absence of fundamental existing results in the literature (some of them derived or co-authored by the same author), whose presentation is central to motivate the relevance of the topic for a general audience.
In the following I will list some non-exhaustive examples which from my point of view should be introduced to overcome the mentioned absences.

i) Some key references should be added when the author list in the introduction the applications:
“They include condensed matter physics … supersymmetric gauge theories, etc.”

ii) I think that when referring to the q-deformed R-matrices in section 1.3 the author should also cite the fundamental papers:
M. Jimbo. A q-difference analogue of U(g) and the Yang-Baxter equation. Lett. Math. Phys., 10:63-69, 1985.
V. G. Drinfel’d. Quantum groups. Proceedings of the International Congress of Mathematicians Berkeley, USA, 10:798-820, 1986.
which together with the paper [4] are at the origin of the quantum group mathematical subject.

iii) The introduction of section 2 is important to motivate the study of off-shell Bethe vectors and NABA in general. However, it misses some fundamental elements to make it intelligible to a wide audience. I suggest that some further statement based on the existing literature should be introduced to explain why and how the action of a (quasi-local) operator over an on-shell Bethe vector can be computed as linear combination of off-shell Bethe vectors. The reconstruction of local operators by matrix elements of the monodromy matrix dressed by products of transfer matrices, as first derived in Nucl. Phys. B 554 (1999) 647-678 and in Nucl.Phys. B575 (2000) 627-644 (even for the higher rank case), is fundamental at this aim. These local operator reconstructions also allow to point out the relevance of the form of the Bethe vectors in the algebraic version of Bethe Ansatz, as being written in terms of actions of off-diagonal elements of the monodromy matrices on reference states. Indeed, they are essential to compute algebraically, by using the Yang-Baxter commutation relations, the action of local operators over on-shell Bethe vectors.
As well the known results on the scalar products of on-shell times off-shell Bethe vectors allowing to compute (2.2) should also be mentioned. These types of results are at the basis of the success of the algebraic version of Bethe Ansatz to go beyond the spectrum description and compute exactly fundamental object like correlation functions. The author himself has given central contributions to some of them and I think that their description starting from the gl(2) up to the gl(3) case can bring the required clarification on the relevance of the topic to a wide audience.

iv) To clarify that the definition of the off-shell Bethe vectors admits some degree of ambiguity, the author refers to the papers [24,25] for the gl(2) case. Indeed, there is presented an example of different representations of Bethe vectors (by gauge transformed monodromy matrix elements) which are equivalent for on-shell Bethe vectors. One should however mention that the paper [24] is mainly devoted to the higher rank case and that one of its main features is to predict a non-nested form of the on-shell Bethe vectors. This non-nested higher rank form is still of the type (2.7) in terms of the B-operator of the Sklyanin’s quantum separation of variables for some twisted boundary conditions. The first proof for the gl(3) case that these are indeed transfer matrix eigenvectors and so on-shell Bethe vectors has been done in the NABA framework by the same author with Liashyk in the 2018. While in the paper “J.Math.Phys. 60 (2019) no.3, 032701” a proof is given for the general higher rank cases in the framework of the quantum separation of variable developed in “J.Math.Phys. 59 (2018), 091417”.
As one aim of the manuscript is to present different representations of Bethe vectors occurring in the gl(3) case it seems natural to me to point out also the existence of this non-nested one and the associated literature.

Requested changes

I strongly recommend the publication of these lecture notes once the underlined missing descriptions i)-iv) are covered.

---

## Round 2 · Author Response

We changed the text according to the referee's comments.

---

## Round 2 · List of Changes

We extend description of the application of the nested algebraic Bethe ansatz in physics.
We explain how off-shell Bethe vectors appear in calculations of matrix elements of local operators.
We give the explicit solution of the quantum inverse problem.
We mention a method to construct Bethe vectors via Sklyanin's B-operator.
The content of two appendices is moved to the main text.
The list of references is extended.
Typos are corrected
We explain how off-shell Bethe vectors appear in calculations of matrix elements of local operators.
We give the explicit solution of the quantum inverse problem.
We mention a method to construct Bethe vectors via Sklyanin's B-operator.
The content of two appendices is moved to the main text.
The list of references is extended.
Typos are corrected

---

## Editorial Decision

published